

# The Virasoro minimal string

Scott Collier[1,2], Lorenz Eberhardt[3], Beatrix Muehlmann[4] and Victor A. Rodriguez[5]

**1** Princeton Center for Theoretical Science, Princeton University, Princeton, NJ 08544, USA
**2** Center for Theoretical Physics, Massachusetts Institute of Technology,
Cambridge, MA 02139, USA
**3** School of Natural Sciences, Institute for Advanced Study, Princeton, NJ 08540, USA
**4** Department of Physics, McGill University Montréal, H3A 2T8, QC Canada
**5** Joseph Henry Laboratories, Princeton University, Princeton, NJ 08544, USA

⋆ sac@mit.edu , † elorenz@ias.edu ,
‡ beatrix.muehlmann@mcgill.ca , ◦ vrodriguez@princeton.edu

## Abstract

We introduce a critical string theory in two dimensions and demonstrate that this theory, viewed as two-dimensional quantum gravity on the worldsheet, is equivalent to a double-scaled matrix integral. The worldsheet theory consists of Liouville CFT with central charge $c \geq 25$ coupled to timelike Liouville CFT with central charge $26 - c$. The double-scaled matrix integral has as its leading density of states the universal Cardy density of primaries in a two-dimensional CFT, thus motivating the name Virasoro minimal string. The duality holds for any value of the continuous parameter $c$ and reduces to the JT gravity/matrix integral duality in the large central charge limit. It thus provides a precise stringy realization of JT gravity. The main observables of the Virasoro minimal string are quantum analogues of the Weil-Petersson volumes, which are computed as absolutely convergent integrals of worldsheet CFT correlators over the moduli space of Riemann surfaces. By exploiting a relation of the Virasoro minimal string to three-dimensional gravity and intersection theory on the moduli space of Riemann surfaces, we are able to give a direct derivation of the duality. We provide many checks, such as explicit numerical — and in special cases, analytic — integration of string diagrams, the identification of the CFT boundary conditions with asymptotic boundaries of the two-dimensional spacetime, and the matching between the leading non-perturbative corrections of the worldsheet theory and the matrix integral. As a byproduct, we discover natural conformal boundary conditions for timelike Liouville CFT.

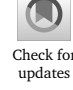

## Part I

# Introduction and summary

## 1  Introduction

String theories with a low number of target spacetime dimensions have proven to be valuable laboratories for understanding fundamental aspects of string theory. Rich phenomena such as holographic duality (for reviews, see [1–7]), non-perturbative effects mediated by D-instantons [8–20], and time-dependent stringy dynamics such as rolling tachyons [21–24], persist in low-dimensional string theories yet remain more computationally tractable than in their higher-dimensional counterparts.

At the same time, the direct approach of worldsheet string perturbation theory in the Polyakov formalism of integrating conformal field theory (CFT) correlators over the moduli space of Riemann surfaces, while being explicit and familiar, often obscures the underlying simplicity of the physics of the model. For instance, the two-dimensional $c = 1$ or type 0A/0B string theories admit a simpler description of the spacetime strings in terms of a double-scaled matrix quantum mechanics. Similarly, worldsheet theories of strings propagating in certain $AdS_3$ backgrounds are more simply described in terms of their spacetime boundary $CFT_2$ dual [25–29]. In these examples, the simpler and more illuminating description is the (spacetime) holographic dual.

Another important low-dimensional string theory model is the minimal string [30, 31], whose worldsheet theory is composed of a Virasoro minimal model CFT with central charge $\hat{c} < 1$ and Liouville CFT with $c > 25$ that together with the $\mathfrak{bc}$-ghost system form a critical worldsheet theory. This string model has been a fruitful arena for investigating aspects of two-dimensional quantum gravity and their relation to double-scaled matrix integrals [32–34] (for reviews, see [2, 35]). As a recent example, several works [36–39] have highlighted the $(2, p)$ minimal string as a candidate string-theoretic description of Jackiw–Teitelboim (or linear) dilaton quantum gravity in the $p \to \infty$ limit.

The main purpose of this paper is to investigate a new critical string theory that we will refer to as Virasoro minimal string theory, for reasons to be described below. When viewed as a model of two-dimensional quantum gravity on the worldsheet itself,[1] this theory admits

---

[1]See [40, 41] however, for a target spacetime interpretation of the worldsheet theory (1.1) for the particular

several distinct presentations that make its solvability more manifest. The Virasoro minimal string is defined by the following worldsheet conformal field theory,[2]

$$
\begin{array}{ccccc}
c \geq 25 & & \hat{c} \leq 1 & & \\
\text{Liouville CFT} & \oplus & \text{Liouville CFT} & \oplus & \mathfrak{bc}\text{-ghosts}\,,
\end{array}
\tag{1.1}
$$

where $\hat{c} = 26 - c$. Importantly, as described in more detail in section 3, the $\hat{c} \leq 1$ Liouville CFT sector of (1.1) is *not* simply the analytic continuation of the $c \geq 25$ Liouville CFT; rather, it is a distinct (non-unitary) solution to the CFT crossing equations for central charge in the range $\hat{c} \leq 1$ that has been independently bootstrapped [42–44]. It has sometimes been referred to as "timelike Liouville CFT" in the literature, and we will adopt that name here.

In contrast to minimal string theory, the Virasoro minimal string (1.1) is a *continuous* family of critical worldsheet theories labeled by a single parameter $c = 1 + 6(b + b^{-1})^2 \in \mathbb{R}_{\geq 25}$. Furthermore, the main observables of the theory, worldsheet CFT correlators integrated over moduli space of Riemann surfaces — or *quantum volumes* of the string worldsheet — have analytic dependence on both the parameter $c$ as well as the "external momenta" $P_i$ labeling the on-shell vertex operator insertions on the worldsheet. For example, we find for the four punctured sphere and the once punctured torus

$$
\mathsf{V}_{0,4}^{(b)}(P_1, P_2, P_3, P_4) = \frac{c-13}{24} + P_1^2 + P_2^2 + P_3^2 + P_4^2\,, \qquad \mathsf{V}_{1,1}^{(b)}(P_1) = \frac{c-13}{576} + \frac{1}{24}P_1^2\,.
\tag{1.2}
$$

Despite their origin as complicated integrals of CFT correlators over the moduli space of Riemann surfaces, the resulting quantum volumes are extraordinarily simple functions of the central charge and external momenta. This suggests that the theory admits a much simpler representation. Indeed, in the main part of this paper we will leverage such alternative descriptions to derive relations that make $\mathsf{V}_{g,n}^{(b)}$ accessible for arbitrary $g$ and $n$.

In this paper, we will show that in addition to the worldsheet CFT description (1.1), the Virasoro minimal string admits the following presentations: as a model of dilaton quantum gravity on the two-dimensional worldsheet subject to a sinh-dilaton potential; as a dimensional reduction of a certain sector of three-dimensional gravity; in terms of intersection theory on moduli space of Riemann surfaces; and in terms of a double-scaled matrix integral. These different presentations are summarized in figure 1.

The double-scaled matrix integral is perturbatively fully determined by its leading density of eigenvalues, which is given by

$$
\varrho_0^{(b)}(E)\,\mathrm{d}E = 2\sqrt{2}\,\frac{\sinh(2\pi b\sqrt{E})\sinh(2\pi b^{-1}\sqrt{E})}{\sqrt{E}}\,\mathrm{d}E\,,
\tag{1.3}
$$

where $E$ is the energy in the double-scaled matrix integral. Since (1.3) is the Cardy formula that universally governs the asymptotic density of states in any unitary compact $\text{CFT}_2$, we call (1.1) the Virasoro minimal string. In the limit $b \to 0$ (equivalently $c \to \infty$) and upon rescaling the energy $E$ the eigenvalue density of the Virasoro minimal string reduces to the $\sinh(\sqrt{E})\,\mathrm{d}E$ density of JT gravity. At finite values of $c$ the Virasoro minimal string (1.1) corresponds to a deformation of JT gravity, which is however completely distinct from the $(2, p)$ minimal string.

---

case of $\hat{c} = 1$ and $c = 25$ Liouville CFTs, as strings propagating in a two-dimensional cosmological background.

[2]A brief aside on terminology: we refer to this as "Virasoro minimal string theory" because it is in a sense the minimal critical worldsheet theory involving only ingredients from Virasoro representation theory. Another point of view is that any bosonic string theory without a tachyon defines a minimal string theory. In contrast to the ordinary minimal string, the word "minimal" should *not* be read as having anything to do with Virasoro minimal model CFTs.

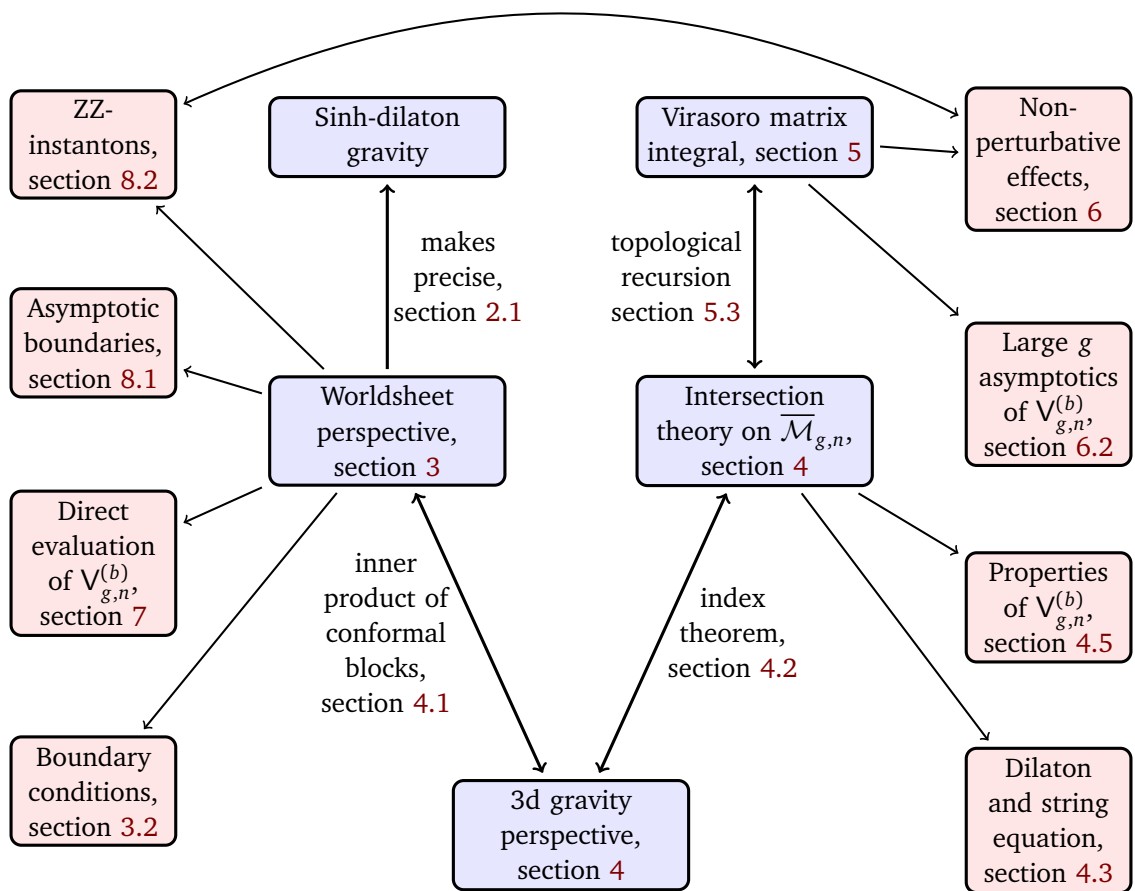

Figure 1: Road map of this paper. The Virasoro minimal string admits five different presentations summarized in the blue shaded boxes. The red shaded boxes refer to more details related to the presentation in consideration.

**Outline of this paper.** The rest of the paper is organized in four parts. In the first part we summarize the different presentations of (1.1) and highlight our main results following the structure outlined in figure 1. Part II is split into three sections: In section 3 we define the worldsheet theory (1.1). We describe the spacelike and timelike Liouville conformal field theories corresponding to the theories with central charge $c \geq 25$ and $\hat{c} \leq 1$ in the Virasoro minimal string (1.1). We introduce suitable boundary conditions which will allow us to study also configurations with asymptotic boundaries. In section 4 we provide a three-dimensional perspective of the Virasoro minimal string and derive a cohomological interpretation for the quantum volumes $\mathsf{V}_{g,n}^{(b)}$ using intersection theory technology on the compactified moduli space of Riemann surfaces, $\overline{\mathcal{M}}_{g,n}$. We introduce and discuss the dual matrix model in section 5. Topological recursion demonstrates the equivalence between the matrix model and the intersection theory expressions for $\mathsf{V}_{g,n}^{(b)}$. Part III contains further applications and direct checks of the Virasoro minimal string, such as a discussion of non-perturbative effects in section 6, the direct evaluation of string diagrams in section 7 and string diagrams in the presence of boundaries in section 8. We conclude in part IV with a discussion and a summary of open problems. Details of various calculations and conventions are summarized in appendices A, B, C and D.

## 2 Summary of results

### 2.1 Sinh-dilaton gravity

We begin by considering a two-dimensional theory of dilaton gravity. Its classical Euclidean action on a surface $\Sigma$ takes the form

$$S_\Sigma[g, \Phi] = -\frac{1}{2}\int_\Sigma \mathrm{d}^2x\,\sqrt{g}\left(\Phi\mathcal{R} + W(\Phi)\right) - \int_{\partial\Sigma}\mathrm{d}x\,\sqrt{h}\,\Phi(K-1)$$
$$-\frac{S_0}{2\pi}\left(\frac{1}{2}\int_\Sigma \mathrm{d}^2x\sqrt{g}\,\mathcal{R} + \int_{\partial\Sigma}\mathrm{d}x\,\sqrt{h}K\right),\quad W(\Phi) = \frac{\sinh(2\pi b^2\Phi)}{\sin(\pi b^2)}.\quad (2.1)$$

Here $S_0^{-1}$ plays the role of a gravitational coupling. The model reduces to JT gravity in the limit $b \to 0$, where the dilaton potential becomes linear [45, 46]. The second line in (2.1) is the Euler term which weighs different topologies according to their genus, see e.g. [36]. This theory has been considered before, see e.g. [37, 38, 47–49], but is not yet solvable by standard techniques, since it in particular falls outside the class of dilaton gravities considered in [39, 50–52]. We will not discuss the theory directly in the metric formulation. Instead, we will make use of the following field redefinition

$$\phi = b^{-1}\rho - \pi b\Phi,\qquad \chi = b^{-1}\rho + \pi b\Phi,\qquad (2.2)$$

where $\rho$ is the Weyl factor of the worldsheet metric $g = \mathrm{e}^{2\rho}\tilde{g}$. At the level of the classical actions, this maps the theory to the direct sum of a spacelike Liouville theory of central charge $c = 1 + 6(b + b^{-1})^2$ and a timelike Liouville theory of central charge $\hat{c} = 26 - c$. See [38, 47] for more details. We can thus describe the theory as a two-dimensional string theory with a spacelike Liouville theory coupled to a timelike Liouville theory. The classical actions of spacelike and timelike Liouville theory are respectively given by

$$S_L[\phi] = \frac{1}{4\pi}\int_\Sigma \mathrm{d}^2x\,\sqrt{\tilde{g}}\left(\tilde{g}^{ij}\partial_i\phi\,\partial_j\phi + Q\widetilde{\mathcal{R}}\phi + 4\pi\mu_{sL}\mathrm{e}^{2b\phi}\right),\qquad (2.3a)$$

$$S_{tL}[\chi] = \frac{1}{4\pi}\int_\Sigma \mathrm{d}^2x\,\sqrt{\tilde{g}}\left(-\tilde{g}^{ij}\partial_i\chi\,\partial_j\chi - \widehat{Q}\widetilde{\mathcal{R}}\chi + 4\pi\mu_{tL}\mathrm{e}^{2\hat{b}\chi}\right).\qquad (2.3b)$$

The dimensionless parameters $Q$, $b$ and $\widehat{Q}$, $\hat{b}$ and their relation with each other is explained in the next section; $\mu_{sL}$ and $\mu_{tL}$ are dimensionful parameters of the theory that satisfy $\mu_{sL} = -\mu_{tL}$.[3] We emphasize that although we have introduced these theories at the level of their worldsheet Lagrangians, in what follows we will treat them as non-perturbatively well-defined conformal field theories that together define the worldsheet CFT.

### 2.2 Worldsheet definition

The most direct description of the Virasoro minimal string is that of a critical bosonic worldsheet theory consisting of spacelike and timelike Liouville conformal field theories with paired central charges $c \geq 25$ and $\hat{c} \leq 1$ respectively, together with the usual $\mathfrak{bc}$-ghost system with central charge $c_{gh} = -26$. We emphasize that we view this string theory as a 2d theory of quantum gravity on the worldsheet (as opposed to a theory in target space), as depicted in figure 2.

---

[3]In the references [38,39,49,53,54], the timelike Liouville factor is replaced by a minimal model at the quantum level which then leads to the usual minimal string. In this paper, we will take the timelike Liouville factor seriously which leads to a completely different theory at the quantum level.

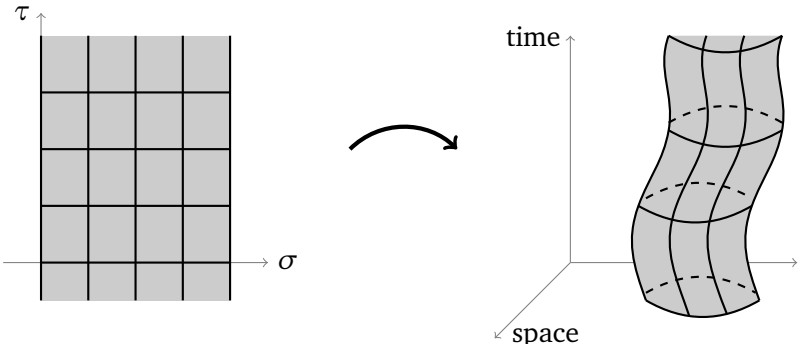

Quantum gravity on the worldsheet.          Strings in target spacetime.

Figure 2: A critical string background can be viewed as a model of quantum gravity on the two-dimensional worldsheet of the string, or as a model of strings propagating in target spacetime.

We refer to Liouville theory with $c \geq 25$ as spacelike Liouville theory whereas we refer to Liouville theory with $\hat{c} \leq 1$ as timelike Liouville theory [55–58]. This distinction is important as the CFT data of timelike Liouville theory is not simply the analytic continuation of that of spacelike Liouville theory. In this paper, we will place a typographical hat on quantities that refer to the timelike Liouville sector of the worldsheet theory (1.1) in order to distinguish them from those in the spacelike Liouville sector. We parametrize the central charges and the Virasoro conformal weights of their operator spectra by

$$\text{spacelike Liouville CFT:} \quad c = 1 + 6Q^2, \quad Q = b + b^{-1}, \quad h_P = \frac{Q^2}{4} + P^2, \tag{2.4a}$$

$$\text{timelike Liouville CFT:} \quad \hat{c} = 1 - 6\widehat{Q}^2, \quad \widehat{Q} = \hat{b}^{-1} - \hat{b}, \quad \hat{h}_{\widehat{P}} = -\frac{\widehat{Q}^2}{4} + \widehat{P}^2. \tag{2.4b}$$

The parameters $P$ and $\widehat{P}$ are often referred to as the "Liouville momenta." With this parametrization $b$ and $\hat{b}$ are real valued and we can choose $b, \hat{b} \in (0, 1]$. Both spacelike and timelike Liouville CFT are noncompact solutions to the crossing equations with a continuous spectrum of (delta-function normalizable) scalar primary operators with conformal weights bounded from *below* by $\frac{c-1}{24}$ and $\frac{\hat{c}-1}{24}$ respectively. This corresponds to real values of the Liouville momenta $P, \widehat{P}$. We defer a more comprehensive discussion of these worldsheet CFTs to section 3.1.

The Virasoro minimal string is described on the worldsheet by coupling a spacelike Liouville theory to a timelike Liouville theory, as described classically in (2.2). Vanishing of the conformal anomaly of the combined theory imposes the condition $\hat{c} = 26 - c$ and thus $\hat{b} = b$. The mass shell condition for physical states $h_P + \hat{h}_{\widehat{P}} = 1$ further implies $\widehat{P} = \pm iP$. In summary we have

$$\hat{b} = b, \quad \widehat{P} = iP, \tag{2.5}$$

where we chose one convention for the sign for concreteness. Hence, on-shell vertex operators in Virasoro minimal string theory involve external primary operators in timelike Liouville CFT with *imaginary* values of the Liouville momenta. Notably, imaginary values of $\widehat{P}$ correspond to $\hat{h} \leq \frac{\hat{c}-1}{24}$ and are thus *not* in the spectrum of timelike Liouville theory. Thus we will need to analytically continue the correlation functions of timelike Liouville theory away from real Liouville momenta. In fact this is a harmless operation and, contrary to spacelike Liouville theory, does not require contour deformations in the conformal block decomposition of worldsheet correlators. In [57], such an analytic continuation leads to the distinction of the *internal*

and *external* spectrum. A similar analytic continuation is also necessary for the usual minimal string — there, primaries of the Virasoro minimal model are combined with vertex operators in Liouville theory that are not in the spectrum and so their correlation functions are necessarily defined by analytic continuation.

We will denote the primary operators in the spacelike/timelike Liouville CFTs of conformal weights $h_P$ and $\hat{h}_{\widehat{P}}$ by $V_P(z)$ and $\widehat{V}_{\widehat{P}}(z)$ respectively. Physical operators of the full worldsheet theory are hence represented by the following vertex operators built out of paired primaries of the spacelike and timelike Liouville CFTs, together with $\mathfrak{bc}$-ghosts,

$$\mathcal{V}_P = \mathrm{N}(P)\,\mathfrak{c}\tilde{\mathfrak{c}}\,V_P\,\widehat{V}_{\widehat{P}=iP}\,, \tag{2.6}$$

where $\mathrm{N}(P)$ is a normalization constant that will be fixed in section 7.

The observables in Virasoro minimal string theory are computed by worldsheet diagrams as usual in string theory. For a worldsheet with genus $g$ and $n$ external punctures we define

$$\mathsf{V}_{g,n}^{(b)}(P_1,\dots,P_n) \equiv \int_{\mathcal{M}_{g,n}} Z_{\mathrm{gh}}\langle V_{P_1}\dots V_{P_n}\rangle_g\,\langle \widehat{V}_{iP_1}\dots\widehat{V}_{iP_n}\rangle_g\,. \tag{2.7}$$

Here $\langle V_{P_1}\dots V_{P_n}\rangle_g$ is the correlation function of $n$ primary operators on a genus-$g$ Riemann surface in spacelike Liouville CFT, $\langle \widehat{V}_{iP_1}\dots\widehat{V}_{iP_n}\rangle_g$ is the corresponding correlator in timelike Liouville CFT, $Z_{\mathrm{gh}}$ is the correlator of the $\mathfrak{bc}$-ghost system and the worldsheet CFT correlators are integrated over $\mathcal{M}_{g,n}$, the moduli space of genus-$g$ Riemann surfaces with $n$ punctures. We will typically consider the worldsheet diagrams for real values of the external momenta $P_j$, but we will see that the analytic continuation to complex momenta is often straightforward. A special feature of the Virasoro minimal string is that at least for real values of the external momenta, these diagrams are absolutely convergent integrals over the moduli space of Riemann surfaces. The Liouville momenta $P_j$ play a role analogous to that of the geodesic lengths in JT gravity, with $\mathsf{V}_{g,n}^{(b)}$ playing the role of the Weil-Petersson volumes. We shall discuss the precise reduction of $\mathsf{V}_{g,n}^{(b)}$ to the Weil-Petersson volumes in section 2.7. For this reason we will refer to $\mathsf{V}_{g,n}^{(b)}$ as "quantum volumes." In the full theory of quantum gravity, it is necessary to sum over all topologies which are weighted according to the Euler characteristic. We have

$$\mathsf{V}_n^{(b)}(S_0;P_1,\dots,P_n) \equiv \sum_{g=0}^{\infty} \mathrm{e}^{(2-2g-n)S_0}\,\mathsf{V}_{g,n}^{(b)}(P_1,\dots,P_n)\,. \tag{2.8}$$

This sum is asymptotic, but can be made sense of via resurgence.

Given the relationship between the Virasoro minimal string and two-dimensional dilaton gravity, it is natural to anticipate that it can compute observables with asymptotic boundaries in addition to the string diagrams with finite boundaries corresponding to external vertex operator insertions.[4] This is achieved on the worldsheet by equipping the worldsheet CFT with particular boundary conditions. We summarize the mechanism by which we incorporate asymptotic boundaries in section 2.5 and the precise worldsheet boundary conformal field theory in section 3.2. We in particular introduce a new family of conformal boundary conditions for timelike Liouville theory — which we dub "half-ZZ" boundary conditions — that will play an important role in the incorporation of asymptotic boundaries and in mediating non-perturbative effects in Virasoro minimal string theory.

---

[4]In the JT gravity limit, these finite boundaries become geodesic boundaries with lengths fixed in terms of the data of the vertex operator insertions as in (2.22). For this reason, in a slight abuse of notation, we will sometimes use the terms finite boundaries and geodesic boundaries interchangeably.

## 2.3 Dual matrix integral

The central claim of this paper is that the Virasoro minimal string is dual to a double-scaled Hermitian matrix integral. We will provide evidence that the leading density of states for this double-scaled matrix integral is given by

$$\varrho_0^{(b)}(E)\,dE = 2\sqrt{2}\,\frac{\sinh(2\pi b\sqrt{E})\sinh(2\pi b^{-1}\sqrt{E})}{\sqrt{E}}\,dE\,, \tag{2.9}$$

where $E = P^2 = h_P - \frac{c-1}{24}$ is the energy in the matrix integral. For $b \to 0$, one of the sinh's linearizes and we recover the famous $\sinh(\sqrt{E})\,dE$ density of states of JT gravity [36].

(2.9) is the universal normalized Cardy density of states in any unitary $\text{CFT}_2$, which is what motivated us to call the bulk theory the Virasoro minimal string. It is the modular S-matrix for the vacuum Virasoro character that controls the high energy growth of states in a $\text{CFT}_2$,

$$\chi_{\text{vac}}^{(b)}\left(-\tfrac{1}{\tau}\right) = \int_0^\infty dP\,\rho_0^{(b)}(P)\,\chi_P^{(b)}(\tau)\,, \quad \text{with} \quad \rho_0^{(b)}(P) \equiv 4\sqrt{2}\,\sinh(2\pi bP)\sinh(2\pi b^{-1}P)\,, \tag{2.10}$$

where $\chi_P^{(b)}(\tau) = q^{P^2}\eta(\tau)^{-1}$ are the non-degenerate Virasoro characters with weight $h_P$. Here $\tau$ is the torus modulus, with $q = e^{2\pi i\tau}$ and $\eta(\tau)$ the Dedekind eta function. The density of states is directly related to the spectral curve [59] which is the basic data for the topological recursion/loop equations in a double-scaled matrix integral. Since in recent CFT literature and the random matrix theory literature it is common to denote the densities of states by the same Greek letter, we distinguish the two cases by using $\rho_0^{(b)}$ in the CFT and $\varrho_0^{(b)}$ (2.9) in the matrix integral context.

The matrix integral associated to (2.9) turns out to be non-perturbatively unstable, unless $b = 1$. This is diagnosed by computing the first non-perturbative correction to the density of states. Perturbatively, no eigenvalue can be smaller than zero, but non-perturbatively, eigenvalues can tunnel to this classically forbidden regime. The leading non-perturbative contribution to the density of states in the forbidden $E < 0$ region takes the form

$$\langle\varrho^{(b)}(E)\rangle = -\frac{1}{8\pi E}\exp\left(2\sqrt{2}\,e^{S_0}\left(\frac{\sin(2\pi Q\sqrt{-E})}{Q} - \frac{\sin(2\pi\widehat{Q}\sqrt{-E})}{\widehat{Q}}\right)\right)\,, \tag{2.11}$$

where $Q$ and $\widehat{Q}$ were defined in (2.4). Unless $b = 1$, this can become arbitrarily large for sufficiently negative $E$ and thus renders the model unstable. One can define a non-perturbative completion of the matrix integral by modifying the integration contour over the eigenvalues of the matrices. Such a non-perturbative completion is ambiguous and any choice requires the inclusion of non-perturbative corrections to the gravity partition functions. These non-perturbative corrections correspond to ZZ-instanton corrections on the worldsheet and will be discussed in section 6.1. The worldsheet exhibits the same non-perturbative ambiguities, presumably related to the choice of integration contour in string field theory [60]. Via resurgence, the computation of non-perturbative effects allows us also to extract the large-genus asymptotics of the quantum volumes,

$$V_{g,n}^{(b)}(P_1,\ldots,P_n) \overset{g\gg 1}{\sim} \frac{\prod_{j=1}^n \frac{\sqrt{2}\sinh(2\pi bP_j)}{P_j}}{2^{\frac{3}{2}}\pi^{\frac{5}{2}}(1-b^4)^{\frac{1}{2}}} \times \left(\frac{4\sqrt{2}\,b\sin(\pi b^2)}{1-b^4}\right)^{2-2g-n} \times \Gamma\left(2g+n-\tfrac{5}{2}\right). \tag{2.12}$$

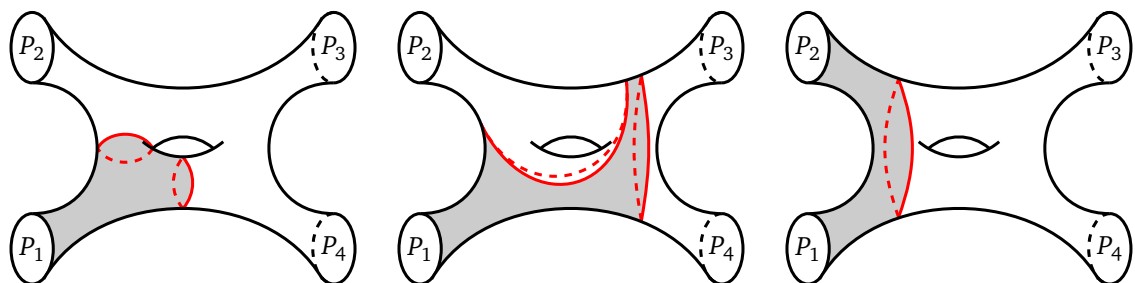

Figure 3: The three different ways of embedding a three-punctured sphere into a surface, corresponding to the three different contributions in eq. (2.13).

## 2.4 Deformed Mirzakhani recursion relation

Our conjecture for the dual matrix integral leads to recursion relations for the quantum volumes $V_{g,n}^{(b)}$. In particular we have

$$
P_1 V_{g,n}^{(b)}(P_1, \mathbf{P}) = \int_0^\infty (2P\,\mathrm{d}P)(2P'\,\mathrm{d}P')H(P+P',P_1)\bigg( V_{g-1,n+1}^{(b)}(P,P',\mathbf{P})
$$
$$
+ \sum_{h=0}^g \sum_{I \sqcup J = \{2,\ldots,n\}} V_{h,|I|+1}^{(b)}(P,\mathbf{P}_I)V_{g-h,|J|+1}^{(b)}(P',\mathbf{P}_J) \bigg)
$$
$$
+ \sum_{i=2}^n \int_0^\infty (2P\,\mathrm{d}P)\big(H(P,P_1+P_i)+H(P,P_1-P_i)\big)V_{g,n-1}^{(b)}(P,\mathbf{P}\setminus P_i), \quad (2.13)
$$

where $\mathbf{P} = (P_2, \ldots, P_n)$. The different terms correspond to the three topologically different ways in which one can embed a three-punctured sphere with boundary $P_1$ into $\Sigma_{g,n}$. They are displayed in figure 3. The function $H(x, y)$ takes the following form

$$
H(x,y) = \frac{y}{2} - \int_0^\infty \mathrm{d}t\, \frac{\sin(4\pi t x)\sin(4\pi t y)}{\sinh(2\pi b t)\sinh(2\pi b^{-1}t)}. \quad (2.14)
$$

The integral over $t$ is not elementary, except in special cases. For example, we have for $b = 1$

$$
H(x,y)\big|_{b=1} = \frac{-y\cosh(2\pi y) + x\sinh(2\pi y) + y\,\mathrm{e}^{-2\pi x}}{4\sinh(\pi(x+y))\sinh(\pi(x-y))}. \quad (2.15)
$$

This is a deformed version of Mirzakhani's celebrated recursion relation [61] to which it reduces in the limit $b \to 0$. We wrote an efficient implementation of this recursion relation in `Mathematica`, which is appended as an ancillary file to the submission.

## 2.5 Asymptotic boundaries

So far, we have only explained how to efficiently compute gravity partition functions with finite boundaries. One can add asymptotic boundaries just like in JT gravity by computing the partition function of a disk and of a punctured disk (aka trumpet) and glue them to the bulk volumes.

The disk and trumpet partition function take the form

$$
\mathcal{Z}_{\mathrm{disk}}^{(b)}(\beta) = \mathrm{e}^{\frac{\pi^2 c}{6\beta}} \prod_{n=2}^\infty \frac{1}{1-\mathrm{e}^{-\frac{4\pi^2 n}{\beta}}} = \frac{1}{\eta(\frac{\beta i}{2\pi})}\sqrt{\frac{2\pi}{\beta}}\left( \mathrm{e}^{\frac{\pi^2 Q^2}{\beta}} - \mathrm{e}^{\frac{\pi^2 \hat{Q}^2}{\beta}}\right), \quad (2.16a)
$$

$$
\mathcal{Z}_{\mathrm{trumpet}}^{(b)}(\beta;P) = \mathrm{e}^{-\frac{4\pi^2}{\beta}(P^2-\frac{1}{24})} \prod_{n=1}^\infty \frac{1}{1-\mathrm{e}^{-\frac{4\pi^2 n}{\beta}}} = \frac{1}{\eta(\frac{\beta i}{2\pi})}\sqrt{\frac{2\pi}{\beta}}\,\mathrm{e}^{-\frac{4\pi^2 P^2}{\beta}}. \quad (2.16b)
$$

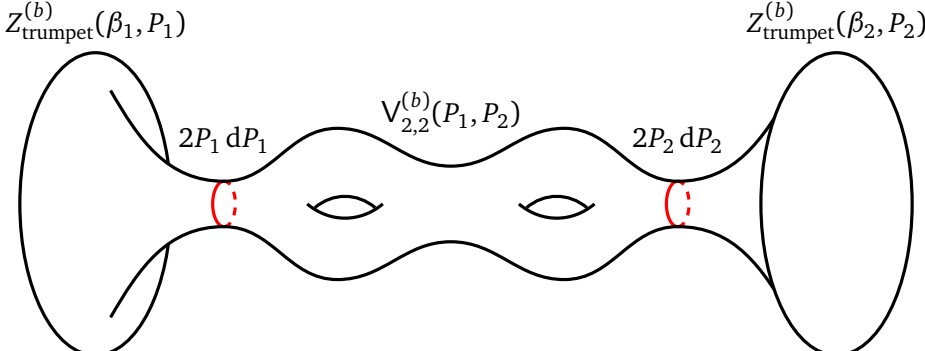

Figure 4: Gluing trumpets to the bulk gives the partition function of the Virasoro minimal string on arbitrary topologies with asymptotic boundaries.

From the first expression, one can recognize that these partition functions are simply the Virasoro vacuum character and non-vacuum character in the dual channel, respectively. In the second expression, we used the modular properties of the eta-function to rewrite it in terms of the $\beta$ channel.

The reason why the Virasoro character appears is that these 2d gravity partition functions are actually equal to a partition function of a chiral half of three-dimensional gravity theory on $\Sigma_{g,n} \times S^1$. We will explain this in section 4, where we derive these formulas. In our convention of $\beta$, the size of the thermal circle is $\frac{4\pi^2}{\beta}$. Thus, for the disk, we are actually computing the chiral 3d gravity partition function on a solid cylinder which gives the vacuum Virasoro character in the boundary. Similarly the trumpet partition function is equal to the 3d gravity partition function on a solid cylinder with a black hole inside, which gives a generic Virasoro character in the boundary.

The dual matrix integral explained in section 2.3 only captures the partition function of primaries. This should be intuitively clear since Virasoro descendants are dictated by symmetry and thus cannot be statistically independent from the primaries. We account for this by stripping off the factor $\eta(\frac{\beta i}{2\pi})$ and denote the primary partition functions by $Z^{(b)}$. Thus we have

$$Z_{\text{disk}}^{(b)}(\beta) = \sqrt{\frac{2\pi}{\beta}} \left( e^{\frac{\pi^2 Q^2}{\beta}} - e^{\frac{\pi^2 \hat{Q}^2}{\beta}} \right), \tag{2.17a}$$

$$Z_{\text{trumpet}}^{(b)}(\beta; P) = \sqrt{\frac{2\pi}{\beta}} \, e^{-\frac{4\pi^2 P^2}{\beta}} . \tag{2.17b}$$

The trumpet partition function has the same form as in JT gravity [36]. Taking the inverse Laplace transform of the disk partition function of the primaries $Z_{\text{disk}}^{(b)}$ leads to the eigenvalue distribution $\varrho_0^{(b)}$ given in equation (2.9), see subsection 5.2 for more details.

We can then compute the partition function with any number of asymptotic boundaries as follows

$$Z_{g,n}^{(b)}(\beta_1, \ldots, \beta_n) = \int_0^\infty \prod_{j=1}^n \left( 2P_j \, dP_j \, Z_{\text{trumpet}}^{(b)}(\beta_j, P_j) \right) V_{g,n}^{(b)}(P_1, \ldots, P_n). \tag{2.18}$$

Notice that the same measure $2P \, dP$ appears as in the deformed Mirzakhani's recursion relation (2.13). We derive this gluing measure from 3d gravity in section 4.1. Up to normalization, this is the same measure as in JT gravity. The gluing procedure is sketched in figure 4.

## 2.6 Intersection theory on moduli space

There is a last way to describe the theory – in terms of intersection theory on the compactified moduli space of Riemann surfaces $\overline{\mathcal{M}}_{g,n}$. This forms the conceptual bridge between the worldsheet description of section 2.2 and the description in terms of a random matrix integral in section 2.3 and allows us to essentially derive the duality.

From a bulk perspective, this also gives a far more efficient way to compute the integrals over $\mathcal{M}_{g,n}$ defined in (2.7), thanks to efficient algorithms to compute intersection numbers on moduli space. We used `admcycles` [62] in practice. We obtain with the intersection theory approach for example

$$V_{0,4}^{(b)}(P_1,\ldots,P_4) = \frac{c-13}{24} + \sum_{j=1}^{4} P_j^2, \tag{2.19a}$$

$$V_{1,1}^{(b)}(P_1) = \frac{1}{24}\left(\frac{c-13}{24} + P_1^2\right), \tag{2.19b}$$

$$V_{0,5}^{(b)}(P_1,\ldots,P_5) = \frac{5c^2 - 130c + 797}{1152} + \frac{c-13}{8}\sum_{j=1}^{5} P_j^2 + \frac{1}{2}\sum_{j=1}^{5} P_j^4 + 2\sum_{j<k} P_j^2 P_k^2, \tag{2.19c}$$

$$V_{1,2}^{(b)}(P_1,P_2) = \frac{c^2 - 26c + 153}{9216} + \frac{c-13}{288}(P_1^2 + P_2^2) + \frac{1}{48}(P_1^2 + P_2^2)^2. \tag{2.19d}$$

These can of course also be obtained from the recursion (2.13). We have compiled a much larger list of quantum volumes in appendix B.

Our main claim, which connects the worldsheet and matrix integral descriptions of the Virasoro minimal string, is that $V_{g,n}^{(b)}(P_1,\ldots,P_n)$ defined in eq. (2.7) is given by the following intersection number of $\overline{\mathcal{M}}_{g,n}$:

$$V_{g,n}^{(b)}(P_1,\ldots,P_n) = \int_{\overline{\mathcal{M}}_{g,n}} \mathrm{td}(\mathcal{M}_{g,n}) \exp\left(\frac{c}{24}\kappa_1 + \sum_{j=1}^{n}\left(P_j^2 - \frac{1}{24}\right)\psi_j\right)$$

$$= \int_{\overline{\mathcal{M}}_{g,n}} \exp\left(\frac{c-13}{24}\kappa_1 + \sum_{j=1}^{n} P_j^2\psi_j - \sum_{m\geq 1}\frac{B_{2m}}{(2m)(2m)!}\kappa_{2m}\right). \tag{2.20}$$

Here, $\psi_j$ and $\kappa_n$ are standard cohomology classes on $\overline{\mathcal{M}}_{g,n}$ whose definition we briefly recall in appendix A. $B_{2m}$ are the Bernoulli numbers. The Todd class of the tangent bundle of moduli space that appears in the first line, can be rewritten in terms of the $\psi$- and $\kappa$-classes via the Grothendieck-Riemann-Roch theorem, which leads to the expression in the second line.[5] Note that the integrand should be viewed as a formal power series. We expand the exponential and pick out the terms of the top degree $3g-3+n$ and integrate them over moduli space.

It is straightforward to derive two identities from (2.20) which are the analogue of the dilaton and string (or puncture) equations of topological gravity [63–65]. This requires some algebraic geometry and the proof can be found in appendix D. They take the form

$$V_{g,n+1}^{(b)}\left(P = \frac{i\widehat{Q}}{2}, \mathbf{P}\right) - V_{g,n+1}^{(b)}\left(P = \frac{iQ}{2}, \mathbf{P}\right) = (2g-2+n)V_{g,n}^{(b)}(\mathbf{P}), \tag{2.21a}$$

$$\int_{\frac{iQ}{2}}^{\frac{i\widehat{Q}}{2}} 2P\,\mathrm{d}P\, V_{g,n+1}^{(b)}(P,\mathbf{P}) = \sum_{j=1}^{n}\int_{0}^{P_j} 2P_j\,\mathrm{d}P_j\, V_{g,n}^{(b)}(\mathbf{P}). \tag{2.21b}$$

---

[5]Here it is important whether we talk about the Todd class of the tangent bundle of $\mathcal{M}_{g,n}$ or $\overline{\mathcal{M}}_{g,n}$, since they differ in their behaviour near the boundary of moduli space. We will mention further details about this subtlety in section 4.2.

To state these formulas, one has to analytically continue the quantum volumes to complex values of $P_i$. We used the parametrization (2.4). These two equations together with polynomiality of the quantum volumes that follows from the intersection expression (2.20) determine them completely at genus 0 and 1 [63].

## 2.7 Relation to JT gravity and the minimal string

As already noticed at the level of the action (2.1) or the density of states for the dual matrix integral (2.9), the Virasoro minimal string reduces to JT gravity in the limit $b \to 0$. JT gravity has been studied extensively in the literature, see [36] and many subsequent works. This reduction precisely realizes an idea of Seiberg and Stanford about the relation between the minimal string and JT gravity [37].

Let us make this precise at the level of the quantum volumes $\mathsf{V}_{g,n}^{(b)}$ and the partition functions $Z_{g,n}^{(b)}$. In the limit $b \to 0$, one has to scale the Liouville momenta like

$$P = \frac{\ell}{4\pi b}, \tag{2.22}$$

where $\ell$ are the geodesic lengths on hyperbolic surfaces. This relation is further explained in section 4.2. We also scale the boundary temperatures as follows,

$$\beta = \frac{1}{b^2}\beta^{\mathrm{JT}}, \tag{2.23}$$

and hold $\beta^{\mathrm{JT}}$ fixed in the limit $b \to 0$. From the intersection point of view (2.20), it is obvious that the quantum volumes reduce to the ordinary Weil-Petersson volumes by using eq. (A.6) and the fact that the Todd class becomes subleading in this limit. We have

$$\mathsf{V}_{g,n}^{(b)}(P_1,\ldots,P_n) \xrightarrow{b\to 0} (8\pi^2 b^2)^{-3g+3-n} V_{g,n}(\ell_1,\ldots,\ell_n)\big(1+\mathcal{O}(b^2)\big), \tag{2.24}$$

where $V_{g,n}$ denote the Weil-Petersson volumes. In the presence of asymptotic boundaries, we have[6]

$$Z_{g,n}^{(b)}(\beta_1,\ldots,\beta_n) \xrightarrow{b\to 0} (8\pi^2 b^2)^{\frac{3}{2}(2-2g-n)} Z_{g,n}^{\mathrm{JT}}(\beta_1^{\mathrm{JT}},\ldots,\beta_n^{\mathrm{JT}}). \tag{2.25}$$

The prefactor is raised to the Euler characteristic and hence can be absorbed into the definition of $S_0$ in (2.1).

One might also wonder whether the Virasoro minimal string is related to the $(2,p)$ minimal string which also admits a double-scaled dual matrix integral description [66–68]. Moreover, there are hints that the $(2,p)$ minimal model could be obtained from timelike Liouville theory on the worldsheet by a certain gauging [69,70]. It has also been argued that the large $p$ limit of the minimal string reduces to the JT gravity, albeit in the regime where vertex operators correspond to conical defect insertions instead of geodesic boundaries [37–39,52,54]. However, let us emphasize that the $(2,p)$ minimal string and the Virasoro minimal string correspond to two completely different deformations of JT gravity and do not seem to have a direct relation. In particular the density of states of the dual matrix integrals are genuinely different.

---

[6]Here we are using standard conventions in JT gravity. In the language of [36], we set $\alpha = 1$ and $\gamma = \frac{1}{2}$.

# Part II
# Dual descriptions

## 3 A worldsheet perspective

In this section we elucidate in more detail the worldsheet description of the Virasoro minimal string. Throughout we emphasize the exact formulation of the worldsheet CFTs in terms of their operator spectrum and OPE data.

### 3.1 Description of the worldsheet CFT

**Spacelike Liouville CFT.** Spacelike Liouville CFT is a non-perturbative solution to the CFT crossing equations that exists for all complex values of the central charge $c$ away from the half-line $(-\infty, 1]$. It defines a unitary CFT only if the central charge is real and satisfies $c > 1$. Its spectrum consists of a continuum of scalar Virasoro primary operators $V_P$ with conformal weights lying above the threshold $\frac{Q^2}{4} = \frac{c-1}{24}$ as parameterized in (2.4). It is a non-compact solution to the bootstrap equations, meaning that the identity operator is not a normalizable operator in the spectrum of the theory.[7] There is significant evidence that Liouville CFT is the unique unitary CFT with $c > 1$ whose spectrum consists of only scalar Virasoro primaries (and indeed with primaries of bounded spins) [71–73].

The structure constants of Liouville CFT were famously bootstrapped by [74–77], and are given by the well-known DOZZ formula. In this work we find it convenient to adopt operator normalization conventions such that the DOZZ formula is equivalent to the universal formula $C_b$ that governs the asymptotics of CFT structure constants [73], namely[8]

$$\langle V_{P_1}(0)V_{P_2}(1)V_{P_3}(\infty)\rangle = C_b(P_1, P_2, P_3) \equiv \frac{\Gamma_b(2Q)\Gamma_b(\frac{Q}{2} \pm iP_1 \pm iP_2 \pm iP_3)}{\sqrt{2}\Gamma_b(Q)^3 \prod_{k=1}^3 \Gamma_b(Q \pm 2iP_k)}. \tag{3.1}$$

Here $\Gamma_b$ denotes the meromorphic double gamma function (see appendix C for a compendium of properties and representations of $\Gamma_b$) and the $\pm$ notation indicates a product over all eight possible sign choices. As an example $\Gamma_b(\frac{Q}{2} \pm iP_1 \pm iP_2 \pm iP_3)$ is a product over eight different factors. This in particular has the feature that it is invariant under reflections $P_j \to -P_j$ of the Liouville momenta. Although it is not a normalizable operator in the spectrum of Liouville theory, the identity operator is obtained by analytic continuation $P \to \frac{iQ}{2} \equiv \mathbb{1}$. The two-point function inherited from (3.1) is then given by

$$\langle V_{P_1}(0)V_{P_2}(1)\rangle = C_b(P_1, P_2, \mathbb{1}) = \frac{1}{\rho_0^{(b)}(P_1)}(\delta(P_1 - P_2) + \delta(P_1 + P_2)). \tag{3.2}$$

Here $\rho_0^{(b)}$ is given by the universal formula

$$\rho_0^{(b)}(P) = 4\sqrt{2}\sinh(2\pi bP)\sinh(2\pi b^{-1}P). \tag{3.3}$$

---

[7]The "spectrum" of Liouville CFT is a somewhat ambiguous notion; although sub-threshold operators are not (delta-function) normalizable in Liouville theory, we will see that one can often analytically continue observables in the theory to arbitrary values of the external Liouville momenta, corresponding for example to sub-threshold values of the conformal weights. However the fact that sub-threshold operators are non-normalizable means that they do not appear as internal states in the conformal block decomposition of generic observables, and for this reason we reserve the term "spectrum" for the normalizable, above-threshold operators.

[8]This function has been referred to as $C_0$ in the recent CFT literature. Here we find it convenient to make the dependence on the central charge explicit. Also we find it appropriate to reserve the 0 subscript for $\rho_0^{(b)}$, which plays the role of the leading density of eigenvalues in the matrix model, whereas in the present application $C_b$ is an exact CFT three-point function.

Both the two-point function and the three-point function of Liouville CFT are universal quantities in two-dimensional conformal field theory. The reason for this is that they are crossing kernels for conformal blocks involving the identity operator. We have already seen in section 2.3 that $\rho_0^{(b)}$ is the modular crossing kernel for the torus vacuum character, which is asymptotic to Cardy's formula for the universal density of high-energy states in a unitary compact 2d CFT. Similarly, $C_b$ — which describes the asymptotic structure constants of high-energy states in a unitary compact 2d CFT — is the crossing kernel for the sphere four-point conformal block describing the exchange of the identity Virasoro Verma module:

$$\text{(diagram)} = \int_0^\infty dP\, \rho_0^{(b)}(P) C_b(P_1, P_2, P)\, \text{(diagram)}\,. \tag{3.4}$$

The diagrams on the left- and right-hand sides of the above equation are respectively meant to denote the $t$- and $s$-channel Virasoro conformal blocks for the sphere four-point function of pairwise identical operators with conformal weights $h_{P_1}$ and $h_{P_2}$.

Together, this data is sufficient to compute any correlation function of local operators on any closed Riemann surface. This is achieved by the conformal block decomposition as follows:

$$\langle V_{P_1} \cdots V_{P_n} \rangle_g = \int_{\mathbb{R}_{\geq 0}} \left( \prod_a dP_a\, \rho_0^{(b)}(P_a) \right) \left( \prod_{(j,k,l)} C_b(P_j, P_k, P_l) \right) |\mathcal{F}_{g,n}^{(b)}(\mathbf{P}^{\text{ext}}; \mathbf{P}|\mathbf{m})|^2\,. \tag{3.5}$$

Here $\mathcal{F}_{g,n}^{(b)}$ are the genus-$g$ $n$-point Virasoro conformal blocks with central charge $c = 1 + 6Q^2$, $Q = b + b^{-1}$; $\mathbf{P}^{\text{ext}} = (P_1, \ldots, P_n)$ denote the external Liouville momenta, and $\mathbf{P}$ and $\mathbf{m}$ collectively denote the $3g - 3 + n$ internal Liouville momenta $P_a$ and the worldsheet moduli respectively. Left implicit in the definition of the conformal block is the choice of a channel $\mathcal{C}$ of the conformal block decomposition, which is specified by a decomposition of the worldsheet Riemann surface into $2g - 2 + n$ pairs of pants sewn along $3g - 3 + n$ cuffs, together with a choice of dual graph. The conformal block decomposition of the resulting correlator includes a factor of $\rho_0^{(b)}$ for each internal weight corresponding to the complete set of states inserted at each cuff, and a factor of $C_b$ for each pair of pants corresponding to the CFT structure constants. The resulting correlator is independent of the choice of channel in the conformal block decomposition because Liouville CFT solves the crossing equations.

A priori, for fixed worldsheet moduli, the correlation function (3.5) is a function defined for real external Liouville momenta $\mathbf{P}^{\text{ext}}$ in the spectrum of the theory. However, the structure constants $C_b$ are meromorphic functions of the Liouville momenta and we can readily consider the analytic continuation of (3.5) to complex $\mathbf{P}^{\text{ext}}$. But there may be subtleties in this analytic continuation. Even restricting to real values of the conformal weights, if the external operators have weights sufficiently below the threshold $\frac{c-1}{24}$, then poles of the structure constants cross the contour of integration and the contour must be deformed such that the conformal block decomposition picks up additional discrete contributions associated with the residues of these poles. This can happen for example whenever there is a pair of external momenta $P_j, P_k$ such that $|\text{Im}(P_j \pm P_k)| > \frac{Q}{2}$.

**Timelike Liouville CFT.** Timelike Liouville CFT is a solution to the CFT crossing equations for all values of the central charge on the half-line $\hat{c} \leq 1$. Although less well-known than (and with some peculiar features compared to) spacelike Liouville theory, it furnishes an equally good solution to the CFT bootstrap that has been developed from various points of view over the years [43, 44, 56, 57, 78]. It is essential that timelike Liouville CFT is *not* given by the

analytic continuation of spacelike Liouville theory to $c \leq 1$, although as we will see the CFT data of the two theories are related.

Similarly to spacelike Liouville theory, the spectrum of timelike Liouville theory consists of a continuum of scalar Virasoro primaries $\widehat{V}_{\widehat{P}}$ with conformal weights $\hat{h}_{\widehat{P}} \geq \frac{\hat{c}-1}{24} = -\frac{\widehat{Q}^2}{4}$ parameterized as in (2.4).[9] Unlike spacelike Liouville theory, timelike Liouville theory with $\hat{c} < 1$ never defines a unitary CFT in the sense that the spectrum contains primaries with negative conformal weights that violate the unitarity bound. Nevertheless, we will see that the structure constants of the theory are real in the cases of interest.

We adopt conventions such that the structure constants in timelike Liouville CFT are given by the inverse of an analytic continuation of the spacelike structure constants (3.1), in particular [43, 44, 56, 78, 79] .

$$
\begin{aligned}
\langle \widehat{V}_{\widehat{P}_1}(0)\widehat{V}_{\widehat{P}_2}(1)\widehat{V}_{\widehat{P}_3}(\infty) \rangle &= \widehat{C}_{\hat{b}}(\widehat{P}_1, \widehat{P}_2, \widehat{P}_3) \\
&\equiv \frac{1}{C_{\hat{b}}(i\widehat{P}_1, i\widehat{P}_2, i\widehat{P}_3)} \\
&= \frac{\sqrt{2}\Gamma_b(\hat{b}+\hat{b}^{-1})^3}{\Gamma_b(2\hat{b}+2\hat{b}^{-1})} \frac{\prod_{k=1}^{3}\Gamma_b(\hat{b}+\hat{b}^{-1}\pm 2\widehat{P}_k)}{\Gamma_b(\frac{\hat{b}+\hat{b}^{-1}}{2}\pm \widehat{P}_1 \pm \widehat{P}_2 \pm \widehat{P}_3)} \,.
\end{aligned}
\tag{3.6}
$$

With a suitable contour of integration of the internal Liouville momenta in the conformal block decomposition that we will discuss shortly, correlation functions in timelike Liouville CFT with these structure constants have been shown to solve the CFT crossing equations numerically [78, 80], see also [40]. We note in passing that although the spectrum of timelike Liouville contains a weight zero operator (with $\widehat{P} = \frac{\widehat{Q}}{2}$), it is *not* the degenerate representation corresponding to the identity operator; indeed the two-point function is not obtained by analytic continuation of (3.6) to $\widehat{P}_3 = \frac{\widehat{Q}}{2}$. The latter is instead given by

$$
\langle \widehat{V}_{\widehat{P}_1}(0)\widehat{V}_{\widehat{P}_2}(1) \rangle = \frac{2\rho_0^{(\hat{b})}(i\widehat{P})}{(i\widehat{P})^2}(\delta(\widehat{P}_1 - \widehat{P}_2) + \delta(\widehat{P}_1 + \widehat{P}_2)) \,.
\tag{3.7}
$$

Correlation functions in timelike Liouville CFT are then computed by the following conformal block decomposition

$$
\langle \widehat{V}_{\widehat{P}_1} \cdots \widehat{V}_{\widehat{P}_n} \rangle_g = \int_{\mathcal{C}} \prod_a \frac{d\widehat{P}_a (i\widehat{P}_a)^2}{2\rho_0^{(\hat{b})}(i\widehat{P}_a)} \left( \prod_{(j,k,l)} \frac{1}{C_{\hat{b}}(i\widehat{P}_j, i\widehat{P}_k, i\widehat{P}_l)} \right) |\mathcal{F}_{g,n}^{(i\hat{b})}(\widehat{\mathbf{P}}^{\text{ext}}; \widehat{\mathbf{P}}|\mathbf{m})|^2 \,,
\tag{3.8}
$$

where $\mathcal{C}$ denotes the contour $\mathbb{R} + i\varepsilon$, $\varepsilon > 0$ (see figure 5). It warrants further emphasis that the contour of integration over the internal Liouville momenta $\widehat{\mathbf{P}}$ in the conformal block decomposition of the timelike Liouville correlation function is shifted by an amount $\varepsilon$ above the real axis. Such a shift is required to avoid the infinitely many poles of the timelike Liouville structure constants on the real $\widehat{P}$ axis at

$$
\text{poles of } \widehat{C}_{\hat{b}}: \qquad \widehat{P}_j = \pm\frac{1}{2}\left((m+1)\hat{b} + (n+1)\hat{b}^{-1}\right), \quad m, n \in \mathbb{Z}_{\geq 0} \,.
\tag{3.9}
$$

These are the only singularities of $\widehat{C}_{\hat{b}}$ in the complex $\widehat{P}_i$ plane. Similarly, the $\hat{c} \leq 1$ Virasoro conformal blocks have poles on the real $\widehat{P}_i$ axis corresponding to degenerate representations of the Virasoro algebra

$$
\text{poles of } \mathcal{F}: \qquad \widehat{P}_j = \pm\frac{1}{2}\left((r+1)\hat{b} - (s+1)\hat{b}^{-1}\right), \quad r, s \in \mathbb{Z}_{\geq 0} \,.
\tag{3.10}
$$

---

[9]Sometimes states with purely imaginary $\widehat{P}$ are described as the spectrum of timelike Liouville theory, since they turn out to be natural from the point of view of the Lagrangian formulation of the theory. Here we will reserve that terminology for operators that appear in the conformal block decomposition of correlation functions.

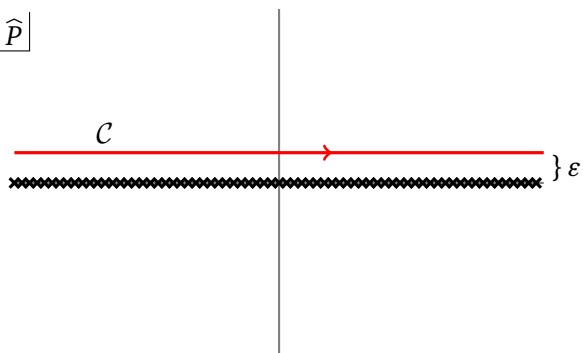

Figure 5: Contour of integration $\mathcal{C}$ over the intermediate states in the Virasoro conformal block decomposition of the genus $g$ $n$-point function (3.8) in Liouville CFT at $\hat{c} \leq 1$. Poles in the $\widehat{P}$-integrand, coming from the three-point coefficient (3.9) as well as the Virasoro conformal blocks (3.10), are marked with crosses. The contour $\mathcal{C}$ runs parallel to the real axis and shifted vertically by a small $\varepsilon > 0$ amount in the imaginary direction in order to avoid the poles. Due to the reflection symmetry of the timelike Liouville structure constant (3.6), the contour $\mathcal{C}$ could also be shifted vertically by a small $\varepsilon < 0$.

Together with the poles of the measure, the integrand has then poles for

$$\widehat{P}_j = \frac{m}{2}\hat{b} + \frac{n}{2}\hat{b}^{-1}, \quad (m,n) \in \mathbb{Z}^2 \setminus \{(0,0)\}, \tag{3.11}$$

which for $\hat{b}^2 \notin \mathbb{Q}$ is a dense set on the real line.

Since the location of the poles in the internal Liouville momenta are independent of the external Liouville momenta, analytic continuation of the timelike Liouville correlators to complex values of the external momenta $\widehat{\mathbf{P}}^{\text{ext}}$ is straightforward, and does not require the further contour deformations that are sometimes needed for analytic continuation of the spacelike Liouville correlators. Indeed, in the Virasoro minimal string we will mostly be interested in the case that the external operators have imaginary timelike Liouville momentum.

The need to shift the OPE contour as described above is perhaps an unfamiliar aspect of timelike Liouville theory. It renders the notion of the spectrum of timelike Liouville somewhat ambiguous, since we may freely deform the OPE contour provided that the poles (3.9), (3.10) on the real axis are avoided. One may wonder about the possibility of different OPE contours. For example, although states with imaginary Liouville momentum are from some points of view natural in timelike Liouville theory, it is clear that with a vertical contour the conformal block decomposition would badly diverge, since with that prescription the OPE would contain internal states with arbitrarily negative conformal dimension. With this prescription where the OPE contour runs parallel to the real axis, the correlation functions of timelike Liouville CFT have been shown to solve the bootstrap equations numerically [40,78]. Since it satisfies these basic CFT consistency conditions, our view is that despite some subtleties (including non-unitarity of the spectrum) timelike Liouville theory is non-perturbatively well-defined as a CFT in the same sense as spacelike Liouville theory.

**The Virasoro minimal string background.** Equipped with our knowledge of the OPE data of spacelike and timelike Liouville theories that together with the $\mathfrak{bc}$-ghost system defines the worldsheet CFT of the Virasoro minimal string, we can now proceed to compute string worldsheet diagrams as usual in string theory. On-shell vertex operators $\mathcal{V}_P$ (2.6) are labelled by a single Liouville momentum $P$ and are defined by combining primaries in spacelike and timelike Liouville CFT with the $\mathfrak{bc}$-ghosts as in (2.6). In string perturbation theory, the observables are

string worldsheet diagrams $V_{g,n}^{(b)}(P_1, \ldots, P_n)$ ("quantum volumes"), which we define by integrating correlation functions of the worldsheet CFT over the moduli space of Riemann surfaces as outlined in (2.7).

Let us pause to briefly comment on the convergence properties of the moduli integral (2.7) that defines the string worldsheet diagrams that we compute in this paper. In string perturbation theory one often has to worry about divergences in the integrals over worldsheet moduli space that define string diagrams due to intermediate states going on shell. These divergences are associated with particular degenerations in moduli space — for instance, the genus-$g$ worldsheet may split into two components $\Sigma_{g,n} \to \Sigma_{g_1,n_1+1} \cup \Sigma_{g_2,n_2+1}$ with $g = g_1 + g_2$ and $n = n_1 + n_2$, or in the case of non-separating degenerations, in which a handle pinches and the genus of the worldsheet drops by one but remains connected. The behaviour of the worldsheet integrand near such degenerations is sensitive to the exchange of the lightest operators in the spectrum of the worldsheet CFT. In the Virasoro minimal string theory, the absence of the identity operator (in other words, the non-compact nature of the worldsheet CFT) and the scaling dimensions of the lightest operators in spacelike and timelike Liouville CFT ensure that the resulting moduli integral is in fact absolutely convergent in degenerating limits. We see this explicitly in the case of the torus one-point and sphere four-point diagrams discussed in sections 7.1 and 7.2.

Let us make this more concrete with an example. Consider for instance the moduli integrand in the sphere four-point diagram $V_{0,4}^{(b)}(P_1, \ldots, P_4)$, which is computed by integrating the sphere four-point functions of spacelike and timelike Liouville CFT over the complex cross-ratio plane:[10]

$$V_{0,4}^{(b)}(P_1, \ldots, P_4) = \int_{\mathbb{C}} \mathrm{d}^2 z \, \langle V_{P_1} \cdots V_{P_4} \rangle \langle \widehat{V}_{iP_1} \cdots \widehat{V}_{iP_4} \rangle \,. \tag{3.12}$$

We will be interested in the behaviour of the worldsheet integrand in the limit in which two of the vertex operators, say those corresponding to the momenta $P_1$ and $P_2$, coincide. In this degeneration limit the sphere four-point Virasoro blocks can be approximated by the leading term in the small cross-ratio expansion

$$\mathcal{F}_{0,4}^{(b)}(\mathbf{P}^{\mathrm{ext}}; P|z) \approx z^{P^2 - P_1^2 - P_2^2 - \frac{Q^2}{4}} \,. \tag{3.13}$$

In this limit the OPE integrals appearing in the spacelike and timelike Liouville four-point functions will be dominated by the $P, \widehat{P} \approx 0$ regions,[11] for which we have

$$\rho_0^{(b)}(P) \approx 16\sqrt{2}\pi^2 P^2 \,. \tag{3.14}$$

Hence we can approximate the sphere four-point functions of spacelike and timelike Liouville CFT as follows in the degeneration limit

$$\langle V_{P_1} \cdots V_{P_4} \rangle \overset{z \to 0}{\approx} \frac{2\pi^{\frac{5}{2}} C_b(P_1, P_2, 0) C_b(P_3, P_4, 0) |z|^{-2P_1^2 - 2P_2^2 - \frac{Q^2}{2}}}{(-\log|z|)^{\frac{3}{2}}} \,,$$

$$\langle \widehat{V}_{iP_1} \cdots \widehat{V}_{iP_4} \rangle \overset{z \to 0}{\approx} \frac{|z|^{2P_1^2 + 2P_2^2 + \frac{\widehat{Q}^2}{2}}}{64\pi^{\frac{3}{2}} (-\log|z|)^{\frac{1}{2}} C_b(P_1, P_2, 0) C_b(P_3, P_4, 0)} \,. \tag{3.15a}$$

In particular the product of four-point functions that appears in the moduli integrand has the following behaviour in the degeneration limit

$$\langle V_{P_1} \cdots V_{P_4} \rangle \langle \widehat{V}_{iP_1} \cdots \widehat{V}_{iP_4} \rangle \overset{z \to 0}{\approx} \frac{\pi |z|^{-2}}{32(-\log|z|)^2} \,, \tag{3.16}$$

---

[10]In what follows we will typically omit the explicit dependence on the worldsheet moduli of the worldsheet CFT correlators for brevity of notation. For example below we have $\langle V_{P_1} \cdots V_{P_4} \rangle = \langle V_{P_1}(0) V_{P_2}(z) V_{P_3}(1) V_{P_4}(\infty) \rangle$.

[11]Here we are assuming that the external Liouville momenta are such that the contour in the conformal block decomposition does not need to be deformed. This is always the case for real Liouville momenta.

and thus the moduli integral (3.12) receives convergent contributions from the degeneration limit locally of the form

$$\int_{\mathbb{C}} \frac{\mathrm{d}^2 z}{|z|^2 (-\log |z|)^2} \,. \tag{3.17}$$

Similar considerations apply to all other degeneration limits of the sphere four-point diagram (which can be studied exactly analogously by working in different OPE channels), and to degeneration limits of more complicated observables. It is interesting to compare eq. (3.16) with the leading behaviour of the Weil-Petersson volume form, which appears in JT gravity. Using the explicit form $\omega_{\mathrm{WP}} = \mathrm{d}\ell \wedge \mathrm{d}\theta$ of the Weil-Petersson form in Fenchel-Nielsen coordinates [81] and the leading relation

$$\ell \sim \frac{2\pi^2}{-\log |z|} \,, \qquad \frac{2\pi\theta}{\ell} \sim \arg(z) \,, \tag{3.18}$$

between $z$ and the Fenchel-Nielsen coordinates, gives the leading behaviour [82]

$$\omega_{\mathrm{WP}} \sim \frac{8\pi^3 i \, \mathrm{d}z \wedge \mathrm{d}\bar{z}}{|z|^2 (-\log |z|)^3} \,, \tag{3.19}$$

which is slightly faster decaying than (3.16).

**A trivial worldsheet diagram.** As a trivial example, let us consider the three-punctured sphere. In this case there are no moduli to integrate over, and the three-point diagram is simply given by the product of the corresponding structure constants in spacelike and timelike Liouville theory given by (3.1) and (3.6) respectively. On the solution to the mass-shell condition (2.5) the sphere three-point diagram is then simply given by

$$
\begin{aligned}
\mathsf{V}_{0,3}^{(b)}(P_1, P_2, P_3) &\equiv C_{\mathrm{S}^2} \langle \mathcal{V}_{P_1}(0) \mathcal{V}_{P_2}(1) \mathcal{V}_{P_3}(\infty) \rangle \\
&= C_{\mathrm{S}^2} \mathsf{N}(P_1) \mathsf{N}(P_2) \mathsf{N}(P_3) C_b(P_1, P_2, P_3) \widehat{C}_b(iP_1, iP_2, iP_3) \\
&= C_{\mathrm{S}^2} \mathsf{N}(P_1) \mathsf{N}(P_2) \mathsf{N}(P_3) \frac{C_b(P_1, P_2, P_3)}{C_b(P_1, P_2, P_3)} \\
&= C_{\mathrm{S}^2} \mathsf{N}(P_1) \mathsf{N}(P_2) \mathsf{N}(P_3) \,,
\end{aligned}
\tag{3.20}
$$

where we have used the relation between the structure constants of timelike and spacelike Liouville given in (3.6) together with reflection invariance of $C_b$. Here, $C_{\mathrm{S}^2}$ reflects the arbitrary normalization of the string path integral.

We fix the arbitrary normalizations $\mathsf{N}(P)$ of the vertex operators by requiring that

$$\mathsf{V}_{0,3}^{(b)}(P_1, P_2, P_3) \overset{!}{=} 1 \,, \tag{3.21}$$

which implies that $\mathsf{N}(P) \equiv \mathsf{N}$ is independent of $P$ and

$$C_{\mathrm{S}^2} = \mathsf{N}^{-3} \,. \tag{3.22}$$

## 3.2 Worldsheet boundary conditions

In order to discuss configurations with asymptotic boundaries we need to supplement the worldsheet CFT with conformal boundary conditions. Here we review the conformal boundary conditions of spacelike and timelike Liouville CFT, and describe their role in the worldsheet description of configurations with asymptotic boundaries in Virasoro minimal string theory. Throughout we emphasize the definition of the conformal boundary conditions in terms of abstract boundary conformal field theory (BCFT) data rather than in terms of specific boundary conditions for the Liouville fields in the Lagrangian descriptions of the theories.

**Conformal boundary conditions for spacelike Liouville**

Spacelike Liouville CFT admits two main types of conformal boundary conditions, whose properties we summarize in turn.

**ZZ boundary conditions.** The first are the ZZ boundary conditions [83], which are labelled by a degenerate representation of the Virasoro algebra. In defining conformal boundary conditions, it is convenient to map the upper half-plane to the unit disk by a conformal transformation so that the boundary condition defines a state in the Hilbert space of the CFT on the circle by the usual radial quantization. The ZZ boundary states can be represented in terms of the Ishibashi states $|V_P\rangle\rangle$ associated with the primaries in the spectrum as follows[12]

$$|\text{ZZ}_{(m,n)}^{(b)}\rangle = \int_0^\infty dP\, \rho_0^{(b)}(P)\Psi_{(m,n)}^{(b)}(P)|V_P\rangle\rangle\,. \tag{3.23}$$

The quantity $\Psi_{(m,n)}^{(b)}(P)$, which we will specify shortly, is the disk one-point function of the primary $V_P$ in the presence of the $(m,n)$ ZZ boundary condition.

Consider the annulus formed by cutting a circle of radius $e^{-\pi t}$ out of the unit disk, with Ishibashi states $|V_{P_1}\rangle\rangle$ and $|V_{P_2}\rangle\rangle$ on the inner and outer boundary circles respectively. This configuration corresponds by the usual exponential map to the following partition function on a cylinder with unit radius and length $\pi t$:

$$\langle\langle V_{P_1}|e^{-\pi t(L_0+\bar{L}_0-\frac{c}{12})}|V_{P_2}\rangle\rangle = \frac{\delta(P_1-P_2)+\delta(P_1+P_2)}{\rho_0^{(b)}(P_1)}\chi_{P_1}^{(b)}(it)\,, \tag{3.24}$$

where

$$\chi_P^{(b)}(\tau) = \frac{q^{P^2}}{\eta(\tau)}\,, \quad q = e^{2\pi i\tau}\,, \tag{3.25}$$

is the non-degenerate Virasoro character associated with a primary of conformal weight $h_P$. The ZZ boundary states are defined by the property that the cylinder partition function with the $(m,n)$ and $(1,1)$ boundary conditions assigned to the two ends is given by the corresponding Virasoro character in the open string channel [83]

$$\begin{aligned}
\langle\text{ZZ}_{(m,n)}^{(b)}|e^{-\pi t(L_0+\bar{L}_0-\frac{c}{12})}|\text{ZZ}_{(1,1)}^{(b)}\rangle &= \int_0^\infty dP\, \rho_0^{(b)}(P)\Psi_{(m,n)}^{(b)}(P)\Psi_{(1,1)}^{(b)}(P)\chi_P^{(b)}(it) \\
&\overset{!}{=} \chi_{(m,n)}^{(b)}\left(\tfrac{i}{t}\right) \\
&= \text{Tr}_{\mathcal{H}_{(m,n),(1,1)}}e^{-\frac{2\pi}{t}(L_0-\frac{c}{24})}\,,
\end{aligned} \tag{3.26}$$

with

$$\chi_{(m,n)}^{(b)}(\tau) = \frac{q^{-\frac{1}{4}(mb+nb^{-1})^2}-q^{-\frac{1}{4}(mb-nb^{-1})^2}}{\eta(\tau)}\,, \quad q = e^{2\pi i\tau}\,, \tag{3.27}$$

the torus character of the $(m,n)$ degenerate representation of the Virasoro algebra. This fixes the bulk one-point functions to be

$$\Psi_{(m,n)}^{(b)}(P) = \frac{4\sqrt{2}\sinh(2\pi mbP)\sinh(2\pi nb^{-1}P)}{\rho_0^{(b)}(P)}\,. \tag{3.28}$$

---

[12]The convention of including $\rho_0^{(b)}(P)$ in the measure of the integral over $P$ is natural in our normalization of Liouville theory. This will also lead to analytic expressions for the wave-functions, contrary to the perhaps more familiar conventions from the literature.

In particular we have $\Psi^{(b)}_{(1,1)}(P) = 1$, for which the cylinder partition function is the Virasoro identity character in the open-string channel. In the last line of (3.26) we have reminded the reader that the cylinder partition function admits an interpretation in terms of a trace over the Hilbert space of the CFT on the strip with thermal circle of size $\frac{2\pi}{t}$. The more general cylinder partition function with mixed ZZ boundary conditions is given by the following sum over degenerate Virasoro characters in the open string channel [83]

$$\langle ZZ^{(b)}_{(m,n)}| e^{-\pi t(L_0 + \bar{L}_0 - \frac{c}{12})} |ZZ^{(b)}_{(m',n')}\rangle = \sum_{r\overset{2}{=}|m-m'|+1}^{m+m'-1} \sum_{s\overset{2}{=}|n-n'|+1}^{n+n'-1} \chi^{(b)}_{(r,s)}(\tfrac{i}{t}), \tag{3.29}$$

where the notation $\overset{2}{=}$ is meant to indicate that the variable increases in steps of 2.

**FZZT boundary conditions.** Spacelike Liouville theory also admits a distinct one-parameter family of conformal boundaries known as the FZZT boundary conditions [84, 85]. It is described by the following boundary state

$$|FZZT^{(b)}(s)\rangle = \int_0^\infty dP\, \rho_0^{(b)}(P)\Psi^{(b)}(s; P)|V_P\rangle\rangle. \tag{3.30}$$

The FZZT parameter $s$ takes real values. Indeed we will see that it labels a state in the spectrum of Liouville theory. The FZZT boundary state is defined such that the Hilbert space of Liouville CFT on the strip with FZZT boundary conditions on one end and $(1, 1)$ ZZ boundary conditions on the other is spanned by a single primary state labelled by the Liouville momentum $s$. Indeed, the mixed cylinder partition function is given by a single non-degenerate Virasoro character in the open-string channel

$$\langle ZZ^{(b)}_{(1,1)}|e^{-\pi t(L_0 + \bar{L}_0 - \frac{c}{12})}|FZZT^{(b)}(s)\rangle = \int_0^\infty dP\, \rho_0^{(b)}(P)\Psi^{(b)}_{(1,1)}(P)\Psi^{(b)}(s; P)\chi^{(b)}_P(it)$$

$$\overset{!}{=} \chi^{(b)}_s(\tfrac{i}{t}). \tag{3.31}$$

Hence the FZZT bulk one-point function $\Psi^{(b)}(s; P)$ is given by

$$\Psi^{(b)}(s; P) = \frac{\mathbb{S}_{sP}[\mathbb{1}]}{\rho_0^{(b)}(P)} = \frac{2\sqrt{2}\cos(4\pi sP)}{\rho_0^{(b)}(P)}. \tag{3.32}$$

Here $\mathbb{S}[\mathbb{1}]$ is the crossing kernel for Virasoro characters on the torus.

In what follows the partition function of Liouville CFT on the cylinder with FZZT boundary conditions at the two ends will play an important role. It is given by

$$\langle FZZT^{(b)}(s_1)|e^{-\pi t(L_0 + \bar{L}_0 - \frac{c}{12})}|FZZT^{(b)}(s_2)\rangle = \frac{1}{2}\int_\Gamma dP\, \rho_0^{(b)}(P)\Psi^{(b)}(s_1; P)\Psi^{(b)}(s_2; P)\chi^{(b)}_P(it)$$

$$= \frac{1}{\sqrt{2}}\int_\Gamma dP\, \frac{\cos(4\pi s_1 P)\cos(4\pi s_2 P)}{\sinh(2\pi b P)\sinh(2\pi b^{-1} P)}\chi^{(b)}_P(it). \tag{3.33}$$

Here we have promoted the integral over the positive $P$ axis to a horizontal contour $\Gamma$ in the complex $P$ plane that avoids the pole of the integrand at the origin. Since the residue at $P = 0$ vanishes, it does not matter whether the contour passes above or below 0. The open string spectrum consists of a continuum of states with weights above the $\frac{c-1}{24}$ threshold.

**Conformal boundary conditions for timelike Liouville**

When we add boundaries to the worldsheet in Virasoro minimal string theory we will pair particular conformal boundaries for the spacelike Liouville sector with those of the timelike Liouville sector. Conformal boundary conditions for timelike Liouville CFT have been relatively unexplored compared to their spacelike counterparts (see however [86]). Here we will introduce a new family of ZZ-like boundary conditions for timelike Liouville CFT that will play a distinguished role in the Virasoro minimal string. Before moving on, let us emphasize that conformal boundaries of non-unitary and non-compact CFTs are relatively weakly constrained[13] and thus it is a priori not particularly clear what wavefunctions should be allowed. Nevertheless, we find the following boundary condition very natural.

**"Half-ZZ" boundary conditions.** Consider the following boundary states for timelike Liouville CFT

$$|\widehat{ZZ}^{(i\hat{b})}_{(m,\pm)}\rangle = \int_{\mathcal{C}} d\widehat{P} \, \frac{(i\widehat{P})^2}{2\rho_0^{(\hat{b})}(i\widehat{P})} \widehat{\Psi}^{(i\hat{b})}_{(m,\pm)}(\widehat{P})|\widehat{V_{\widehat{P}}}\rangle\rangle \,, \tag{3.34}$$

where $|\widehat{V_{\widehat{P}}}\rangle\rangle$ is the Ishibashi state associated to the primary $\widehat{V}_{\widehat{P}}$ in the spectrum of timelike Liouville CFT, normalized such that

$$\langle\langle\widehat{V}_{\widehat{P}_1}|e^{-\pi t(L_0+\bar{L}_0-\frac{\hat{c}}{12})}|\widehat{V}_{\widehat{P}_2}\rangle\rangle = \frac{2\rho_0^{(\hat{b})}(i\widehat{P})}{(i\widehat{P})^2}\big(\delta(\widehat{P}_1-\widehat{P}_2)+\delta(\widehat{P}_1+\widehat{P}_2)\big)\chi^{(i\hat{b})}_{\widehat{P}}(it)\,. \tag{3.35}$$

In (3.34) we have again included the measure that descends from the two-point function of timelike Liouville CFT (see e.g. (3.8)), which is natural in our normalization. The contour is also the same as appears in section 3.1 and that avoids all the poles on the real line, $\mathcal{C} = \mathbb{R}+i\varepsilon$. The corresponding conformal boundary conditions come in two infinite families, labelled by a positive integer $m \in \mathbb{Z}_{\geq 1}$ and a sign. We declare that the bulk one-point functions on the disk $\widehat{\Psi}^{(i\hat{b})}_{(m,\pm)}$ are given by[14]

$$\widehat{\Psi}^{(i\hat{b})}_{(m,\pm)}(\widehat{P}) = \frac{4\sin(2\pi m\hat{b}^{\pm1}\widehat{P})}{\widehat{P}} \,. \tag{3.36}$$

In what follows we will refer to these as "half-ZZ" boundary conditions. The reason for the "half-ZZ" name is that the product of the $(m,+)$ and $(n,-)$ wavefunctions (3.36) is functionally similar (but not identical) to that of the $(m,n)$ ordinary ZZ boundary conditions (3.28) adapted to timelike Liouville CFT with $\hat{c} \leq 1$.

In order to assess these boundary states, we scrutinize the cylinder partition functions associated with them. In particular, consider the cylinder partition function with $(m,+)$ half-ZZ boundary conditions on one end and $(n,+)$ on the other. It is given by

$$Z^{(i\hat{b})}_{(m,+;n,+)}(t) = \langle\widehat{ZZ}^{(i\hat{b})}_{(m,+)}|e^{-\pi t(L_0+\bar{L}_0-\frac{\hat{c}}{12})}|\widehat{ZZ}^{(i\hat{b})}_{(n,+)}\rangle$$

$$= \int_{\mathcal{C}} d\widehat{P} \, \frac{(i\widehat{P})^2}{2\rho_0^{(\hat{b})}(i\widehat{P})} \int_{\mathcal{C}} d\widehat{P}' \, \frac{(i\widehat{P}')^2}{2\rho_0^{(\hat{b})}(i\widehat{P}')} \widehat{\Psi}^{(i\hat{b})}_{(m,+)}(\widehat{P})\widehat{\Psi}^{(i\hat{b})}_{(n,+)}(\widehat{P}')\langle\langle\widehat{V}_{\widehat{P}}|e^{-\pi t(L_0+\bar{L}_0-\frac{\hat{c}}{12})}|\widehat{V}_{\widehat{P}'}\rangle\rangle$$

---

[13]Here we mean that in non-compact and non-unitary CFT, in implementing the cylinder bootstrap the spectrum in the open-string channel is a priori not subject to the usual constraints of positivity, discreteness, and integrality. Nevertheless we will see that the cylinder partition functions involving the conformal boundary conditions that we will introduce obey these properties.

[14]Here the $\pm$ on the RHS is correlated to that on the LHS; it does not mean the product of the expressions with each sign, as was the case in (3.1).

$$= \int_{\mathcal{C}} d\widehat{P} \, \frac{(i\widehat{P})^2}{2\rho_0^{(\hat{b})}(i\widehat{P})} \widehat{\Psi}_{(m,+)}^{(i\hat{b})}(\widehat{P}) \widehat{\Psi}_{(n,+)}^{(i\hat{b})}(\widehat{P}) \chi_{\widehat{P}}^{(i\hat{b})}(it)$$

$$= \sum_{r \stackrel{2}{=} |m-n|+1}^{m+n-1} \sum_{s \stackrel{2}{=} 1}^{\infty} \chi_{(r,s)}^{(i\hat{b})}\left(\tfrac{i}{t}\right). \tag{3.37}$$

The result takes the form of an infinite sum over degenerate characters of the central charge $\hat{c}$ Virasoro algebra in the open-string channel. The structure of degenerate representations of the $\hat{c} \leq 1$ Virasoro algebra is such that this sum is actually convergent. Indeed, the cylinder partition function (3.37) is formally equivalent to that of spacelike Liouville CFT with $(m, \infty)$ and $(n, \infty)$ ordinary ZZ boundary conditions analytically continued to $\hat{c} \leq 1$.[15] Analogously, we have

$$Z_{(m,-;n,-)}^{(i\hat{b})}(t) = \sum_{r \stackrel{2}{=} 1}^{\infty} \sum_{s \stackrel{2}{=} |m-n|+1}^{m+n-1} \chi_{(r,s)}^{(i\hat{b})}\left(\tfrac{i}{t}\right). \tag{3.38}$$

A very similar calculation yields the following for the cylinder partition function in timelike Liouville theory with $(m, +)$ and $(n, -)$ half-ZZ boundary conditions

$$Z_{(m,+;n,-)}^{(i\hat{b})}(t) = \langle \widehat{ZZ}_{(m,+)}^{(i\hat{b})} | e^{-\pi t(L_0 + \bar{L}_0 - \frac{\hat{c}}{12})} | \widehat{ZZ}_{(n,-)}^{(i\hat{b})} \rangle = \sum_{r \stackrel{2}{=} -m+1}^{m-1} \sum_{s \stackrel{2}{=} -n+1}^{n-1} \chi_{\widehat{P} = \frac{1}{2}(r\hat{b} - s\hat{b}^{-1})}^{(i\hat{b})}\left(\tfrac{i}{t}\right). \tag{3.39}$$

The result involves a finite sum over certain *non-degenerate* Virasoro characters in the open-string channel (some of which involve conformal weights equal to those of particular degenerate representations of the Virasoro algebra).

Timelike Liouville CFT presumably also admits a suitable generalization of the FZZT boundary conditions [86], which are conceptually similar to those of spacelike Liouville theory that were discussed in section 3.2. In this paper we will not make use of FZZT boundary conditions for timelike Liouville CFT and so we will not discuss them any further here.

# 4 A three-dimensional perspective

In this section, we give a conceptual derivation of the proposed duality. Our arguments will heavily involve a connection to a chiral half of three-dimensional gravity on the topology $\Sigma_{g,n} \times S^1$.

## 4.1 3d gravity on $\Sigma_{g,n} \times S^1$

We consider three-dimensional quantum gravity with negative cosmological constant. Let $\Sigma_{g,n}$ be an initial-value surface of genus $g$ with $n$ punctures. Then it is known that the Hilbert space of 3d gravity on $\Sigma_{g,n}$ can be identified with $\mathcal{H}_{\text{gravity}} = \mathcal{H}_{g,n} \otimes \overline{\mathcal{H}}_{g,n}$, where $\mathcal{H}_{g,n}$ is the space of Virasoro conformal blocks with all internal conformal weights above the $\frac{c-1}{24}$ threshold [87,88]. Since these are precisely the conformal blocks that appear in Liouville theory, we will often adopt "Liouville conformal blocks" as a shorthand. The central charge of the Liouville theory is given by the Brown-Henneaux central charge $c$, which is an arbitrary parameter of the theory. As in the rest of the paper, we take $c \geq 25$. Insertions of vertex operators on $\Sigma_{g,n}$ correspond to massive particles in the three-dimensional picture (for conformal weights $h \leq \frac{c-1}{24}$) and to black holes (for conformal weight $h > \frac{c-1}{24}$).

---

[15]However for spacelike Liouville theory, the sum over degenerate characters would diverge.

In ordinary 3d gravity, we take the central charge of the two factors $\mathcal{H}_{g,n}$ to be equal, but we can also consider the case where the right-moving central charge $\bar{c}$ is different. In particular, the relation to 2d gravity will appear in a chiral version of gravity, where $\bar{c} = 0$. In this case, we can remove one factor of the Hilbert space and simply take a chiral half

$$\mathcal{H}_{g,n} = \text{space of Liouville conformal blocks.} \tag{4.1}$$

We can endow this space with an inner product to turn it into a Hilbert space. Letting $\mathcal{F}_1$ and $\mathcal{F}_2$ be two Liouville conformal blocks, we have schematically [87,88]

$$\langle \mathcal{F}_1 \,|\, \mathcal{F}_2 \rangle = \int_{\mathcal{T}_{g,n}} \overline{\mathcal{F}}_1 \, \mathcal{F}_2 \, Z_{\text{tL}} \, Z_{\text{gh}}, \tag{4.2}$$

where $Z_{\text{tL}}$ is the partition function of timelike Liouville theory of central charge $26 - c$. $Z_{\text{gh}}$ is the $\mathfrak{b}\mathfrak{c}$-ghost partition function as in string theory that provides the measure to integrate over Teichmüller space $\mathcal{T}_{g,n}$. Let us recall that Teichmüller space is the universal covering space of the moduli space of Riemann surfaces $\mathcal{M}_{g,n}$. Since the conformal blocks are not crossing symmetric it would not make sense to restrict this integral to moduli space. However, just like in string theory, the total central charge needs to equal 26 for the Weyl anomaly to cancel. In the presence of punctures $Z_{\text{tL}}$ should be thought of as a correlation function in timelike Liouville theory, where the vertex operators are chosen such that all the combined external conformal weights sum to one.

Only Liouville conformal blocks are (delta-function) normalizable with respect to this inner product. In fact, there is an explicit formula for this inner product [88]. For the four-punctured sphere, it takes the following form[16]

$$\langle \mathcal{F}_{0,4}^{(b)}(\mathbf{P}^{\text{ext}}; P) \,|\, \mathcal{F}_{0,4}^{(b)}(\mathbf{P}^{\text{ext}}; P') \rangle = \frac{\rho_0^{(b)}(P)^{-1} \, \delta(P - P')}{C_b(P_1, P_2, P) C_b(P_3, P_4, P)}, \tag{4.3}$$

where we assumed the two conformal blocks to be in the same OPE channel. We also wrote $\mathbf{P}^{\text{ext}} = (P_1, P_2, P_3, P_4)$. Here and throughout we use the notation $|\mathcal{F}_{g,n}^{(b)}(\mathbf{P}^{\text{ext}}; \mathbf{P})\rangle$ for the states in $\mathcal{H}_{g,n}$ whose wavefunction at some fixed value of the moduli $\mathbf{m}$ is given by $\mathcal{F}_{g,n}^{(b)}(\mathbf{P}^{\text{ext}}; \mathbf{P}|\mathbf{m})$. More generally, we get a factor of $C_b(P_j, P_k, P_l)^{-1}$ for every three-punctured sphere appearing in the pair-of-pants decomposition of the conformal block and a factor of $\rho_0^{(b)}(P)^{-1}$ for every cuff. This is precisely the inverse of the OPE density of spacelike Liouville theory, for which we summarized our conventions in section 3.1 and appendix C. This formula can be derived in a variety of ways [88]. It is for example fully fixed up to overall normalization by requiring that crossing transformations on conformal blocks act unitarily.

The inner product (4.2) is tantalizingly close to the integral that we want to compute for the two-dimensional theory of gravity under consideration. In fact, it tells us about the integral over Teichmüller space of the worldsheet partition/correlation function *before* integrating over the internal Liouville momenta. Let us make this a bit more precise as follows. Recall that the moduli space of Riemann surfaces is the quotient of Teichmüller space by the mapping class group. For example, in the simplest case of a once-punctured torus, this mapping class group is simply given by the group of modular transformations $\text{Map}(\Sigma_{1,1}) = \text{SL}(2, \mathbb{Z})$. There is a subgroup of the mapping class group $\text{Map}(\Sigma_{g,n})$ generated by Dehn twists around the curves used to define the pair of pants decomposition. It is an abelian group $\mathbb{Z}^{3g-3+n}$. The conformal blocks transform with a simple phase $e^{2\pi i h_P}$ under such a Dehn twist, where $P$ denotes the Liouville momentum through the curve around which we perform the Dehn twist. In particular, this phase cancels once one combines the left- and right-movers. We consider the

---

[16]This formula implicitly sets a convention for the normalization of the ghost partition function.

case of the four-punctured sphere for simplicity. Then we have the following integral identity (we suppress the ghosts in the notation)

$$\int_{\mathcal{T}_{0,4}/\mathbb{Z}} \rho_0^{(b)}(P) C_b(P_1, P_2, P) C_b(P_3, P_4, P) \left| \mathcal{F}_{0,4}^{(b)}(\mathbf{P}^{\text{ext}}; P|z) \right|^2 \left\langle \prod_{j=1}^4 \widehat{V}_{iP_j}(z_j) \right\rangle = 2P. \qquad (4.4)$$

This equation follows from eq. (4.3) as follows. Consider $P$ close to $P'$. Then we can write the integral over Teichmüller space that defines the inner product (4.2) as follows:

$$\frac{\rho_0^{(b)}(P)^{-1} \delta(P - P')}{C_b(P_1, P_2, P) C_b(P_3, P_4, P)} = \sum_{n \in \mathbb{Z}} e^{2\pi i n (h_P - h_{P'})} \int_{\mathcal{T}_{0,4}/\mathbb{Z}} \overline{\mathcal{F}_{0,4}^{(b)}(\mathbf{P}^{\text{ext}}; P|z)} \mathcal{F}_{0,4}^{(b)}(\mathbf{P}^{\text{ext}}; P'|z) \left\langle \prod_{j=1}^4 \widehat{V}_{iP_j}(z_j) \right\rangle$$

$$= \delta(h_P - h_{P'}) \int_{\mathcal{T}_{0,4}/\mathbb{Z}} \overline{\mathcal{F}_{0,4}^{(b)}(\mathbf{P}^{\text{ext}}; P|z)} \mathcal{F}_{0,4}^{(b)}(\mathbf{P}^{\text{ext}}; P'|z) \left\langle \prod_{j=1}^4 \widehat{V}_{iP_j}(z_j) \right\rangle. \qquad (4.5)$$

In the first line, we chopped up the integral over Teichmüller space. We made some arbitrary choice of fundamental domain in the integration over $\mathcal{T}_{0,4}/\mathbb{Z}$ and used that the conformal blocks transform simply under Dehn twists. We can now strip off the delta-function and compare the coefficients. Since $h_P = \frac{c-1}{24} + P^2$, we have (recall that we assume $P, P' \geq 0$):

$$\delta(h_P - h_{P'}) = \delta(P^2 - (P')^2) = \frac{1}{2P} \delta(P - P'). \qquad (4.6)$$

Thus (4.4) follows.

Coming back to the chiral half of 3d gravity, the partition function on a 3-manifold of the form $\Sigma_{g,n} \times S^1$ can be formally obtained as follows

$$Z_{\Sigma_{g,n} \times S^1} = \frac{1}{|\text{Map}(\Sigma_{g,n})|} \dim \mathcal{H}_{g,n}$$

$$= \frac{1}{|\text{Map}(\Sigma_{g,n})|} \int d^{3g-3+n} \mathbf{P} \, \text{tr} \frac{|\mathcal{F}_{g,n}^{(b)}(\mathbf{P}^{\text{ext}}; \mathbf{P})\rangle \langle \mathcal{F}_{g,n}^{(b)}(\mathbf{P}^{\text{ext}}; \mathbf{P})|}{\langle \mathcal{F}_{g,n}^{(b)}(\mathbf{P}^{\text{ext}}; \mathbf{P}) | \mathcal{F}_{g,n}^{(b)}(\mathbf{P}^{\text{ext}}; \mathbf{P}) \rangle}$$

$$= \frac{1}{|\text{Map}(\Sigma_{g,n})|} \int d^{3g-3+n} \mathbf{P} \prod_a \rho_0^{(b)}(P_a) \prod_{(j,k,l)} C_b(P_j, P_k, P_l)$$

$$\times \int_{\mathcal{T}_{g,n}} \left| \mathcal{F}_{g,n}^{(b)}(\mathbf{P}^{\text{ext}}; \mathbf{P}|\mathbf{m}) \right|^2 \left\langle \prod_{j=1}^n \widehat{V}_{iP_j}(z_j) \right\rangle_g Z_{\text{gh}}$$

$$= \frac{1}{|\text{Map}(\Sigma_{g,n})|} \int_{\mathcal{T}_{g,n}} \left\langle \prod_{j=1}^n V_{P_j}(z_j) \right\rangle_g \left\langle \prod_{j=1}^n \widehat{V}_{iP_j}(z_j) \right\rangle_g Z_{\text{gh}}$$

$$= \int_{\mathcal{M}_{g,n}} \left\langle \prod_{j=1}^n V_{P_j}(z_j) \right\rangle_g \left\langle \prod_{j=1}^n \widehat{V}_{iP_j}(z_j) \right\rangle_g Z_{\text{gh}}$$

$$= \mathsf{V}_{g,n}^{(b)}(P_1, \ldots, P_n). \qquad (4.7)$$

Here we used that the mapping class group $\text{Map}(\Sigma_{g,n})$ is gauged in gravity and that the three-dimensional mapping class group on $\Sigma_{g,n} \times S^1$ coincides with the two-dimensional one. We have by definition $\mathcal{M}_{g,n} = \mathcal{T}_{g,n}/\text{Map}(\Sigma_{g,n})$. We also used that the Hamiltonian of gravity vanishes and the partition function before dividing by the mapping class group is simply given by the (infinite) dimension of the Hilbert space. We wrote this dimension as a trace of the identity, which we in turn wrote by inserting a complete set of conformal blocks, for which we

used a braket notation to emphasize that they span the Hilbert space. By the inner product $\langle \mathcal{F}_{g,n}^{(b)} | \mathcal{F}_{g,n}^{(b)} \rangle$ in the second line of (4.7), we mean the coefficient of the delta-function appearing in (4.3). We then use the formula in terms of an integral over Teichmüller space (4.2) in the numerator and the explicit formula (4.3) in the denominator. We recognize the conformal block expansion of the spacelike Liouville correlation function in the third line of the above equation (4.7). Finally, we can gauge the mapping class group by using the crossing symmetry of the spacelike Liouville correlation function and restrict the integral to moduli space $\mathcal{M}_{g,n}$. We thus reach the conclusion that the 2d gravity partition functions that we want to study are nothing else but the partition functions of chiral gravity on $\Sigma_{g,n} \times S^1$. Punctures in the 2d theory become Wilson lines in the 3d gravity theory that wrap the thermal circle.

Some comments are in order. First, the reader may worry that this derivation was a bit formal, since both the integral over Teichmüller space diverges and $\mathrm{Map}(\Sigma_{g,n})$ is an infinite group. There are however several ways to get around this. For example, the inner product (4.2) can be derived from the path integral of 3d gravity, see [87]. Gauging of $\mathrm{Map}(\Sigma_{g,n})$ in that path integral indeed reduces the integral to the quotient $\mathcal{M}_{g,n} = \mathcal{T}_{g,n} / \mathrm{Map}(\Sigma_{g,n})$. Thus we could have gauged $\mathrm{Map}(\Sigma_{g,n})$ from the very beginning and the gravity path integral can be brought to the form (4.7), thus circumventing the formal step in our argument. One can also compute equivariantly with respect to $\mathrm{Map}(\Sigma_{g,n})$. The Hilbert space carries an action of the mapping class group that acts by crossing and while there are infinitely many conformal blocks, one can decompose the Hilbert space into irreducible representations of $\mathrm{Map}(\Sigma_{g,n})$ and every irreducible representation appears only finitely many times. This removes the formal infinities appearing in the problem.

Second, the partition function appearing in (4.7) has no reason to be a positive integer. This is perhaps confusing since we would have expected that the gravity partition function would count the number of states of the Hilbert space obtained after dividing by $\mathrm{Map}(\Sigma_{g,n})$. Such a chiral gravity theory can indeed be defined. However it differs in a rather subtle way from what we discuss here. To define it, one starts from a compactified phase space $\overline{\mathcal{M}}_{g,n}$, but the theory explicitly depends on the chosen compactification. Consistency then requires that the framing anomalies of the theory cancel, which imposes $c \in 24\mathbb{Z}$ and $h \in \mathbb{Z}$. Moreover, since $\mathcal{M}_{g,n}$ has orbifold singularities, one needs to include contributions from twisted sectors. Such a theory is discussed in [89]. However, since we do not insist on a fully three-dimensional interpretation, we do not have to worry that these partition functions are non integer-valued.

## 4.2 Quantization and index theorem

We will now discuss an alternative way to compute the chiral gravity partition function on $\Sigma_{g,n} \times S^1$, which will make contact with the intersection theory on the moduli space of Riemann surfaces. This discussion follows closely [89, 90]. Let us again start with the phase space of gravity, which is given by $\mathcal{T}_{g,n}$ (or $\mathcal{M}_{g,n}$ if we want to divide by $\mathrm{Map}(\Sigma_{g,n})$ before quantization). The symplectic form on $\mathcal{T}_{g,n}$ is the Weil-Petersson form $\frac{c}{48\pi^2} \omega_{\mathrm{WP}}(\ell_1, \ldots, \ell_n)$. In the case that punctures are present, the external conformal weights $h_j$ are related to the lengths of the geodesic boundaries of the Weil-Petersson form as follows:

$$h_j = \frac{c}{24}\left(1 + \frac{\ell_j^2}{4\pi^2}\right). \tag{4.8}$$

To pass to the quantum theory, we want to quantize this phase space. Since Teichmüller space is a Kähler manifold, a convenient way of doing so is to use Kähler quantization. The result is that the wavefunctions are holomorphic sections of a line bundle $\mathscr{L}$ over Teichmüller space whose curvature is

$$c_1(\mathscr{L}) = \frac{c}{48\pi^2}\,\omega_{\mathrm{WP}}(\ell_1, \ldots, \ell_n). \tag{4.9}$$

Holomorphic sections of this line bundle can be identified with Liouville conformal blocks which lead to the description of the Hilbert space discussed above. The non-triviality of the line bundle is an expression of the conformal anomaly, since conformal blocks are not *functions* of the moduli; this is only true after fixing an explicit metric, i.e. trivialization of the bundle. Of course, $\mathcal{T}_{g,n}$ is a contractible space and thus we could trivialize this line bundle (in a non-canonical way). However, this will not be true once we restrict to moduli space and thus it is important to keep the curvature at this point.

We can then compute the partition function of chiral gravity on $\Sigma_{g,n} \times S^1$ by counting the number of holomorphic sections of this line bundle. It can be computed from the Hirzebruch-Riemann-Roch index theorem:

$$\dim \mathcal{H}_{g,n} = \int_{\mathcal{T}_{g,n}} \mathrm{td}(\mathcal{T}_{g,n}) \, e^{\frac{c}{48\pi^2}\omega_{\mathrm{WP}}(\ell_1,\dots,\ell_n)}. \tag{4.10}$$

Here, td denotes the Todd class of the tangent bundle. Thus the partition function of 3d gravity may be computed by restricting this divergent integral to moduli space:

$$Z_{\Sigma_{g,n} \times S^1} = \int_{\mathcal{M}_{g,n}} \mathrm{td}(\mathcal{M}_{g,n}) \, e^{\frac{c}{48\pi^2}\omega_{\mathrm{WP}}(\ell_1,\dots,\ell_n)}. \tag{4.11}$$

We used that the tangent bundle of moduli space has the same curvature as the tangent bundle of Teichmüller space and thus the characteristic classes agree. We can then extend the integral to $\overline{\mathcal{M}}_{g,n}$ and treat the integrand as cohomology classes. Using that the cohomology class of the Weil-Petersson form is given by (A.6) and the relation of the lengths and conformal weights (4.8), we arrive at eq. (2.20).

This computation contains the same formal infinities as before. However, this is again not a problem. We could have used an equivariant version of the index theorem to render the expressions well-defined. We also remark that the proof of the index theorem via the heat kernel is a local computation which is unaffected by the compactness of the manifold.

Thus, we arrive at a central claim of the paper, namely

$$V_{g,n}^{(b)}(P_1,\dots,P_n) = \int_{\overline{\mathcal{M}}_{g,n}} \mathrm{td}(\mathcal{M}_{g,n}) \, e^{\frac{c}{24}\kappa_1 + \sum_{j=1}^n (P_j^2 - \frac{1}{24})\psi_j}. \tag{4.12}$$

We recall the definition of the $\psi$- and $\kappa$-classes for the benefit of the reader in appendix A. We also extended the integral to the Deligne-Mumford compactification of $\overline{\mathcal{M}}_{g,n}$ in order to use the standard intersection theory on moduli space.

We can then use the following formula for the Todd class of the tangent bundle:

$$\mathrm{td}(\mathcal{M}_{g,n}) = \exp\left(-\frac{13}{24}\kappa_1 + \frac{1}{24}\sum_{j=1}^n \psi_j - \sum_{m\geq 1} \frac{B_{2m}}{(2m)(2m)!}\kappa_{2m}\right), \tag{4.13}$$

where $B_{2m}$ are the Bernoulli numbers. This formula was derived in [89] for the tangent bundle of $\overline{\mathcal{M}}_{g,n}$. The two formulas differ slightly, because the treatment of the boundary divisor is different. It is clear that the formula of interest should not get contributions from boundary divisors since it is obtained by restricting an integrand on $\mathcal{T}_{g,n}$. To derive this formula, one applies the Grothendieck-Riemann-Roch theorem to the forgetful map $\mathcal{M}_{g,n+1} \longrightarrow \mathcal{M}_{g,n}$ and the line bundle of quadratic differentials on the Riemann surface, which in turn span the cotangent space of $\mathcal{M}_{g,n}$. This application is standard in algebraic geometry, see e.g. [91] for a general context. We thus obtain

$$V_{g,n}^{(b)}(P_1,\dots,P_n) = \int_{\overline{\mathcal{M}}_{g,n}} \exp\left(\frac{c-13}{24}\kappa_1 + \sum_{j=1}^n P_j^2 \psi_j - \sum_{m\geq 1} \frac{B_{2m}}{(2m)(2m)!}\kappa_{2m}\right). \tag{4.14}$$

This reproduces eq. (2.20). Similar generalizations of the Weil-Petersson volumes from an intersection point of view were considered for example in [92]. This establishes the links between the worldsheet formulation, 3d gravity and the intersection theory on $\overline{\mathcal{M}}_{g,n}$ as depicted in figure 1.

## 4.3 Dilaton and string equation

Fully analyzing (4.14) requires fairly deep mathematics in the form of topological recursion, which we will discuss in section 5.3. However, it is more straightforward to deduce two simpler equations for the quantum volumes directly. Borrowing terminology from topological gravity, we call them the dilaton and the string equation. We already wrote them down without further explanation in eqs. (2.21a) and (2.21b) and repeat them here

$$V^{(b)}_{g,n+1}(P = \tfrac{i\widehat{Q}}{2}, \mathbf{P}) - V^{(b)}_{g,n+1}(P = \tfrac{iQ}{2}, \mathbf{P}) = (2g-2+n)V^{(b)}_{g,n}(\mathbf{P}),$$ 

(4.15a)

$$\int_{\frac{iQ}{2}}^{\frac{i\widehat{Q}}{2}} 2P \, dP \, V^{(b)}_{g,n+1}(P, \mathbf{P}) = \sum_{j=1}^{n} \int_{0}^{P_j} 2P_j \, dP_j \, V^{(b)}_{g,n}(\mathbf{P}).$$ 

(4.15b)

The reason for the existence of these equations is that one can integrate out the location of the $(n+1)$-st marked point of the integrand on the LHS. In the language of the cohomology of the moduli space, this is implemented by the pushforward in cohomology. Let

$$\pi : \overline{\mathcal{M}}_{g,n+1} \longrightarrow \overline{\mathcal{M}}_{g,n},$$ 

(4.16)

be the map between moduli spaces that forgets the location of the $(n+1)$-st marked point. Then integrating over its location is given by the pushforward

$$\pi_* : H^\bullet(\overline{\mathcal{M}}_{g,n+1}, \mathbb{C}) \longrightarrow H^{\bullet-2}(\overline{\mathcal{M}}_{g,n}, \mathbb{C}).$$ 

(4.17)

In appendix D, we show that the integrands of the dilaton and string equation (4.15a) and (4.15b) are simple to pushforward and the result can again be expressed in terms of the cohomology classes of the integrand for the quantum volumes. Integrating over $\overline{\mathcal{M}}_{g,n}$ then gives the two equations. We refer the reader to appendix D for details.

## 4.4 Disk and trumpet partition functions

The 3d gravity point of view is very useful to understand the meaning of asymptotic boundaries, since an asymptotically (nearly) $AdS_2$ boundary uplifts simply to an asymptotically $AdS_3$ boundary.

The simplest topology with an asymptotic boundary is the disk $D^2$, for which the corresponding 3d topology is a solid cylinder. From the point of view of chiral gravity, it is thus clear that $Z_{D^2 \times S^1}$ evaluates to the vacuum Virasoro character of the boundary torus, see e.g. [93]. The vacuum Virasoro character depends on the thermal length $\tilde{\beta}$ of $S^1$. It is related by a modular transformation to the boundary circle of the disk, which plays the role of time in the dual matrix model of our two-dimensional gravity theory. We thus set $\beta = \frac{4\pi^2}{\tilde{\beta}}$. This recovers (2.16a). A similar argument determines the trumpet partition function (2.16b).

They can also directly be derived from the integral (4.12) over moduli space. The relevant moduli space for the disk is the Virasoro coadjoint orbit $\text{Diff}(S^1)/PSL(2,\mathbb{R})$, where $PSL(2,\mathbb{R})$ corresponds to the three reparametrization modes of the disk. Quantization of the phase space $\text{Diff}(S^1)/PSL(2,\mathbb{R})$ is thus achieved by quantizing Virasoro coadjoint orbits which leads again to Virasoro characters [94]. Finally, the integral (4.12) over $\text{Diff}(S^1)/PSL(2,\mathbb{R})$ can also be performed equivariantly, where $\beta$ enters as an equivariant parameter. One can then use

equivariant localization to compute it directly. We refer to [89, 95] for details on this. Similarly the trumpet partition function is obtained by the quantization of a generic Virasoro coadjoint orbit Diff($S^1$)/$S^1$.

It now also follows that one can glue the trumpet partition function to the bulk part of the two-dimensional geometry as in JT gravity. We already determined the correct gluing measure $2P \, dP$ in eq. (4.4). Indeed, when gluing a trumpet, the geodesic where we are gluing the trumpet is unique and is in particular preserved by any mapping class group transformation. Thus the only mapping class group transformation interacting non-trivially with the trumpets are Dehn twists along the gluing geodesic and hence taking the $\mathbb{Z}$ quotient as in (4.4) reduces the integral over Teichmüller space to an integral over moduli space. Of course there can be still non-trivial mapping class group transformations acting only on the bulk part of the surface, but they do not interact with the gluing of trumpets. Hence (4.4) tells us that before integrating over $P$ we get a factor of $2P$, so that the total gluing measure is $2P \, dP$. Thus (2.18) follows.

## 4.5 Further properties of the quantum volumes

Contrary to the worldsheet definition, the intersection theory approach gives manifestly analytic expressions for the quantum volumes $V_{g,n}^{(b)}$. The integral over $\overline{\mathcal{M}}_{g,n}$ picks out the top form in the power series expansion of the integrand. Thus, it follows directly from (4.14) that the quantum volumes are polynomial in $c$ and $P_1^2, \ldots, P_n^2$ with rational coefficients

$$V_{g,n}^{(b)}(P_1, \ldots, P_n) \in \mathbb{Q}\left[c, P_1^2, \ldots, P_n^2\right]. \tag{4.18}$$

The degree is $3g - 3 + n$, which generalizes the well-known polynomial behaviour of the Weil-Petersson volumes [96].

This also makes it clear that eq. (4.14) exhibits the following unexpected duality symmetry:

$$V_{g,n}^{(ib)}(iP_1, \ldots, iP_n) = (-1)^{3g-3+n} V_{g,n}^{(b)}(P_1, \ldots, P_n). \tag{4.19}$$

Indeed, sending $c \to 26 - c$ and $P_j \to iP_j$ acts on (4.14) by a minus sign on the coefficients of $\kappa_1$ and $\psi_j$ in the exponent. The other classes are in $H^{4\bullet}(\overline{\mathcal{M}}_{g,n})$ and thus we simply act by a minus sign on $H^{4\bullet+2}(\overline{\mathcal{M}}_{g,n})$. The integral picks out the top form on moduli space, which leads to the identification (4.19). In the presence of a boundary, it follows from (2.18) that the symmetry is modified to

$$Z_{g,n}^{(ib)}(\beta_1, \ldots, \beta_n) = i^{2g-2+n} Z_{g,n}^{(b)}(-\beta_1, \ldots, -\beta_n). \tag{4.20}$$

Note however that because of the appearance of the square root in the trumpet partition function (2.16b), the symmetry extends to a $\mathbb{Z}_4$ symmetry.

From the worldsheet point of view, such a duality symmetry cannot even be defined, since the central charge of the timelike Liouville theory is constrained to $\hat{c} \leq 1$ and thus only makes sense after analytically continuing the result for the quantum volumes in $c$ and $P_j$. However, the presence of this symmetry means that timelike and spacelike Liouville theory are at least morally on democratic footing.

# 5 Virasoro matrix integral

In this section we study the dual matrix integral for the Virasoro minimal string. We start by collecting some important equations and results in the bigger scheme of random matrix theory, particularly Hermitian matrix integrals.

## 5.1 A brief review of matrix integrals

A Hermitian matrix integral is an integral of the form

$$\mathcal{M}_N = \int_{\mathbb{R}^{N^2}} [dH] \, e^{-N \operatorname{tr} V(H)}, \tag{5.1}$$

where $H$ is a Hermitian $N \times N$ matrix and $V(H)$ is a polynomial in $H$. Matrix integrals of the form (5.1) are solvable in the large $N$ limit [97–102] (for reviews see [35, 103, 104]) and $\mathcal{F}_N \equiv -\log(\mathcal{M}_N)$ admits a perturbative expansion in powers of $1/N$. Using a saddle point approximation we can obtain the leading contribution (of order $N^2$) and using e.g. orthogonal polynomials, loop equations and topological recursion we get higher-order contributions [105–107]. Of particular interest is the so called double scaling limit. In this limit the full genus expansion can be reduced to solving a differential equation [32–34, 108].

Every Hermitian matrix can be diagonalized using a unitary matrix $\mathcal{U}$ such that $H = \mathcal{U} D_H \mathcal{U}^\dagger$ with $D_H \equiv \operatorname{diag}(\lambda_1, \dots, \lambda_N)$ a real diagonal matrix. The trace is invariant under this diagonalisation, but the measure in (5.1) picks up a non-trivial Jacobian: This Jacobian is known as the Vandermonde determinant $\Delta_N(\lambda) \equiv \prod_{i \neq j} |\lambda_i - \lambda_j|$. Explicitly we have

$$\mathcal{M}_N = \int_{\mathbb{R}^N} \prod_{i=1}^N d\lambda_i \, e^{-N^2 S[\lambda]}, \quad S[\lambda] = \frac{1}{N} \sum_{i=1}^N V(\lambda_i) - \frac{1}{N^2} \sum_{i \neq j} \log |\lambda_i - \lambda_j|. \tag{5.2}$$

Note the reduction from $N^2$ to $N$ degrees of freedom. The saddle point equations for (5.2) are

$$V'(\lambda_i) = \frac{2}{N} \sum_{j \neq i} \frac{1}{\lambda_i - \lambda_j}. \tag{5.3}$$

To solve this equation we introduce the normalized eigenvalue density

$$\varrho(\lambda) = \frac{1}{N} \sum_{i=1}^N \delta(\lambda - \lambda_i), \quad \int_{a_-}^{a_+} d\lambda \, \varrho(\lambda) = 1, \tag{5.4}$$

where we assume that all eigenvalues are located within the strip $[a_-, a_+]$ on the real axis. Additionally we introduce the resolvent

$$R_N(E) \equiv \frac{1}{N} \operatorname{Tr}(E \, \mathbb{1}_N - H)^{-1} = \frac{1}{N} \sum_{i=1}^N \frac{1}{E - \lambda_i}, \quad E \in \mathbb{C} \setminus \{\lambda_i\}. \tag{5.5}$$

Sending $N \to \infty$ the sum can be replaced by an integral where each eigenvalue is weighted by its average density

$$\lim_{N \to \infty} R_N(E) \equiv R(E) = \int_{a_-}^{a_+} d\mu \, \frac{\varrho(\mu)}{E - \mu}, \tag{5.6}$$

where we assume that the eigenvalue distribution is connected and has compact support on a single real interval $[a_-, a_+]$. The resolvent relates to the eigenvalue density and the matrix potential through the following relations

$$\varrho(E) = \frac{1}{2\pi i} (R(E - i\varepsilon) - R(E + i\varepsilon)), \quad E \in \operatorname{supp}(\varrho), \tag{5.7a}$$

$$V'(E) = R(E + i\varepsilon) + R(E - i\varepsilon), \quad E \in \operatorname{supp}(\varrho), \tag{5.7b}$$

where $\varepsilon$ is a small positive number and we used the large $N$ limit of (5.3) to obtain (5.7b). Additionally it satisfies $\lim_{E \to \infty} E R(E) = 1$ which immediately follows from the definition (5.6).

In the next subsection we discuss methods to obtain correlation functions of the resolvents. These satisfy an expansion of the form

$$\langle R(E_1)\dots R(E_n)\rangle_{\text{conn.}} \approx \sum_{g=0}^{\infty} \frac{R_{g,n}(E_1,E_2,\dots,E_n)}{N^{2g-2+n}}. \tag{5.8}$$

On the right hand side the power of $N$ accounts for the genus and the number of boundaries. The resolvent (5.6) is equal to $R_{0,1}(E_1)$ in this expansion. Without providing details since they can be found in multiple recent papers (see e.g. in [36, 109]) we also have

$$R_{0,2}(E_1,E_2) = \frac{1}{4}\frac{1}{\sqrt{-E_1}\sqrt{-E_2}(\sqrt{-E_1}+\sqrt{-E_2})^2}. \tag{5.9}$$

This result is universal for matrix integrals with support on a single interval.

## 5.2 Density of states and resolvent

In the double scaling limit we take the limit $N \to \infty$ and zoom into one edge of the eigenvalue distribution. In this limit the perturbative eigenvalue distribution is supported on the entire real positive axis and becomes non-normalizable. The double-scaled matrix integral is perturbatively completely fixed by this density of eigenvalues. Upon double scaling the eigenvalue density is given by [36]

$$\varrho_0^{\text{total}}(E) = e^{S_0}\,\varrho_0^{(b)}(E), \tag{5.10}$$

and hence $e^{S_0}$ is a rough analogue of $N$ and plays the role of the parameter that controls the perturbative genus expansion. For example, (5.8) still holds after double scaling but with $N$ replaced by $e^{S_0}$. In the Virasoro matrix integral,

$$\varrho_0^{(b)}(E)\,dE = \rho_0^{(b)}(P)\,dP = 4\sqrt{2}\,\sinh(2\pi bP)\sinh(2\pi b^{-1}P)\,dP, \tag{5.11}$$

where $E = P^2 = h_P - \frac{c-1}{24}$ is the energy in the matrix model. For $b \to 0$, one of the sinh's linearizes and we recover the famous $\sinh(\sqrt{E})\,dE$ density of states of JT gravity [36]. As already stressed in section 2.3, this is the universal Cardy density of states that endows the Virasoro matrix integral with its name.

One way to obtain $\varrho_0^{(b)}(E)$ is through the inverse Laplace transform of the disk partition function (2.17a)

$$\varrho_0^{(b)}(E) = 2\sqrt{2}\,\frac{\sinh(2\pi b\sqrt{E})\sinh(2\pi b^{-1}\sqrt{E})}{\sqrt{E}} = \int_{-i\infty+\gamma}^{i\infty+\gamma}\frac{d\beta}{2\pi i}\,e^{\beta E}Z_{\text{disk}}^{(b)}(\beta), \tag{5.12}$$

where $\gamma \in \mathbb{R}_+$ is such that the contour is to the right of the singularities of $Z_{\text{disk}}^{(b)}$ in the complex $\beta$ plane.

Recall that the leading density of states $\varrho_0^{(b)}$ may also be computed as the discontinuity of the genus-zero contribution to the resolvent, see equation (5.7a). For $(g,n) \neq (0,1)$, all other resolvents may be obtained from the partition functions $Z_{g,n}^{(b)}$ (which are in turn related to the quantum volumes by gluing trumpets as in (2.18)) by Laplace transform as

$$R_{g,n}^{(b)}(-z_1^2,\dots,-z_n^2) = \int_0^{\infty}\left(\prod_{j=1}^{n}d\beta_j\,e^{-\beta_j z_j^2}\right)Z_{g,n}^{(b)}(\beta_1,\dots,\beta_n). \tag{5.13}$$

Here we have written the energies $E_i = -z_i^2$ as negative for convergence of the integrals, but may analytically continue to positive energies afterwards. Hence by combining eq. (2.18)

and (5.13) the quantum volumes themselves may be obtained from the resolvents by inverse Laplace transform

$$V_{g,n}^{(b)}(P_1, \ldots, P_n) = \int_{-i\infty+\gamma}^{i\infty+\gamma} \left( \prod_{j=1}^{n} \frac{\mathrm{d}z_j}{2\pi i} \, \mathrm{e}^{4\pi P_j z_j} \frac{\sqrt{2}z_j}{P_j} \right) R_{g,n}^{(b)}(-z_1^2, \ldots, -z_n^2), \qquad (5.14)$$

for $\gamma$ sufficiently large.

## 5.3 Topological recursion

We now define the spectral curve [36, 107] of the Virasoro matrix integral

$$y^{(b)}(z) = -2\sqrt{2}\pi \frac{\sin(2\pi bz)\sin(2\pi b^{-1}z)}{z}, \qquad (5.15)$$

where $z^2 \equiv -E$ as before. We also define $\omega_{0,1}^{(b)}(z) \equiv 2zy^{(b)}(z)\mathrm{d}z$. Adjusting our notation to [36] we introduce the following modified resolvents

$$\omega_{g,n}^{(b)}(z_1, \ldots, z_n) \equiv (-1)^n 2^n z_1 \ldots z_n R_{g,n}^{(b)}(-z_1^2, \ldots, -z_n^2)\mathrm{d}z_1 \ldots \mathrm{d}z_n. \qquad (5.16)$$

In particular using (5.16) it follows from (5.9)

$$\omega_{0,2}^{(b)}(z_1, z_2) = \frac{\mathrm{d}z_1\mathrm{d}z_2}{(z_1 - z_2)^2}, \qquad (5.17)$$

where a convenient branch choice was made. For $2g-2+n > 0$ we obtain the $\omega_{g,n}^{(b)}(z_1, \ldots, z_n)$ from the recursion

$$\omega_{g,n}^{(b)}(z_1, z_2, \ldots, z_n) = \mathrm{Res}_{z \to 0} \Big( K^{(b)}(z_1, z) \big[ \omega_{g-1,n+1}^{(b)}(z, -z, z_2, \ldots z_n) $$
$$+ \sum_{h=0}^{g} \sum_{\substack{\mathcal{I} \cup \mathcal{J} = \{z_2, \ldots z_n\} \\ \{h, \mathcal{I}\} \neq \{0, \emptyset\} \\ \{h, \mathcal{J}\} \neq \{g, \emptyset\}}} \omega_{h,1+|\mathcal{I}|}^{(b)}(z, \mathcal{I}) \omega_{g-h,1+|\mathcal{J}|}^{(b)}(-z, \mathcal{J}) \big] \Big), \qquad (5.18)$$

where the recursion kernel $K^{(b)}(z_1, z)$ is given by

$$K^{(b)}(z_1, z) \equiv \frac{\int_{-z}^{z} \omega_{0,2}^{(b)}(z_1, -)}{4\omega_{0,1}^{(b)}(z)} = -\frac{1}{(z_1^2 - z^2)} \frac{z}{8\sqrt{2}\pi \sin(2\pi bz)\sin(2\pi b^{-1}z)}. \qquad (5.19)$$

These are the loop equations of the double-scaled matrix integral in the language of topological recursion. It determines the resolvent correlators (5.8) completely from the initial data $R_{0,1}(E) \equiv R(E)$ (5.6). Let us list some of the $\omega_{g,n}^{(b)}$:

$$\omega_{0,1}^{(b)}(z_1) = -4\sqrt{2}\pi \sin(2\pi bz_1)\sin(2\pi b^{-1}z_1)\mathrm{d}z_1, \qquad (5.20a)$$

$$\omega_{0,2}^{(b)}(z_1, z_2) = \frac{\mathrm{d}z_1\mathrm{d}z_2}{(z_1 - z_2)^2}, \qquad (5.20b)$$

$$\omega_{0,3}^{(b)}(z_1, z_2, z_3) = -\frac{1}{(2\pi)^3 \times 2\sqrt{2}} \frac{\mathrm{d}z_1\mathrm{d}z_2\mathrm{d}z_3}{z_1^2 z_2^2 z_3^2}, \qquad (5.20c)$$

$$\omega_{0,4}^{(b)}(z_1, z_2, z_3, z_4) = \frac{1}{(2\pi)^4} \left( \frac{c-13}{96} + \frac{3}{8(2\pi)^2} \sum_{i=1}^{4} \frac{1}{z_i^2} \right) \frac{\mathrm{d}z_1\mathrm{d}z_2\mathrm{d}z_3\mathrm{d}z_4}{z_1^2 z_2^2 z_3^2 z_4^2}, \qquad (5.20d)$$

$$\omega_{1,1}^{(b)}(z_1) = -\frac{1}{24\pi\sqrt{2}}\left(\frac{c-13}{48}+\frac{3}{(4\pi)^2}\frac{1}{z_1^2}\right)\frac{dz_1}{z_1^2}\,, \tag{5.20e}$$

$$\omega_{1,2}^{(b)}(z_1,z_2) = \frac{1}{(4\pi)^6}\left[\frac{3}{z_1^2 z_2^2}+5\left(\frac{1}{z_2^4}+\frac{1}{z_1^4}\right)+\frac{2\pi^2}{3}(c-13)\left(\frac{1}{z_1^2}+\frac{1}{z_2^2}\right)\right.$$
$$\left.+\frac{\pi^4}{18}(c-17)(c-9)\right]\frac{dz_1 dz_2}{z_1^2 z_2^2}\,. \tag{5.20f}$$

Let us explain how the topological recursion can be obtained from the definition of the quantum volumes in terms of integrals over the moduli space of curves $\overline{\mathcal{M}}_{g,n}$. This is a straightforward application of the result of [59]. [59, Theorem 3.3] states that for any choice of initial data $\omega_{0,1}^{(b)}$, $\omega_{g,n}^{(b)}$ as computed from the topological recursion (5.18) is equal to the following intersection number of $\overline{\mathcal{M}}_{g,n}$:

$$\omega_{g,n}^{(b)}(z_1,\ldots,z_n) = 2^{3g-3+n}\int_{\overline{\mathcal{M}}_{g,n}} e^{\sum_m \tilde{t}_m \kappa_m}\prod_{j=1}^{n}\sum_{\ell\geq 0}\frac{\Gamma(\ell+\frac{3}{2})}{\Gamma(\frac{3}{2})}\frac{\psi_j^\ell\, dz_j}{z_j^{2\ell+2}}\,. \tag{5.21}$$

The numbers $\tilde{t}_m$ are defined in terms of $\omega_{0,1}^{(b)}$ as follows. We expand $\omega_{0,1}^{(b)}(z)$ in (5.20a) for small $z$ leading to

$$\omega_{0,1}^{(b)}(z) = \sum_{m\geq 0}\frac{\Gamma(\frac{3}{2})t_m}{\Gamma(m+\frac{3}{2})}z^{2m+2}\,dz\,. \tag{5.22}$$

The coefficients $\tilde{t}_m$ in (5.21) are then defined via the equality of the following power series in $u$

$$\sum_{m\geq 0}t_m u^m = \exp\left(-\sum_{m\geq 0}\tilde{t}_m u^m\right)\,. \tag{5.23}$$

In our case, it follows from (5.20) that

$$\tilde{t}_0 = -\frac{3}{2}\log(8\pi^2)+\pi i\,, \tag{5.24a}$$

$$\tilde{t}_1 = \frac{c-13}{24}(2\pi)^2\,, \tag{5.24b}$$

$$\tilde{t}_{2m} = -\frac{B_{2m}(2\pi)^{4m}}{(2m)(2m)!}\,,\quad m\geq 1\,. \tag{5.24c}$$

Using that $\kappa_0 = 2g-2+n$, we thus obtain

$$\omega_{g,n}^{(b)}(z_1,\ldots,z_n)$$

$$= (2\pi)^{6-6g-3n}2^{-\frac{n}{2}}(-1)^n\int_{\overline{\mathcal{M}}_{g,n}}\exp\left(\frac{c-13}{24}(2\pi)^2\kappa_1-\sum_{m\geq 1}\frac{B_{2m}(2\pi)^{4m}}{(2m)(2m)!}\kappa_{2m}\right)\sum_{\ell\geq 0}\frac{\Gamma(\ell+\frac{3}{2})}{\Gamma(\frac{3}{2})}\frac{\psi_j^\ell\, dz_j}{z_j^{2\ell+2}}$$

$$= 2^{-\frac{3n}{2}}(-\pi)^{-n}\int_{\overline{\mathcal{M}}_{g,n}}\exp\left(\frac{c-13}{24}\kappa_1-\sum_{m\geq 1}\frac{B_{2m}\kappa_{2m}}{(2m)(2m)!}\right)\sum_{\ell\geq 0}\frac{\Gamma(\ell+\frac{3}{2})}{\Gamma(\frac{3}{2})(2\pi)^{2\ell}}\frac{\psi_j^\ell\, dz_j}{z_j^{2\ell+2}}$$

$$= \int\prod_j(-4\sqrt{2}\pi P_j dP_j\, e^{-4\pi z_j P_j})\int_{\overline{\mathcal{M}}_{g,n}}\exp\left(\frac{c-13}{24}\kappa_1-\sum_{m\geq 1}\frac{B_{2m}\kappa_{2m}}{(2m)(2m)!}\right)\sum_{\ell\geq 0}\frac{P_j^{2\ell}\psi_j^\ell\, dz_j}{\ell!}$$

$$= \left[\int_0^\infty\prod_j(-4\sqrt{2}\pi P_j dP_j\, e^{-4\pi z_j P_j})V_{g,n}^{(b)}(P_1,\ldots,P_n)\right]dz_1\ldots dz_n\,, \tag{5.25}$$

where we used the definition of the quantum volumes in terms of intersection numbers given in eq. (4.14). This formula is valid for $\mathrm{Re}\, z_j > 0$, but can be extended to any complex value of $z_j$ by analytic continuation.

For concreteness we can confirm the above relation (5.25) for the quantum volume $\mathsf{V}_{0,4}^{(b)}$ (2.19a) of the four punctured sphere and the quantum volume $\mathsf{V}_{1,1}^{(b)}$ (2.19b) of the once punctured disk. Using also the expressions for (5.20d) and (5.20e) we easily confirm

$$\omega_{0,4}^{(b)}(z_1, z_2, z_3, z_4) = \left[ (-4\sqrt{2}\pi)^4 \int_0^\infty \prod_{j=1}^4 (P_j dP_j\, e^{-4\pi z_j P_j}) \Big( \frac{c-13}{24} + \sum_{j=1}^4 P_j^2 \Big) \right] dz_1 dz_2 dz_3 dz_4 \,, \tag{5.26a}$$

$$\omega_{1,1}^{(b)}(z_1) = \left[ (-4\sqrt{2}\pi) \int_0^\infty (P_1 dP_1\, e^{-4\pi z_1 P_1}) \Big( \frac{c-13}{576} + \frac{1}{24} P_1^2 \Big) \right] dz_1 \,. \tag{5.26b}$$

This provides the crucial link between intersection theory and the Virasoro matrix integral and hence the last missing arrow in figure 1. The same perturbative data can now be expressed in terms of the resolvents/differentials $\omega_{g,n}^{(b)}$, the partition functions $Z_{g,n}^{(b)}$ or the quantum volumes $\mathsf{V}_{g,n}^{(b)}$. They carry all the same information and are related by simple integral transforms, which we summarize in figure 6. We have already seen most of the required relations in this triangle diagram. For completeness, let us also state the last two relations,

$$\mathsf{V}_{g,n}^{(b)}(P_1, \dots, P_n) = \int_\Gamma \left( \prod_{j=1}^n \frac{d\beta_j}{2\pi i} \sqrt{\frac{2\pi}{\beta_j}} e^{\beta_j P_j^2} \right) Z_{g,n}^{(b)}\Big( \tfrac{4\pi^2}{\beta_1}, \dots, \tfrac{4\pi^2}{\beta_n} \Big) \,, \tag{5.27a}$$

$$Z_{g,n}^{(b)}(\beta_1, \dots, \beta_n) = \int_\Gamma \left( \prod_{j=1}^n \frac{du_j}{2\pi i} e^{\beta_j u_j} \right) R_{g,n}^{(b)}(-u_1, \dots, -u_n) \,, \tag{5.27b}$$

where in both cases the integration contours $\Gamma$ are vertical and to the right of all singularities of the relevant integrands.

## 5.4 Deformed Mirzakhani recursion relation

We can translate the topological recursion (5.18) into a recursion relation for the quantum volumes $\mathsf{V}_{g,n}^{(b)}$. For the original case of Mirzakhani's recursion, this was done for the Weil-Petersson volumes in [110], while various generalizations with supersymmetry were considered in [109]. Let us first note that since the differentials $\omega_{g,n}^{(b)}$ are polynomial in inverse powers of $z_j^{-2}$, we can rewrite (5.14) as

$$\mathsf{V}_{g,n}^{(b)}(P_1, \dots, P_n) = \int_\Gamma \prod_{j=1}^n \frac{i\, e^{4\pi z_j P_j}}{2\sqrt{2}\pi P_j} \omega_{g,n}^{(b)}(z_1, \dots, z_n) = \prod_{j=1}^n \mathrm{Res}_{z_j=0} \frac{-e^{4\pi z_j P_j}}{\sqrt{2} P_j} \omega_{g,n}^{(b)}(z_1, \dots, z_n) \,, \tag{5.28}$$

where again $\Gamma$ is a contour that runs on the positively oriented shifted imaginary axis to the right of all singularities of the integrand. This representation is valid for $\mathrm{Re}\, P_j > 0$, otherwise the result follows from analytic continuation. In the second representation, we used that $z_j = 0$ is the only singularity of $\omega_{g,n}^{(b)}$, provided that $3g - 3 + n \geq 0$.

Let us derive the first term in (2.13) from the topological recursion, all other terms are obtained by very similar computations. We can set $n = 1$, since all $P_j$'s in $\mathbf{P}$ are spectators.

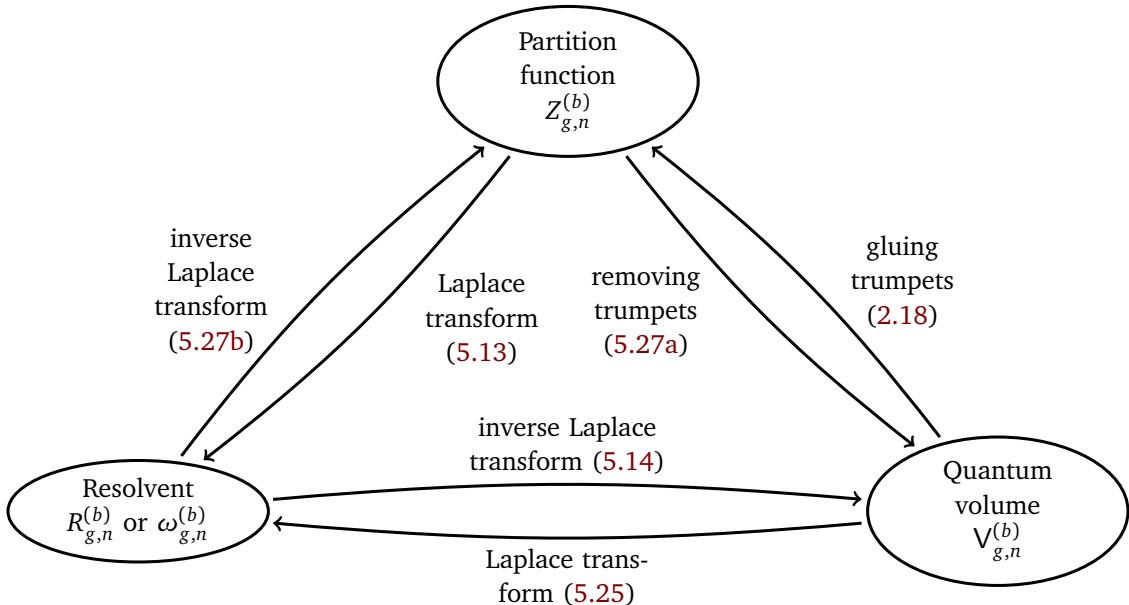

Figure 6: There are three quantities that all capture the same information that we discussed. They are all related by simple integral transformations, which we summarize here. We also recall that the differentials $\omega_{g,n}^{(b)}$ are just a more convenient way to write the resolvent; they are simply related via (5.16).

We have

$$
\begin{aligned}
P_1 V_{g,1}^{(b)}(P_1) &= -\frac{1}{\sqrt{2}} \operatorname*{Res}_{z_1=0} e^{4\pi z_1 P_1} \, \omega_{g,1}^{(b)}(z_1) \\
&\supset -\frac{1}{\sqrt{2}} \operatorname*{Res}_{z_1=0} e^{4\pi z_1 P_1} \operatorname*{Res}_{z=0} K^{(b)}(z_1,z) \, \omega_{g-1,2}^{(b)}(z,-z) \\
&= -\frac{1}{\sqrt{2}} \operatorname*{Res}_{z_1=0} e^{4\pi z_1 P_1} \operatorname*{Res}_{z=0} K^{(b)}(z_1,z) \, \omega_{g-1,2}^{(b)}(z,z) \\
&= -4\pi^2 \sqrt{2} \operatorname*{Res}_{z_1=0} \operatorname*{Res}_{z=0} e^{4\pi z_1 P_1} K^{(b)}(z_1,z) \int (2P\,dP)(2P'\,dP') e^{-4\pi z(P+P')} V_{g-1,2}^{(b)}(P,P').
\end{aligned}
$$
(5.29)

We used that all the multi-differentials (except for $\omega_{0,2}^{(b)}$) are symmetric in $z_j$. We can commute the two residues as follows:

$$
\operatorname*{Res}_{z_1=0} \operatorname*{Res}_{z=0} = \operatorname*{Res}_{z=0} \operatorname*{Res}_{z_1=z} + \operatorname*{Res}_{z=0} \operatorname*{Res}_{z_1=-z} \,,
$$
(5.30)

since as a function of $z_1$, the appearing function only has poles at $z_1 = z$ and $z_1 = -z$. Using the explicit form of the recursion kernel (5.19) we can take the $z_1$-residue, which leads to

$$
\begin{aligned}
P_1 V_{g,1}^{(b)}(P_1) &\supset \operatorname*{Res}_{z=0} \frac{\pi \sinh(4\pi P_1 z)}{2 \sin(2\pi b z) \sin(2\pi b^{-1} z)} \int (2P\,dP)(2P'\,dP') e^{-4\pi z(P+P')} V_{g-1,2}^{(b)}(P,P') \\
&= \operatorname*{Res}_{t=0} \frac{\pi \sin(4\pi P_1 t)}{2 \sinh(2\pi b t) \sinh(2\pi b^{-1} t)} \int (2P\,dP)(2P'\,dP') e^{4\pi i t(P+P')} V_{g-1,2}^{(b)}(P,P'),
\end{aligned}
$$
(5.31)

where we set $z = -it$ in the last equality. We can now rewrite the residue integral as a

difference of two integrals as follows:

$$
P_1 \mathsf{V}_{g,1}^{(b)}(P_1) \supset \left( \int_{\mathbb{R}-i\varepsilon} - \int_{\mathbb{R}+i\varepsilon} \right) \frac{\mathrm{d}t \, \sin(4\pi P_1 t)}{4i \sinh(2\pi b t)\sinh(2\pi b^{-1}t)}
$$
$$
\times \int (2P\,\mathrm{d}P)(2P'\,\mathrm{d}P') e^{4\pi i t(P+P')} \mathsf{V}_{g-1,2}^{(b)}(P,P')
$$
$$
= \int_{\mathbb{R}-i\varepsilon} \frac{\mathrm{d}t \, \sin(4\pi P_1 t)}{4i \sinh(2\pi b t)\sinh(2\pi b^{-1}t)} \int (2P\,\mathrm{d}P)(2P'\,\mathrm{d}P') e^{-4\pi i t(P+P')} \mathsf{V}_{g-1,2}^{(b)}(P,P')
$$
$$
- \int_{\mathbb{R}+i\varepsilon} \frac{\mathrm{d}t \, \sin(4\pi P_1 t)}{4i \sinh(2\pi b t)\sinh(2\pi b^{-1}t)} \int (2P\,\mathrm{d}P)(2P'\,\mathrm{d}P') e^{4\pi i t(P+P')} \mathsf{V}_{g-1,2}^{(b)}(P,P').
$$

$$(5.32)$$

We used that the integral over $P$ and $P'$ is only absolutely convergent for $\operatorname{Im} t > 0$ and is otherwise defined by analytic continuation. However, it is an even function in $t$ and can thus be obtained by replacing $t \to -t$ for the contour $\mathbb{R}-i\varepsilon$. At this point all integrals are absolutely convergent and thus we can exchange the $t$-integral with the $P$ and $P'$ integral. This gives the desired form of Mirzakhani's recursion relation (2.13), with kernel

$$
H(x,y) = \int_{\mathbb{R}-i\varepsilon} \mathrm{d}t \, \frac{\sin(4\pi y t)\, e^{-4\pi i x t}}{4i\sinh(2\pi b t)\sinh(2\pi b^{-1}t)} - \int_{\mathbb{R}+i\varepsilon} \mathrm{d}t \, \frac{\sin(4\pi y t)\, e^{4\pi i x t}}{4i\sinh(2\pi b t)\sinh(2\pi b^{-1}t)}. \quad (5.33)
$$

This can be further massaged as follows to bring it to the form (2.14). Indeed, we can rewrite both integrals in terms of principal value integrals by picking up some part of the residue at $t = 0$. This gives

$$
H(x,y) = \frac{y}{2} - \mathrm{PV} \int_{-\infty}^{\infty} \mathrm{d}t \, \frac{\sin(4\pi y t)(e^{4\pi i x t} - e^{-4\pi i x t})}{4i\sinh(2\pi b t)\sinh(2\pi b^{-1}t)}
$$
$$
= \frac{y}{2} - \int_0^{\infty} \mathrm{d}t \, \frac{\sin(4\pi x t)\sin(4\pi y t)}{\sinh(2\pi b t)\sinh(2\pi b^{-1}t)}, \quad (5.34)
$$

which is the form given in eq. (2.14).

Let us also mention that for an efficient implementation of Mirzkhani's recursion relation, we have the following integral formulas:

$$
\int_0^{\infty} (2x\,\mathrm{d}x)\, x^{2k} H(x,t) = F_k(t), \quad (5.35a)
$$

$$
\int_0^{\infty} (2x\,\mathrm{d}x)(2y\,\mathrm{d}y)\, x^{2k} y^{2\ell} H(x+y,t) = \frac{2(2k+1)!(2\ell+1)!}{(2k+2\ell+3)!} F_{k+\ell+1}(t), \quad (5.35b)
$$

where

$$
F_k(t) = \operatorname*{Res}_{u=0} \frac{(2k+1)!(-1)^{k+1}\sin(2tu)}{2^{2k+3} u^{2k+2} \sinh(bu)\sinh(b^{-1}u)} \quad (5.36)
$$

$$
= \sum_{0 \le \ell+m \le k+1} \frac{(-1)^{\ell+m}(2k+1)! B_{2\ell} B_{2m}(1-2^{1-2\ell})(1-2^{1-2m}) b^{2\ell-2m} t^{2k+3-2\ell-2m}}{(2\ell)!(2m)!(2k+3-2\ell-2m)!}. \quad (5.37)
$$

We provide such an implementation in the ancillary `Mathematica` file.

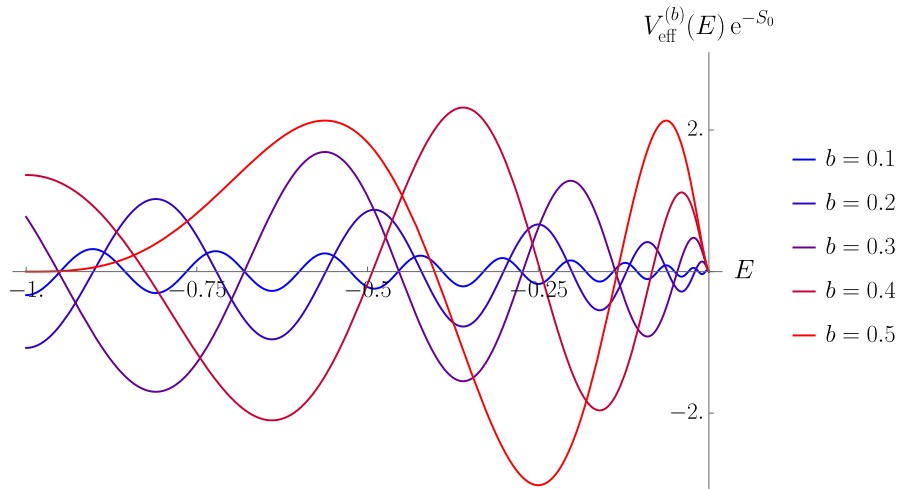

Figure 7: Plot of the effective potential $V_{\text{eff}}^{(b)}(E)$ of the double-scaled Virasoro matrix integral in the region $E < 0$, shown for several values of the parameter $b \neq 1$. Extrema of the effective potential occur at $E_{k,\pm}^* = -\frac{k^2 b^{\pm 2}}{4}$.

# Part III

# Evidence and applications

## 6  Non-perturbative effects

In this section we discuss some of the non-perturbative effects of the Virasoro matrix integral. Our discussion follows the logic in [36] and we avoid adding too many details as they can be found therein. In particular [36, eq. 155] expresses the leading perturbative and leading non-perturbative behaviour of the density of eigenvalues. For the eigenvalue density (5.11) of the Virasoro minimal string we find

$$\langle \varrho^{(b)}(E) \rangle \approx \begin{cases} e^{S_0} \varrho_0^{(b)}(E) - \frac{1}{4\pi E} \cos\left(2\pi e^{S_0} \int_0^E dE' \varrho_0^{(b)}(E')\right), & E > 0, \\ \frac{1}{-8\pi E} \exp\left(-V_{\text{eff}}^{(b)}(E)\right), & E < 0, \end{cases} \tag{6.1}$$

where the effective potential $V_{\text{eff}}^{(b)}$ is defined as

$$V_{\text{eff}}^{(b)}(E) = 2e^{S_0} \int_0^{-E} dx\, y^{(b)}(\sqrt{x}) = 2\sqrt{2}\, e^{S_0} \left(\frac{\sin(2\pi \widehat{Q}\sqrt{-E})}{\widehat{Q}} - \frac{\sin(2\pi Q\sqrt{-E})}{Q}\right), \tag{6.2}$$

with $Q = b^{-1} + b$ and $\widehat{Q} = b^{-1} - b$ defined in section 2.2. The effective potential is the combination of the potential $V(\lambda)$ (5.1) and the Vandermonde Jacobian (5.2). In figure 7 we see the oscillatory behaviour of the effective potential for some values of $b$. As in the JT case the term in the allowed region $E$ is rapidly oscillating and *larger* than the first subleading perturbative contribution. On the other side we find a non-zero contribution in the classically forbidden regime $E < 0$. It accounts for the possibility of one eigenvalue sitting in the regime $E < 0$.

### 6.1  Non-perturbative corrections to the quantum volumes

The leading non-perturbative correction to the quantum volume $\mathsf{V}_n^{(b)}(S_0; P_1, \ldots, P_n)$ is controlled by configurations of the matrix integral where one eigenvalue is in the classically for-

bidden region $E < 0$ and all the others are in the allowed region. Thus the leading non-perturbative correction is naturally given as an integral of the form

$$\int_{-\infty}^{0} dE \, \langle \varrho^{(b)}(E) \rangle \, \dots, \tag{6.3}$$

for some operator insertions $\cdots$ depending on the quantity under consideration. In particular, for the quantum volumes, the operator insertions can be determined intuitively as follows. For a more rigorous derivation, we refer to [36, appendix A].

Let us start by discussing the leading non-perturbative correction to the resummed partition function

$$Z_n^{(b)}(S_0; \beta_1, \dots, \beta_n) \equiv \sum_{g=0}^{\infty} Z_{g,n}^{(b)}(\beta_1, \dots, \beta_n) e^{-(2g-2+n)S_0}. \tag{6.4}$$

$Z_{g,n}^{(b)}(\beta_1, \dots, \beta_n)$ is obtained by inserting $\prod_{j=1}^{n} \mathrm{tr}(e^{-\beta_j H})$ into the matrix integral. Focussing now on the single eigenvalue in the forbidden region, the insertions in (6.3) should be $\prod_{j=1}^{n} e^{-\beta_j E}$. We can then compute the corresponding insertions for the quantum volumes $V_n^{(b)}$ by removing the trumpets, i.e. inverting (2.18). This basically amounts to an inverse Laplace transformation, see eq. (5.27a). However, in the process, we have to commute the integral over $E$ with the integral of the inverse Laplace transform, which is not quite allowed. This makes the present derivation non-rigorous. Let us anyway go ahead. The inverse Laplace transform predicts the following operator insertion for the quantum volumes, assuming that the energy $E < 0$:

$$\frac{1}{2\pi i} \int_{\gamma-i\infty}^{\gamma+i\infty} dx \, e^{P^2 x} \sqrt{\frac{2\pi}{x}} \, e^{-\frac{4\pi^2 E}{x}}, \tag{6.5}$$

for $\gamma$ a positive constant. By deforming the contour appropriately, this is easily evaluated to

$$\frac{\sqrt{2} \, e^{-4\pi|P|\sqrt{-E}}}{|P|}. \tag{6.6}$$

However this is not quite the right result because of the illegal exchange of contours. As usual, the correct result is analytic in $P$ and symmetric under exchange $P \to -P$. Following the analogous more careful derivation of Saad, Shenker and Stanford [36, appendix A], shows that the operator insertion is actually the average of both sign choices in the exponent. This is the unique choice that is both reflection symmetric and analytic in $P$. Summarizing, we hence have for the first non-perturbative correction (that we denote by a superscript [1])

$$V_n^{(b)}(S_0; P_1, \dots, P_n)^{[1]} = \int_{-\infty}^{0} dE \, \langle \varrho^{(b)}(E) \rangle \prod_{j=1}^{n} \frac{\sqrt{2} \sinh(4\pi P_j \sqrt{-E})}{P_j}, \tag{6.7}$$

where $\langle \varrho^{(b)}(E) \rangle$ is given by (6.1).

**Non-perturbative (in)stability.** Before continuing, we have to discuss an important issue. So far, the discussion makes it sound as if the non-perturbative corrections are unique. But this is actually not the case, because the integral in (6.7) is divergent unless $b = 1$. The reason for this is that unless $b = 1$, the sign of $V_{\mathrm{eff}}^{(b)}$ is indefinite and as a consequence, $\langle \varrho^{(b)}(E) \rangle$ can be arbitrarily large for negative energies. This means that the model is non-perturbatively unstable and all eigenvalues will tunnel to minima of $V_{\mathrm{eff}}^{(b)}(E)$ at smaller and smaller energies.

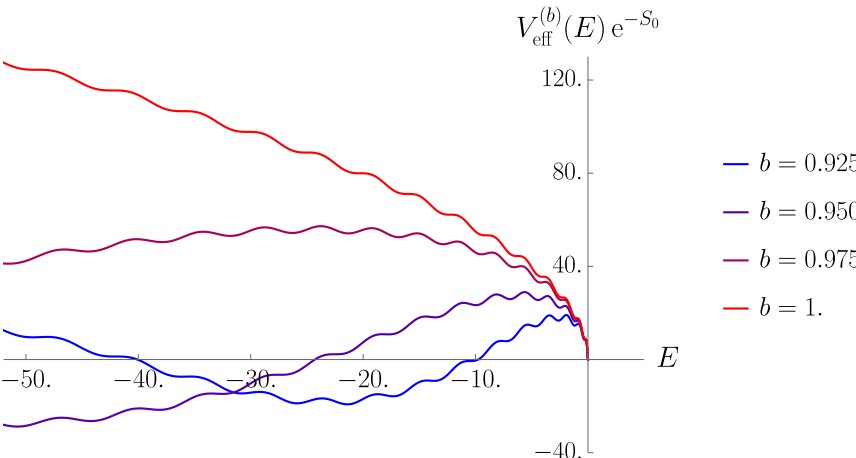

Figure 8: Plot of the effective potential $V_{\text{eff}}^{(b)}(E)$ of the double-scaled Virasoro matrix integral in the region $E < 0$, for $b$ close to one. For $b \neq 1$ the effective potential is oscillatory, while for $b$ exactly equal to one it is monotonically increasing.

For $b = 1$ instead, $V_{\text{eff}}^{(b)}(E)$ is monotonic and $\langle \varrho^{(b)}(E) \rangle$ decays exponentially as $E \to -\infty$. Thus the model is non-perturbatively stable. These two different behaviours are depicted in figure 8.

The non-perturbative instability does *not* mean that the model is non-sensical. Instead, the simplest way out is to deform the integration contour over the eigenvalues of the matrix. This however means that the non-perturbative completion of the model is not unique. As we shall discuss in section 8.2, the same ambiguities also arise when we reproduce these non-perturbative corrections from the worldsheet. For example, we can deform the integration contour to run to an extremum of $\langle \varrho^{(b)}(E) \rangle$ and then turn into the complex plane, as we do below.

Alternatively, one can also follow the route proposed in [111] to construct a different non-perturbative completion of the matrix integral, but it is not clear how to reproduce this structure from the worldsheet.

**Single instanton contribution.** Let us assume that $b \neq 1$ for now. We discuss the special case $b = 1$ further below in subsection 6.3. Each possible instanton correction on the worldsheet will be associated to one of the extrema of $V_{\text{eff}}^{(b)}(E)$. They come in two infinite families and are located at

$$E_{k,\pm}^* = -\frac{k^2 b^{\pm 2}}{4}, \quad k \in \mathbb{Z}_{\geq 1}. \tag{6.8}$$

For the one-instanton correction, we simply have to expand the integrand (6.7) around one of these saddle points. The corresponding non-perturbative correction is thus given by

$$\mathsf{V}_n^{(b)}(S_0; P_1, \ldots, P_n)_{k,\pm}^{[1]} =$$

$$= \int_{\gamma_{k,\pm}} dE \, \frac{-1}{8\pi E_{k,\pm}^*} e^{-V_{\text{eff}}^{(b)}(E_{k,\pm}^*) - \frac{1}{2}(E - E_{k,\pm}^*)^2 (V_{\text{eff}}^{(b)})''(E_{k,\pm}^*)} \prod_{j=1}^n \frac{\sqrt{2} \sinh(4\pi P_j \sqrt{-E_{k,\pm}^*})}{P_j}$$

$$= -\frac{i \, e^{-V_{\text{eff}}^{(b)}(E_{k,\pm}^*)}}{8\pi E_{k,\pm}^*} \sqrt{\frac{-\pi}{2(V_{\text{eff}}^{(b)})''(E_{k,\pm}^*)}} \prod_{j=1}^n \frac{\sqrt{2} \sinh(4\pi P_j \sqrt{-E_{k,\pm}^*})}{P_j}. \tag{6.9}$$

The contour $\gamma_{k,\pm}$ takes the form sketched in figure 9. We should also mention that we only kept the imaginary part of the expression (which does not get contributions from the real line),

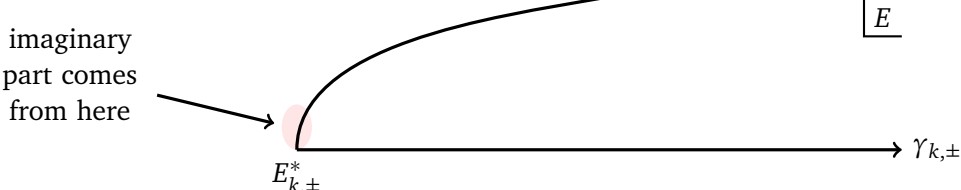

Figure 9: The integration contour $\gamma_{k,\pm}$ for the computation of instanton corrections in the sector $(k, \pm)$. We could have also chosen the contour reflected at the real axis, which would lead to the opposite sign in the result (6.12). This reflects the ambiguity of the non-perturbative completion discussed above on the matrix integral side.

since it is the only unambiguous part of the contour integral. The result is only one half of the Gaussian integral, since the contour turns into the complex plane. This is explained in more detail in [60]. To bring this expression into a form that is interpretable in string theory, let us denote

$$T_{k,\pm}^{(b)} = V_{\text{eff}}^{(b)}(E_{k,\pm}^*) = \frac{4\sqrt{2}\, e^{S_0} b^{\pm 1}(-1)^{k+1}\sin(\pi k b^{\pm 2})}{1 - b^{\pm 4}}\,. \tag{6.10}$$

$T_{k,\pm}^{(b)}$ has the physical interpretation of the tension of the corresponding ZZ-instanton in the bulk description. Notice that it may be positive or negative, reflecting that most of these instanton corrections should not live on the integration contour of the matrix integral. We will nonetheless be able to match them to the corresponding bulk quantities below. We also note that

$$(V_{\text{eff}}^{(b)})''(E_{k,\pm}^*) = T_{k,\pm}^{(b)} \frac{(V_{\text{eff}}^{(b)})''(E_{k,\pm}^*)}{V_{\text{eff}}^{(b)}(E_{k,\pm}^*)} = T_{k,\pm}^{(b)} \frac{4\pi^2(1 - b^{\mp 4})}{k^2}\,. \tag{6.11}$$

Thus we can rewrite (6.9) as

$$V_n^{(b)}(S_0; P_1, \ldots, P_n)_{k,\pm}^{[1]} = \frac{i\, e^{-T_{k,\pm}^{(b)}}}{2^{\frac{5}{2}} \pi^{\frac{3}{2}} (T_{k,\pm}^{(b)})^{\frac{1}{2}} (1 - b^{\pm 4})^{\frac{1}{2}} k} \prod_{j=1}^{n} \frac{\sqrt{2}\sinh(2\pi k b^{\pm 1} P_j)}{P_j}\,. \tag{6.12}$$

## 6.2 Large $g$ asymptotics of $V_{g,n}^{(b)}$

From the leading non-perturbative correction $V_n^{(b)}(S_0; P_1, \ldots, P_n)^{[1]}$ to $V_n^{(b)}(S_0; P_1, \ldots, P_n)$, one can also determine the asymptotic behaviour of the quantum volumes $V_{g,n}^{(b)}(P_1, \ldots, P_n)$ at large genus $g$ using resurgence techniques. Assuming $0 < b < 1$, the closest saddle-point to the origin is the contribution from the saddle point (6.8) $(1, +)$. The existence of non-perturbative corrections indicates that the series (2.8) is asymptotic. Let us look at its Borel transform,

$$\widetilde{V}_n^{(b)}(x; P_1, \ldots, P_n) = \sum_{g=0}^{\infty} \frac{x^{2g}}{(2g)!} V_{g,n}^{(b)}(P_1, \ldots, P_n)\,, \tag{6.13}$$

which has a finite radius of convergence in $x$. $V_n^{(b)}(S_0; P_1, \ldots, P_n)$ can then be recovered via a Laplace transform

$$V_n^{(b)}(S_0; P_1, \ldots, P_n) = e^{-(n-2)S_0} \int_0^{\infty} dx\, e^{-x}\, \widetilde{V}_n^{(b)}(x\, e^{-S_0}; P_1, \ldots, P_n)\,. \tag{6.14}$$

In the cases of interest to us, $\widetilde{V}_n^{(b)}$ will have singularities on the real axis and thus the integral over $x$ actually has to be deformed into the complex plane to give a non-perturbative completion of the summation. This leads to the same non-perturbative ambiguities that were already observed above. In particular, the large $g$ asymptotics of $V_{g,n}^{(b)}$ controls the radius of convergence of the Borel transform in the $x$-plane.

As we shall see, the quantum volumes, $V_{g,n}^{(b)}(P_1,\ldots,P_n)$ have the following universal behaviour as $g \to \infty$,

$$V_{g,n}^{(b)}(P_1,\ldots,P_n) \sim (2g)! \cdot AB^g g^C \,, \tag{6.15}$$

for functions $A$, $B$ and $C$ depending on $b$ and $n$ that we will now determine. The $(2g)!$ growth ensures that the Borel transform will have singularities in the $x$-plane. This behaviour implies that $\widetilde{V}_n^{(b)}(x;P_1,\ldots,P_n)$ behaves as

$$\widetilde{V}_n^{(b)}(x;P_1,\ldots,P_n) \sim A\Gamma(C+1)(1-Bx^2)^{-C-1} + \text{less singular}, \tag{6.16}$$

near the two singularities $x = \pm\frac{1}{\sqrt{B}}$ in the Borel plane. In particular, when $C \notin \mathbb{Z}$, the Borel transform has a branch cut running along the real axis starting from $x = \frac{1}{\sqrt{B}}$. We can then plug this behaviour into (6.14). The branch cut will lead to an imaginary part in the answer, which we can then compare with the first non-perturbative correction (6.12) of the quantum volumes. We deform the contour above the branch cut and only focus on the imaginary part of the answer. Thus resurgence predicts the following asymptotics of the quantum volumes

$$V_n^{(b)}(S_0;P_1,\ldots,P_n)^{[1]} = i\,e^{-(n-2)S_0}\int_{B^{-\frac{1}{2}}e^{S_0}}^{\infty} dx\,e^{-x}A\Gamma(C+1)\,\mathrm{Im}\left(1-Bx^2\,e^{-2S_0}\right)^{-C-1}$$

$$\sim \frac{A\pi i}{2^{C+1}B^{\frac{C+1}{2}}}\,e^{-B^{-\frac{1}{2}}e^{S_0}}e^{(3+C-n)S_0}\,. \tag{6.17}$$

Comparing to (6.12), we hence see that

$$B = e^{-2S_0}\left(T_{1,+}^{(b)}\right)^{-2}\,, \qquad C = n - \frac{7}{2}\,, \tag{6.18}$$

which is required to match the correct $S_0$ dependence. The fact that this matches the $S_0$-dependence of the non-perturbative correction to $V_n^{(b)}$ justifies our ansatz (6.15) *a posteriori*. We can then compare the prefactors to conclude

$$A = \frac{\left(e^{S_0}T_{1,+}^{(b)}\right)^{2-n}}{2^5\pi^{\frac{5}{2}}(1-b^4)^{\frac{1}{2}}}\prod_{j=1}^{n}\frac{2\sqrt{2}\sinh(2\pi bP_j)}{P_j}\,. \tag{6.19}$$

To summarize, we have extracted the following large $g$ behaviour of the quantum volumes,

$$V_{g,n}^{(b)}(P_1,\ldots,P_n) \overset{g\gg 1}{\sim} \frac{\prod_{j=1}^{n}\frac{\sqrt{2}\sinh(2\pi bP_j)}{P_j}}{2^{\frac{3}{2}}\pi^{\frac{5}{2}}(1-b^4)^{\frac{1}{2}}} \times \left(\frac{4\sqrt{2}b\sin(\pi b^2)}{1-b^4}\right)^{2-2g-n} \times \Gamma\left(2g+n-\frac{5}{2}\right), \tag{6.20}$$

where we need to assume that $0 < b < 1$. We also assume in this formula that $P_1,\ldots,P_n$ and $b$ are held constant while taking the large $g$ limit. It is interesting to note that even though the quantum volumes are all polynomial in $P_j^2$ and $c = 1 + 6(b+b^{-1})^2$, the large $g$ asymptotics is highly non-polynomial. We should also note that this formula implies that the string coupling $g_s = e^{-S_0}$ is renormalized to its effective value

$$g_s^{\mathrm{eff}} = \frac{1-b^4}{4\sqrt{2}b\sin(\pi b^2)}\,e^{-S_0} = (T_{1,+}^{(b)})^{-1}\,. \tag{6.21}$$

**Some consistency checks.** We can perform some simple consistency checks on this expression. We first remark that (6.20) is consistent with the dilaton equation (4.15a) in a somewhat non-trivial way. The LHS of the string equation (4.15b) vanishes for the asymptotic formula (6.20). This is consistent with the right hand side, since it is suppressed by one power of $\frac{1}{g}$.

Finally, (6.20) formally reduces to known formulas for the Weil-Petersson volumes when taking the limit $b \to 0$. Using (2.22) as well as (2.24), we obtain

$$V_{g,n}(\ell_1, \ldots, \ell_n) \sim \frac{(4\pi^2)^{2g-2+n}}{2^{\frac{3}{2}} \pi^{\frac{5}{2}}} \Gamma\left(2g + n - \tfrac{5}{2}\right) \prod_{j=1}^{n} \frac{2 \sinh(\frac{\ell_j}{2})}{\ell_j}, \tag{6.22}$$

which matches with the formulas derived in [36, 112–119]. In particular, [119] develops the large $g$ asymptotics much more systematically beyond the leading order.

**Explicit check.** We can compare (6.20) explicitly against the first few quantum volumes as computed from intersection theory or the recursion relation (2.13). Let us first focus on the case $n = 0$. For the Weil-Petersson volumes, this was done in [112] using more efficient algorithms for the computation of the volumes. In our case, we do not know of such an algorithm and the most efficient method for the computation of the volumes is the direct computation via intersection numbers on moduli space. We were able to evaluate the volumes up to $g = 12$ directly. The `Mathematica` notebook implementing this is attached to the publication as an ancillary file. The ratio of the quantum volumes and the asymptotic formula is displayed in figure 10. We also extrapolated the result to $g = \infty$ by using the general fact that the corrections to the asymptotic formula (6.20) are of the form

$$\frac{\mathsf{V}_{g,n}^{(b)}}{(6.20)} = \sum_{j=0}^{\infty} x_j g^{-j}. \tag{6.23}$$

This mirrors the fact that the string perturbation theory expansion is a power series in $g_s$ (as opposed to $g_s^2$) in the one-instanton sector). We fitted $x_0, \ldots, x_{10}$ from the data and plotted the asymptotic value given by $x_0$.

From the figure, it is clear that the asymptotic formula is good for $b$ well away from $b = 1$. This is expected since for $b = 1$, the saddle point approximation above breaks down because two saddles collide in that case.

We also checked the asymptotic formula for $\mathsf{V}_{g,1}^{(b)}(P_1)$. In figure 11, we plotted the ratio of the volume at genus 12 with the formula (6.20) as a function of $P_1$. The approximation is good for $b$ well away from $b = 1$ and $P_1$ sufficiently small.

## 6.3 The special case $b = 1$

The case $b = 1$ needs to be treated separately. For $b$ exactly equal to one the effective potential

$$V_{\text{eff}}^{(b=1)}(E) = \sqrt{2} \, e^{S_0} \left(4\pi\sqrt{-E} - \sin(4\pi\sqrt{-E})\right), \tag{6.24}$$

is no longer oscillatory (see figure 8). We will now repeat the analysis of sections 6.1 and 6.2 for this case. Our discussion will be rather brief, since many aspects are very similar.

**The one-instanton contribution.** In this case, the extrema are located at

$$E_k^* = -\frac{k^2}{4}, \quad k \in \mathbb{Z}_{\geq 1}. \tag{6.25}$$

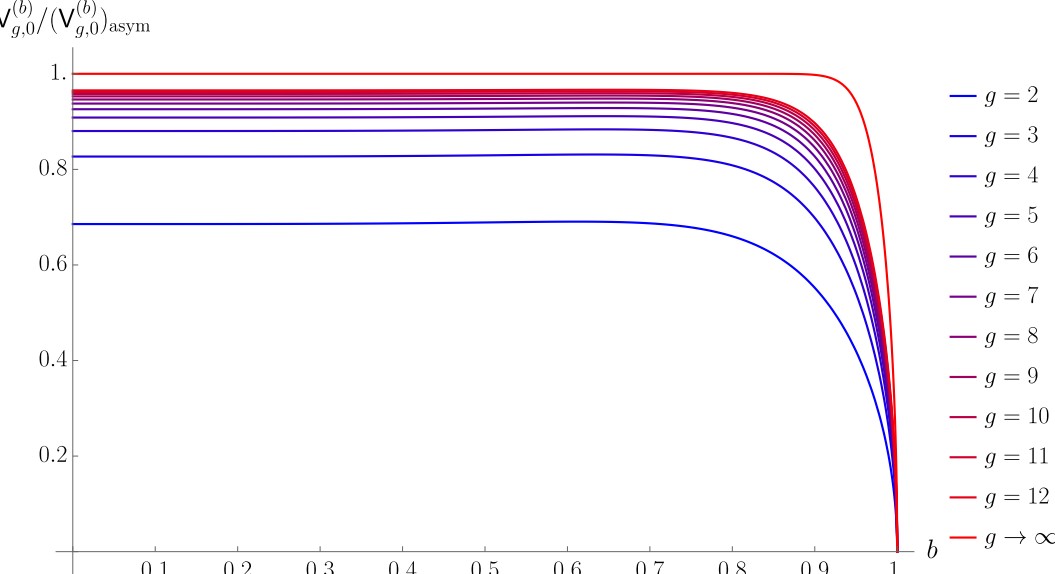

Figure 10: The ratio of the exact volumes and the asymptotic formula (6.20) up to $g = 12$. The last curve is the extrapolation of the low $g$ data to $g = \infty$.

They do not carry a subscript '+' or '−', since both cases coincide. In particular, all extrema of $V_{\text{eff}}^{(b=1)}$ have vanishing second derivative. Thus in the saddle point evaluation of the integral (6.7), we have to go to subleading order in the integral. We take the contour to be a steepest descent contour in the complex plane. Only the imaginary part of the one-instanton contribution is unambiguous since the real part depends on the precise details of the contour. We have

$$
\begin{aligned}
V_n^{(1)}(S_0; P_1, \ldots, P_n)_k^{[1]} &= i \operatorname{Im} \int_{\gamma_k} dE \, \frac{-1}{8\pi E_k^*} e^{-V_{\text{eff}}^{(1)}(E_k^*) - \frac{1}{6}(E - E_k^*)^3 (V_{\text{eff}}^{(b)})'''(E_k^*)} \\
&\quad \times \prod_{j=1}^n \frac{\sqrt{2} \sinh(4\pi P_j \sqrt{-E_k^*})}{P_j} \\
&= -\frac{i \, e^{-V_{\text{eff}}^{(1)}(E_k^*)} \Gamma(\frac{1}{3})}{8\pi E_k^* \left(-4\sqrt{3}(V_{\text{eff}}^{(1)})'''(E_k^*)\right)^{\frac{1}{3}}} \prod_{j=1}^n \frac{\sqrt{2} \sinh(4\pi P_j \sqrt{-E_k^*})}{P_j} \\
&= \frac{i \, e^{-2\sqrt{2} k \pi e^{S_0}} \Gamma(\frac{1}{3})}{8\pi^2 k \, (4\sqrt{6} \, e^{S_0})^{\frac{1}{3}}} \prod_{j=1}^n \frac{\sqrt{2} \sinh(2\pi k P_j)}{P_j} \, .
\end{aligned}
\tag{6.26}
$$

**Large genus asymptotics.** To extract the large genus behaviour of the quantum volumes $V_{g,n}^{(1)}$, we proceed as above. Matching (6.17) and (6.26) with $k = 1$ yields the asymptotics

$$
V_{g,n}^{(1)}(P_1, \ldots, P_n) \overset{g \gg 1}{\sim} \frac{\Gamma(\frac{1}{3}) \prod_{j=1}^n \frac{\sqrt{2} \sinh(2\pi P_j)}{P_j}}{2^{\frac{7}{3}} 3^{\frac{1}{6}} \pi^{\frac{8}{3}}} \left(\frac{1}{2\sqrt{2}\pi}\right)^{2g-2+n} \Gamma\left(2g - \frac{7}{3} + n\right).
\tag{6.27}
$$

Note that these quantum volumes grow slightly faster than the generic volumes, which is consistent with the fact that (6.20) diverges at $b = 1$. (6.27) is again consistent with the dilaton and the string equations (4.15a) and (4.15b), but we are not aware of simple checks beyond these.

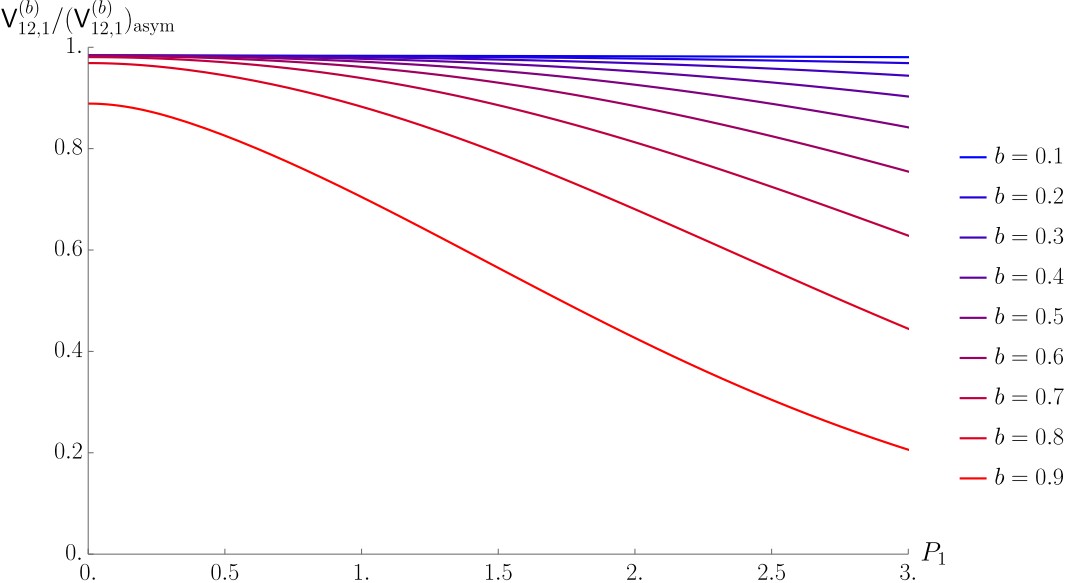

Figure 11: The ratio of the quantum volumes $\mathsf{V}_{12,1}^{(1)}$ and the asymptotic formula (6.20) for different values of $b$.

# 7 Worldsheet string perturbation theory

In this section, we will study the Virasoro minimal string (1.1) directly using worldsheet string perturbation theory. As emphasized in the introduction and in figure 2, we interpret string diagrams as computing quantum volumes of the worldsheet, rather than in terms of amplitudes of asymptotic string states in target spacetime.

## 7.1 Torus one-point diagram

In string perturbation theory, the torus one-point diagram is evaluated as

$$
\begin{aligned}
\mathsf{V}_{1,1}^{(b)}(P_1) &= \frac{(2\pi)^2}{2} \int_{F_0} \mathrm{d}^2\tau \left\langle \mathfrak{b}\tilde{\mathfrak{b}}\,\mathcal{V}_{P_1}(0) \right\rangle_{g=1} \\
&= \mathsf{N}\, \frac{(2\pi)^2}{2} \int_{F_0} \mathrm{d}^2\tau\, |\eta(\tau)|^4 \left\langle V_{P_1}(0) \right\rangle_{g=1} \left\langle \widehat{V}_{iP_1}(0) \right\rangle_{g=1},
\end{aligned}
\tag{7.1}
$$

where $F_0 = \{\tau \in \mathbb{C}\,|\, -\frac{1}{2} \le \mathrm{Re}\,\tau \le \frac{1}{2}, |\tau| \ge 1\}$ is the fundamental domain of the torus moduli space, and where we used the definition (2.6) for the physical vertex operators and the fact that the normalization $\mathsf{N}(P)$ is independent of $P$, see eq. (3.22). In our conventions, $\mathrm{d}^2\tau = \mathrm{d}\tau_1 \mathrm{d}\tau_2$ where $\tau = \tau_1 + i\tau_2$. Contrary to the sphere, see eq. (3.22), we do not have to introduce an additional arbitrary normalization $C_{\mathrm{T}^2}$ of the string path integral, since there is no corresponding counterterm on the torus and the normalization of the path integral is unambiguous and thus $C_{\mathrm{T}^2} = 1$. The factor of $(2\pi)^2$ in (7.1) arises from the correct normalization of the ghost path integral, see e.g. [120, section 7.3]. Finally, the factor of $\frac{1}{2}$ arises from the fact that each torus has a $\mathbb{Z}_2$ symmetry and we need to divide by the order of the automorphism group.

In our conventions, the Liouville one-point correlation functions on the torus $T^2$ with modulus $\tau$ in (7.1) admit the following Virasoro conformal block decompositions

$$\left\langle V_{P_1}(0)\right\rangle_{g=1} = \int_0^\infty \mathrm{d}P\, \rho_0^{(b)}(P) C_b(P_1, P, P) \mathcal{F}_{1,1}^{(b)}(P_1; P|q) \mathcal{F}_{1,1}^{(b)}(P_1; P|\overline{q}),\qquad(7.2a)$$

$$\left\langle \widehat{V}_{iP_1}(0)\right\rangle_{g=1} = \int_{\mathcal{C}} \mathrm{d}\widehat{P}\, \frac{(i\widehat{P})^2}{2\rho_0^{(b)}(i\widehat{P})} \widehat{C}_b(iP_1, \widehat{P}, \widehat{P}) \mathcal{F}_{1,1}^{(ib)}(iP_1; \widehat{P}|q) \mathcal{F}_{1,1}^{(ib)}(iP_1; \widehat{P}|\overline{q}),\qquad(7.2b)$$

where $\mathcal{F}_{1,1}^{(b)}(P_1; P|q)$ is the holomorphic torus one-point Virasoro conformal block at central charge $c = 1 + 6(b + b^{-1})^2$ with external weight $h_{P_1} = \frac{1}{4}(b + b^{-1})^2 + P_1^2$ and internal weight $h_P = \frac{1}{4}(b+b^{-1})^2 + P^2$, evaluated at a value of the parameter $q = \mathrm{e}^{2\pi i\tau}$ where $\tau$ is the modulus of the torus. The contour of integration $\mathcal{C}$ over the intermediate states with Liouville momentum $\widehat{P}$ in the $\hat{c} \leq 1$ torus one-point function (7.2b) is chosen as depicted in figure 5.

The torus one-point Virasoro conformal block $\mathcal{F}_{1,1}^{(b)}(P_1; P|q)$ can be expressed as [121, 122]

$$\mathcal{F}_{1,1}^{(b)}(P_1; P|q) = q^{P^2 - \frac{1}{24}} \left( \prod_{m=1}^\infty \frac{1}{1 - q^m} \right) \mathcal{H}_{1,1}^{(b)}(P_1; P|q),\qquad(7.3)$$

where the so-called elliptic conformal block $\mathcal{H}_{1,1}^{(b)}(P_1; P|q)$ admits a power series expansion in $q$ that starts at 1 and that can be computed efficiently with a recursion relation in the internal weight $h_P$, as briefly reviewed in appendix C.2. Decomposing the Liouville one-point functions in (7.1) into Virasoro conformal blocks and making use of (7.3) we obtain that the torus one-point diagram in Virasoro minimal string theory takes the form,

$$\mathsf{V}_{1,1}^{(b)}(P_1) = \mathsf{N}\, \frac{(2\pi)^2}{2} \int_{F_0} \mathrm{d}^2\tau \int_0^\infty \mathrm{d}P\, \rho_0^{(b)}(P) C_b(P_1, P, P) |q|^{2P^2} \mathcal{H}_{1,1}^{(b)}(P_1; P|q) \mathcal{H}_{1,1}^{(b)}(P_1; P|\overline{q})$$

$$\times \int_{\mathcal{C}} \mathrm{d}\widehat{P}\, \frac{(i\widehat{P})^2}{2\rho_0^{(b)}(i\widehat{P})} \widehat{C}_b(iP_1, \widehat{P}, \widehat{P}) |q|^{2\widehat{P}^2} \mathcal{H}_{1,1}^{(ib)}(iP_1; \widehat{P}|q) \mathcal{H}_{1,1}^{(ib)}(iP_1; \widehat{P}|\overline{q}).\qquad(7.4)$$

As discussed in section 3, an interesting feature of the Virasoro minimal string background (1.1) is that string diagrams in string perturbation theory are manifestly finite for any physical value of the external momenta of the closed strings. This is in contrast to more familiar string backgrounds in which divergences arise in degenerating limits of moduli space and the string diagram (for example, a string S-matrix element) is typically defined via analytic continuation from unphysical values of the external closed string momenta for which the string diagram moduli space integral converges [120].

**Analytic evaluation of $\mathsf{V}_{1,1}^{(b)}(P_1)$ for two special values of $P_1$.** There are a couple of cases in which the torus one-point Virasoro conformal block is known explicitly, for all values of the central charge. The most obvious is the case in which the external operator is the identity, with $P_1 = \frac{iQ}{2}$, in which case the conformal block is simply given by the corresponding non-degenerate Virasoro character with (internal) weight $P$,

$$\mathcal{F}_{1,1}^{(b)}\left(P_1 = \frac{iQ}{2}; P|\tau\right) = \chi_P^{(b)}(\tau) = \frac{\mathrm{e}^{2\pi i\tau P^2}}{\eta(\tau)}.\qquad(7.5)$$

The second case is less obvious. It turns out that when the external weight is equal to one, with $P_1 = \frac{i}{2}(b^{-1} - b) = \frac{i\widehat{Q}}{2}$, then the torus-one point block is also given by the non-degenerate Virasoro character [123]

$$\mathcal{F}_{1,1}^{(b)}\left(P_1 = \frac{i\widehat{Q}}{2}; P|\tau\right) = \chi_P^{(b)}(\tau).\qquad(7.6)$$

In other words, in both cases the elliptic conformal block (7.3) is precisely equal to one, $\mathcal{H}_{1,1}^{(b)}(P_1; P|q) = 1$ for $P_1 = \frac{iQ}{2}$ and $P_1 = \frac{i\widehat{Q}}{2}$. In both these cases, $P_1 \notin \mathbb{R}$. But these values still fall in the range of analyticity of $\mathsf{V}_{g,n}^{(b)}$ since the contour in the conformal block decomposition does not need to be deformed; see section 3.1.

For the case $P_1 = \frac{i\widehat{Q}}{2}$, using the following limit of the three-point coefficient

$$C_b(\tfrac{i\widehat{Q}}{2}, P, P) = \frac{2P^2}{\pi Q \rho_0(P)}, \tag{7.7}$$

as well as (3.6), we obtain that the torus one-point diagram (7.4) evaluates to,

$$\begin{aligned}
\mathsf{V}_{1,1}^{(b)}(P_1 = \tfrac{i\widehat{Q}}{2}) &= \mathsf{N} \frac{(2\pi)^2}{2} \int_{F_0} \mathrm{d}^2\tau \int_0^\infty \mathrm{d}P \, \rho_0^{(b)}(P) C_b(\tfrac{i\widehat{Q}}{2}, P, P) \, e^{-4\pi\tau_2 P^2} \\
&\quad \times \int_{\mathcal{C}} \mathrm{d}\widehat{P} \, \frac{(i\widehat{P})^2}{2\rho_0^{(b)}(i\widehat{P})} \widehat{C}_b(\tfrac{\widehat{Q}}{2}, \widehat{P}, \widehat{P}) \, e^{-4\pi\tau_2 \widehat{P}^2} \\
&= \mathsf{N} \frac{(2\pi)^2}{2} \int_{F_0} \mathrm{d}^2\tau \int_0^\infty \mathrm{d}P \, P^2 \, e^{-4\pi\tau_2 P^2} \int_{-\infty}^\infty \frac{\mathrm{d}\widehat{P}}{2} \, e^{-4\pi\tau_2 \widehat{P}^2} \\
&= \mathsf{N} \frac{(2\pi)^2}{2} \frac{1}{128\pi} \int_{F_0} \mathrm{d}^2\tau \, \tau_2^{-2} = \mathsf{N} \frac{\pi^2}{192}. 
\end{aligned} \tag{7.8}$$

This precisely agrees with (2.19b) evaluated at $P_1 = \frac{i\widehat{Q}}{2}$, provided that

$$\mathsf{N} = \frac{4}{\pi^2}. \tag{7.9}$$

Therefore, making use of (3.22) we obtain that

$$C_{S^2} = \frac{\pi^6}{64}. \tag{7.10}$$

The torus one-point diagram in the case $P = \frac{iQ}{2}$ proceeds essentially identically, except that slightly more care is required in taking the limit. The issue is that the relevant structure constant diverges in this limit

$$C_b\left(i\left(\tfrac{Q}{2} - \varepsilon\right), P, P\right) = \frac{1}{\pi \rho_0^{(b)}(P)} \varepsilon^{-1} + O(\varepsilon^0). \tag{7.11}$$

For this reason the spacelike Liouville correlator diverges and the timelike Liouville correlator vanishes but the combination that appears on the worldsheet remains finite. We find that

$$\begin{aligned}
\mathsf{V}_{1,1}^{(b)}\left(P_1 = \tfrac{iQ}{2}\right) &= \mathsf{N} \frac{(2\pi)^2}{2} \int_{F_0} \mathrm{d}^2\tau \int_0^\infty \mathrm{d}P \, e^{-4\pi\tau_2 P^2} \int_{-\infty}^\infty \frac{\mathrm{d}\widehat{P}}{2} (-\widehat{P}^2) e^{-4\pi\tau_2 \widehat{P}^2} \\
&= -\mathsf{N} \frac{\pi^2}{192},
\end{aligned} \tag{7.12}$$

which also exactly agrees with (2.19b) evaluated at $P_1 = \frac{iQ}{2}$ provided (7.9) is satisfied.

**Direct numerical evaluation of $\mathsf{V}_{1,1}^{(b)}(P_1)$ for generic values of $P_1$.** Let us first be more explicit about the behavior of the torus one-point diagram (7.4) near the cusp $\tau_2 \to \infty$ of

the fundamental domain. In this limit, since to leading order at large $\tau_2$ the torus one-point elliptic conformal blocks $\mathcal{H}_{1,1}^{(b)}(P_1; P|q) \simeq 1$, the moduli integral of (7.4) behaves as

$$\int^\infty \mathrm{d}\tau_2 \int_0^\infty \mathrm{d}P \, \rho_0^{(b)}(P) C_b(P_1, P, P) \, e^{-4\pi\tau_2 P^2} \int_{\mathcal{C}} \mathrm{d}\widehat{P} \, \frac{(i\widehat{P})^2}{2\rho_0^{(b)}(i\widehat{P})} \, \widehat{C}_b(iP_1, \widehat{P}, \widehat{P}) \, e^{-4\pi\tau_2 \widehat{P}^2} \,. \quad (7.13)$$

In the limit $\tau_2 \to \infty$, the integrals over the intermediate Liouville momenta $P$ and $\widehat{P}$ are dominated by their values near $P = 0$ and $\widehat{P} = 0$. Using Laplace's method, we can approximate these integrals as an asymptotic expansion at large $\tau_2$ by

$$\int_0^\infty \mathrm{d}P \, \rho_0^{(b)}(P) C_b(P_1, P, P) \, e^{-4\pi\tau_2 P^2}$$
$$\sim \sum_{n \in 2\mathbb{Z}_{\geq 0}} \frac{2^{-2(n+1)}\pi^{-\frac{n}{2}}}{\Gamma(\frac{n}{2}+1)} \, \tau_2^{-\frac{n+1}{2}} \frac{\mathrm{d}^n}{\mathrm{d}P^n}\bigg|_{P=0} \rho_0^{(b)}(P) C_b(P_1, P, P) \,, \quad (7.14a)$$

$$\int_{\mathcal{C}} \mathrm{d}\widehat{P} \, \frac{(i\widehat{P})^2}{2\rho_0^{(b)}(i\widehat{P})} \widehat{C}_b(iP_1, \widehat{P}, \widehat{P}) \, e^{-4\pi\tau_2 \widehat{P}^2}$$
$$\sim \sum_{m \in 2\mathbb{Z}_{\geq 0}} \frac{2^{-2m-1}\pi^{-\frac{m}{2}}}{\Gamma(\frac{m}{2}+1)} \, \tau_2^{-\frac{m+1}{2}} \frac{\mathrm{d}^m}{\mathrm{d}\widehat{P}^m}\bigg|_{\widehat{P}=0} \frac{(i\widehat{P})^2}{2\rho_0^{(b)}(i\widehat{P})} \widehat{C}_b(iP_1, \widehat{P}, \widehat{P}) \,. \quad (7.14b)$$

For instance, the first nonzero terms in the asymptotic expansions are the $m = 0$ and $n = 2$ terms on the RHS of (7.14), from which we obtain that the moduli integral (7.13) behaves as

$$\frac{1}{128\pi} \int^\infty \mathrm{d}\tau_2 \, \tau_2^{-2} \,, \quad (7.15)$$

and is therefore convergent, as claimed in section 3.1.

In the direct numerical evaluation of (7.4), we will employ the strategy of [41]. We split the fundamental domain $F_0$ of the torus moduli space into two regions: (I) $\tau \in F_0$ with $\tau_2 \leq \tau_2^{\max}$, and (II) $\tau \in F_0$ with $\tau_2 \geq \tau_2^{\max}$, for a sufficiently large value of $\tau_2^{\max}$. In region (I), we first perform the integrals over the intermediate Liouville momenta $\widehat{P}$ and $P$ separately and for a fixed value of $\tau$. These two integrations are performed numerically with the elliptic conformal blocks $\mathcal{H}_{1,1}^{(ib)}(iP_1; \widehat{P}|q)$ and $\mathcal{H}_{1,1}^{(b)}(P_1; P|q)$ computed via the recursion relation (C.12) and truncated to order $q^8$. The integration over $\tau$ in region (I) is then performed numerically. In region (II), we may approximate the moduli integrand by the expressions in (7.13) and (7.14); the moduli integral can then be done analytically. We include a sufficient number of terms in the asymptotic expansions (7.14) such that the resulting moduli integral over region (II) is accurate to order $(\tau_2^{\max})^{-3}$.

For the numerical evaluation of the torus one-point diagram, we will consider values of the Liouville parameter $b$ such that $b^2$ is a rational number. As discussed in appendix C, for such values of $b$ the Liouville three-point coefficients (3.1) and (3.6) can be expressed in terms of the Barnes G-function and thus their numerical implementation is much faster, as opposed to resorting to the integral representation of the $\Gamma_b(x)$ function. For some rational values of $b^2$, the numerical calculation of the torus one-point elliptic Virasoro conformal blocks $\mathcal{H}_{1,1}^{(b)}(P_1; P|q)$ through the recursion relation (C.12) involves delicate cancellations. In order to avoid loss of precision, we compute the conformal blocks with a central charge corresponding to $b = (\frac{m}{n})^{\frac{1}{2}} + \delta$ and $\hat{b} = (\frac{m}{n})^{\frac{1}{2}} + \delta$, with $m, n \in \mathbb{Z}_{\geq 1}$, with the choice of small $\delta = 10^{-7}$ for the $c \geq 25$ and $\hat{c} \leq 1$ Liouville CFT sectors, respectively. Lastly, in the numerical calculation of (7.4) we parametrize the contour of integration $\mathcal{C}$ over the intermediate $\hat{c} \leq 1$ Liouville

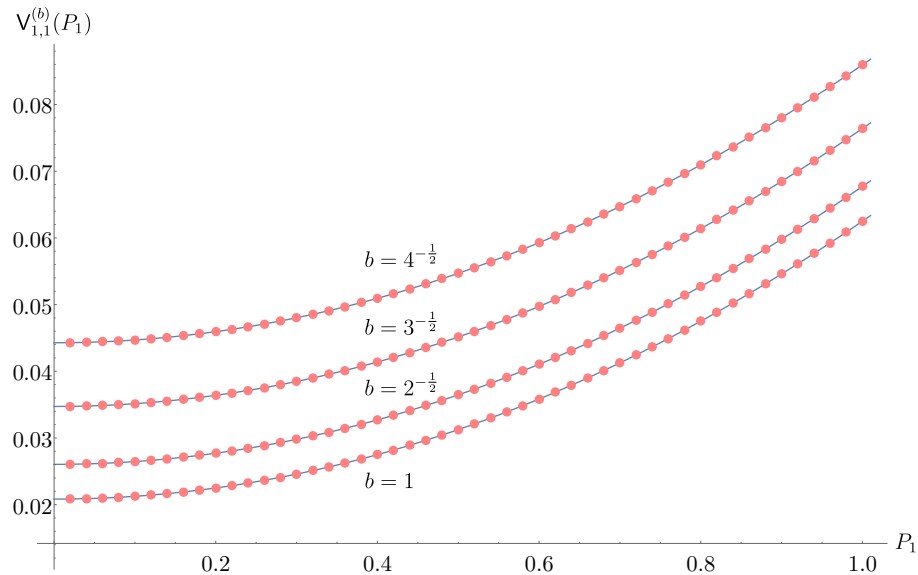

Figure 12: Shown in dots are the numerical results for the torus one-point string diagram (7.4) in Virasoro minimal string theory for a range of external momentum $P_1 \in [0,1]$ of the asymptotic closed string state, for the choice of the Liouville parameter $b = 1, 2^{-\frac{1}{2}}, 3^{-\frac{1}{2}}, 4^{-\frac{1}{2}}$ as labeled in the plot. The exact result (7.16) is shown in the solid curve.

momentum by $\widehat{P} = p + i\epsilon$ with $p \in \mathbb{R}$ and $\epsilon = 10^{-1}$, and set $\tau_2^{\max} = 15$ in the splitting of the fundamental domain $F_0$ described in the previous paragraph.

Figures 12 and 13 show numerical results for the torus one-point diagram (7.4) in Virasoro minimal string theory, with the fixed value (7.9) for the normalization constant N, computed with the strategy outlined above. Figure 12 shows results for the torus one-point diagram as a function of the external closed string momenta in the range $P_1 \in [0,1]$, for the following four choices of the Liouville parameter $b = 1, 2^{-\frac{1}{2}}, 3^{-\frac{1}{2}}, 4^{-\frac{1}{2}}$.[17] Figure 13 shows results for the torus one-point diagram as a function of the spacelike Liouville CFT central charge in the range $c \in [25, 26]$, for three choices of external closed string momenta $P_1 = \frac{1}{3}, \frac{1}{2}, \frac{2}{3}$.

These numerical results exhibit a remarkable level of agreement with the exact result (2.19b)

$$\mathsf{V}_{1,1}^{(b)}(P_1) = \frac{1}{24}\left(\frac{c-13}{24} + P_1^2\right),\tag{7.16}$$

and provide a highly nontrivial direct check of the duality. The largest discrepancy between the numerical results shown in figure 12 and the exact result (7.16) is of order $10^{-4}$ % for $b = 1, 2^{-\frac{1}{2}}$ and $10^{-3}$ % for $b = 3^{-\frac{1}{2}}, 4^{-\frac{1}{2}}$. Likewise, the largest discrepancy between the numerical results in figure 13 and the function (7.16) is of order $10^{-4}$ %.

## 7.2 Sphere four-point diagram

Next, we consider the four-punctured sphere diagram in Virasoro minimal string theory. After using its conformal Killing group to fix the positions of three vertex operators $\mathcal{V}_j(z_j, \overline{z}_j)$ with $j = 1, 3, 4$ to $z_1 = 0, z_3 = 1$, and $z_4 = \infty$, the sphere four-point diagram has one remaining

---

[17]The numerical results for $b = 1$ agree with those of [41], which followed a different normalization convention for the $c = 25$ and $c = 1$ Liouville CFT three-point coefficients.

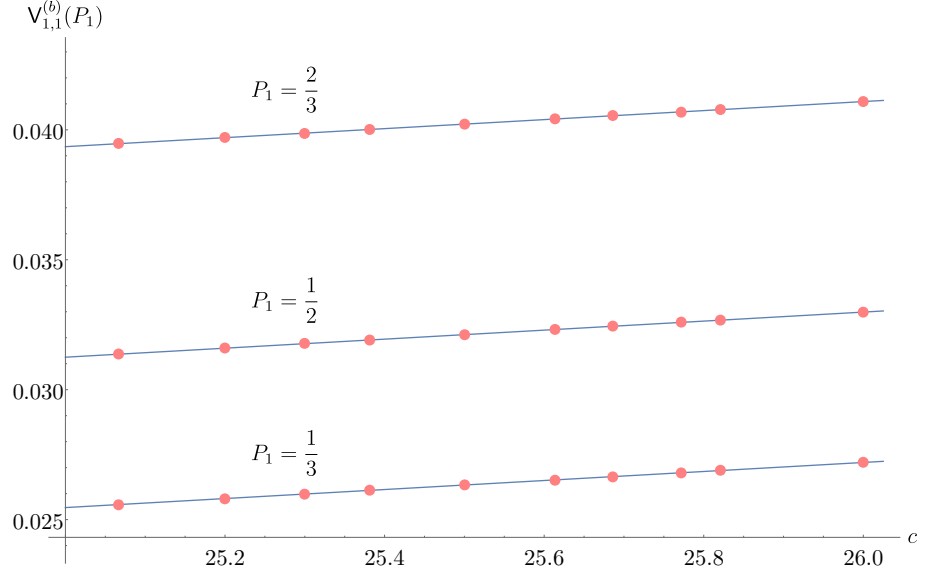

Figure 13: Shown in dots are the numerical results for the torus one-point string diagram (7.4) in Virasoro minimal string theory for a fixed value of external momentum $P_1 = \frac{1}{3}, \frac{1}{2}, \frac{2}{3}$ of the asymptotic closed string state, as labeled in each curve, and for varying central charge $c \in [25, 26]$. Specifically, the data points calculated numerically correspond to $b^2 = \frac{9}{10}, \frac{5}{6}, \frac{4}{5}, \frac{7}{9}, \frac{3}{4}, \frac{8}{11}, \frac{5}{7}, \frac{7}{10}, \frac{9}{13}, \frac{2}{3}$ for each value of $P_1$. The exact result (7.16) is shown in the solid curve.

modulus, the position $z \in \mathbb{C}$ of the last vertex operator $\mathcal{V}_2(z, \bar{z})$, and takes the form

$$\mathsf{V}_{0,4}^{(b)}(P_1, P_2, P_3, P_4) = C_{\mathrm{S}^2} \mathrm{N}^4 \int_{\mathbb{C}} \mathrm{d}^2 z \left\langle V_{P_1}(0) V_{P_2}(z, \bar{z}) V_{P_3}(1) V_{P_4}(\infty) \right\rangle_{g=0}$$
$$\times \left\langle \widehat{V}_{iP_1}(0) \widehat{V}_{iP_2}(z, \bar{z}) \widehat{V}_{iP_3}(1) \widehat{V}_{iP_4}(\infty) \right\rangle_{g=0}. \quad (7.17)$$

The Liouville CFT sphere four-point functions in (7.17) admit the following Virasoro conformal block decompositions,

$$\left\langle V_{P_1}(0) V_{P_2}(z, \bar{z}) V_{P_3}(1) V_{P_4}(\infty) \right\rangle_{g=0} = \int_0^\infty \mathrm{d}P\, \rho_0^{(b)}(P) C_b(P_1, P_2, P) C_b(P_3, P_4, P)$$
$$\times \mathcal{F}_{0,4}^{(b)}(P_1, P_2, P_3, P_4; P|z) \mathcal{F}_{0,4}^{(b)}(P_1, P_2, P_3, P_4; P|\bar{z}), \quad (7.18\mathrm{a})$$

$$\left\langle \widehat{V}_{iP_1}(0) \widehat{V}_{iP_2}(z, \bar{z}) \widehat{V}_{iP_3}(1) \widehat{V}_{iP_4}(\infty) \right\rangle_{g=0} = \int_{\mathcal{C}} \mathrm{d}\widehat{P}\, \frac{(i\widehat{P})^2}{2\rho_0^{(b)}(i\widehat{P})} \widehat{C}_b(iP_1, iP_2, \widehat{P}) \widehat{C}_b(iP_3, iP_4, \widehat{P})$$
$$\times \mathcal{F}_{0,4}^{(ib)}(iP_1, iP_2, iP_3, iP_4; \widehat{P}|z) \mathcal{F}_{0,4}^{(ib)}(iP_1, iP_2, iP_3, iP_4; \widehat{P}|\bar{z}), \quad (7.18\mathrm{b})$$

where $\mathcal{F}_{0,4}^{(b)}(P_1, P_2, P_3, P_4; P|z)$ is the sphere four-point holomorphic Virasoro conformal block with external weights $h_{P_i} = \frac{Q^2}{4} + P_i^2$ for $i = 1, \dots, 4$, intermediate weight $h_P = \frac{Q^2}{4} + P^2$, evaluated at the cross-ratio $z$. Further, the conformal block $\mathcal{F}_{0,4}^{(b)}(P_1, P_2, P_3, P_4; P|z)$ can be expressed in terms of an elliptic conformal block $\mathcal{H}_{0,4}^{(b)}(P_1, P_2, P_3, P_4; P|q)$ as [124]

$$\mathcal{F}_{0,4}^{(b)}(P_1, P_2, P_3, P_4; P|z) = (16q)^{P^2} z^{-\frac{Q^2}{4} - P_1^2 - P_2^2} (1-z)^{-\frac{Q^2}{4} - P_2^2 - P_3^2} \theta_3(q)^{-Q^2 - 4(P_1^2 + P_2^2 + P_3^2 + P_4^2)}$$
$$\times \mathcal{H}_{0,4}^{(b)}(P_1, P_2, P_3, P_4; P|q), \quad (7.19)$$

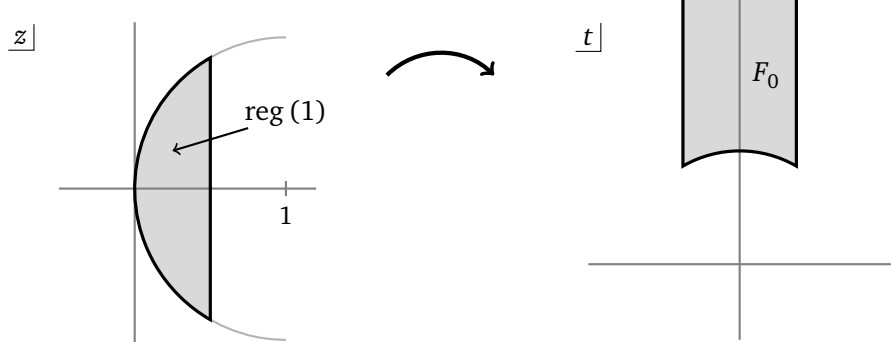

Figure 14: The fundamental domain in the cross ratio $z$-plane of the sphere four-point diagram, region (1) $= \{z \in \mathbb{C} \,|\, \mathrm{Re}\, z \leq \frac{1}{2}, |1-z| \leq 1\}$, is mapped to the fundamental domain $F_0 = \{t \in \mathbb{C} \,|\, -\frac{1}{2} \leq \mathrm{Re}\, t \leq \frac{1}{2}, |t| \geq 1\}$ in the complex $t$-plane via the change of variables (7.22).

where $\theta_3(q)$ is a Jacobi theta function, and the elliptic nome $q$ is related to the cross-ratio $z$ by

$$q(z) = \exp\left(-\pi \frac{K(1-z)}{K(z)}\right), \quad \text{where} \quad K(z) = {}_2F_1\left(\tfrac{1}{2}, \tfrac{1}{2}; 1 | z\right). \tag{7.20}$$

The elliptic conformal block $\mathcal{H}_{0,4}^{(b)}(P_1, P_2, P_3, P_4; P|q)$ admits a power series expansion in $q$ that can be efficiently computed via Zamolodchikov's recursion relation, as reviewed in appendix C.2. Whereas the conformal block expansion in the cross ratio $z$ a priori converges only in the unit $z$-disk ($|z| < 1$), the expansion in the elliptic nome $q$ variable converges everywhere inside the unit $q$-disk, which in particular covers the entire complex $z$-plane [125]. Furthermore, at any given point in the $z$-plane, the conformal block expansion in the $q$ variable converges much faster.

The crossing symmetry relations of the $\hat{c} \leq 1$ and $c \geq 25$ Liouville CFT sphere four-point correlation functions (7.18), generated by (C.15) and (C.16), may be used to reduce the moduli integration of the four-point diagram (7.17) over the complex $z$-plane into a finite domain near $z = 0$ [126, 127]. We divide the complex $z$-plane into six regions: (1) $\mathrm{Re}\, z \leq \frac{1}{2}$, $|1-z| \leq 1$, (2) $|z| \leq 1$, $|1-z| \geq 1$, (3) $\mathrm{Re}\, z \leq \frac{1}{2}$, $|z| \geq 1$, (4) $\mathrm{Re}\, z \geq \frac{1}{2}$, $|z| \leq 1$, (5) $|1-z| \leq 1$, $|z| \geq 1$, and (6) $\mathrm{Re}\, z \geq \frac{1}{2}$, $|1-z| \geq 1$. Denoting the transformation $z \to 1-z$, for which (C.15) holds, by $T$ and the transformation $z \to z^{-1}$, for which (C.16) holds, by $S$, the regions (2)–(6) can be mapped to region (1) by the transformations $STS, TS, T, ST, S$, respectively. Hence, the four-point string diagram (7.17) can be written as

$$\mathrm{V}_{0,4}^{(b)}(P_1, P_2, P_3, P_4) = C_{\mathrm{S}^2} \mathrm{N}^4 \int\limits_{\mathrm{reg}(1)} \mathrm{d}^2 z \left[ \left\langle \widehat{V}_{iP_1}(0) \widehat{V}_{iP_2}(z, \bar{z}) \widehat{V}_{iP_3}(1) \widehat{V}_{iP_4}(\infty) \right\rangle_{g=0} \right.$$

$$\times \left\langle V_{P_1}(0) V_{P_2}(z, \bar{z}) V_{P_3}(1) V_{P_4}(\infty) \right\rangle_{g=0}$$

$$\left. + \left( 5 \text{ other perms of } \{123\} \right) \right]. \tag{7.21}$$

Lastly, performing a change of variable defined by

$$t = i \frac{K(1-z)}{K(z)}, \tag{7.22}$$

from the cross-ratio $z$ to the complex $t$-plane, such that the elliptic nome is $q = e^{i\pi t}$, region (1) of the complex $z$-plane is mapped to the fundamental domain $F_0 = \{t \in \mathbb{C} \,|\, -\frac{1}{2} \leq \mathrm{Re}\, t \leq \frac{1}{2}, |t| \geq 1\}$

in the complex $t$-plane. Decomposing the Liouville CFT four-point functions in (7.17) into Virasoro conformal blocks, making use of (7.19), performing the change of variables (7.22),[18] and plugging in the constant values (7.9) and (7.10), we obtain that the four-point string diagram in Virasoro minimal string theory can be written as

$$
\begin{aligned}
\mathsf{V}_{0,4}^{(b)}(P_1, P_2, P_3, P_4) = 4 \int_{F_0} \mathrm{d}^2 t \bigg[ & \int_0^\infty \mathrm{d}P\, \rho_0^{(b)}(P) C_b(P_1, P_2, P) C_b(P_3, P_4, P) \\
& \times |16q|^{2P^2} \mathcal{H}_{0,4}^{(b)}(P_1, P_2, P_3, P_4; P|q) \mathcal{H}_{0,4}^{(b)}(P_1, P_2, P_3, P_4; P|\bar{q}) \\
& \times \int_{\mathcal{C}} \mathrm{d}\widehat{P}\, \frac{(i\widehat{P})^2}{2\rho_0^{(b)}(i\widehat{P})} \widehat{C}_b(iP_1, iP_2, \widehat{P}) \widehat{C}_b(iP_3, iP_4, \widehat{P}) \\
& \times |16q|^{2\widehat{P}^2} \mathcal{H}_{0,4}^{(ib)}(iP_1, iP_2, iP_3, iP_4; \widehat{P}|q) \mathcal{H}_{0,4}^{(ib)}(iP_1, iP_2, iP_3, iP_4; \widehat{P}|\bar{q}) \\
& + \Big( 5 \text{ other perms of } \{123\} \Big) \bigg].
\end{aligned}
\tag{7.24}
$$

As was the case for the torus one-point diagram considered in the previous section, the sphere four-point diagram takes the slightly simpler form (7.24) when expressed in terms of the elliptic Virasoro conformal blocks.

**Analytic evaluation of $\mathsf{V}_{0,4}^{(b=1)}(P_1, P_2, P_3, P_4)$ for special values of $P_i$ and $b$.** Unlike the case of the torus one-point diagram, we are not aware of any value of the conformal weights for which we can compute both the timelike and spacelike Liouville CFT four-point functions exactly for any value of the central charge. However, for the special case of $c = 25$, or $b = 1$, with external weights all equal to $h_i = \frac{15}{16}$, as well as for the case of $c = 1$, or $b = i$, with external weights all equal to $\hat{h}_i = \frac{1}{16}$, the elliptic sphere four-point blocks (7.19) are known to be given simply by [128]

$$
\mathcal{H}_{0,4}^{(b=1)}\left(\tfrac{i}{4}, \tfrac{i}{4}, \tfrac{i}{4}, \tfrac{i}{4}; P|q\right) = 1, \qquad \mathcal{H}_{0,4}^{(b=i)}\left(\tfrac{1}{4}, \tfrac{1}{4}, \tfrac{1}{4}, \tfrac{1}{4}; \widehat{P}|q\right) = 1,
\tag{7.25}
$$

respectively. For this special case, and making use of

$$
C_{b=1}\left(\tfrac{i}{4}, \tfrac{i}{4}, P\right) = \frac{2^{-\frac{11}{2} - 4P^2} P}{\sinh(2\pi P)},
\tag{7.26}
$$

we obtain that the sphere four-point diagram (7.24) evaluates to

$$
\begin{aligned}
\mathsf{V}_{0,4}^{(b=1)}\left(\tfrac{i}{4}, \tfrac{i}{4}, \tfrac{i}{4}, \tfrac{i}{4}\right) &= 6 \times 4 \int_{F_0} \mathrm{d}^2 t \int_0^\infty \mathrm{d}P\, \rho_0^{(b)}(P) C_1\left(\tfrac{i}{4}, \tfrac{i}{4}, P\right)^2 2^{8P^2} e^{-2\pi t_2 P^2} \\
& \quad \times \int_{\mathcal{C}} \mathrm{d}\widehat{P}\, \frac{(i\widehat{P})^2}{2\rho_0^{(b)}(i\widehat{P})} \widehat{C}_1\left(\tfrac{i}{4}, \tfrac{i}{4}, \widehat{P}\right)^2 2^{8\widehat{P}^2} e^{-2\pi t_2 \widehat{P}^2} \\
&= 24 \int_{F_0} \mathrm{d}^2 t \int_0^\infty \mathrm{d}P\, P^2 e^{-2\pi t_2 P^2} \int_{-\infty}^\infty \frac{\mathrm{d}\widehat{P}}{2} e^{-2\pi t_2 \widehat{P}^2} \\
&= \frac{1}{4},
\end{aligned}
\tag{7.27}
$$

which exactly agrees with (2.19a) evaluated at $c = 25$ with $P_i = \frac{i}{4}$.

---

[18]The Jacobian of the map from the cross-ratio $z$ to the elliptic nome $q = e^{i\pi t}$

$$
\left| \frac{\mathrm{d}z}{\mathrm{d}t} \right|^2 = \left| \pi i \left( \frac{\theta_2(q)\theta_4(q)}{\theta_3(q)} \right)^4 \right|^2,
\tag{7.23}
$$

exactly cancels the combined prefactors appearing in the product of the conformal blocks (7.19).

**Direct numerical evaluation of $V_{0,4}^{(b)}(P_1, P_2, P_3, P_4)$ for generic values of $P_i$ and $b$.** The behavior of each of the six terms in (7.24) near the cusp $t_2 \to \infty$ of the fundamental domain $F_0$ in the complex $t$-plane, with $t = t_1 + it_2$, can be analyzed similarly to the case of the torus one-point diagram considered in the previous section. In the limit $t_2 \to \infty$, the sphere four-point elliptic conformal blocks $\mathcal{H}_{0,4}^{(b)}(P_i; P|q) \simeq 1$ and using Laplace's method we can approximate the $\hat{c} \le 1$ and $c \ge 25$ Liouville correlation functions as an asymptotic expansion at large $t_2$ by

$$
\int_0^\infty \mathrm{d}P \, \rho_0^{(b)}(P) C_b(P_1, P_2, P) C_b(P_3, P_4, P) \, \mathrm{e}^{-(2\pi t_2 - 8 \log 2)P^2}
$$

$$
\sim \sum_{n \in 2\mathbb{Z}_{\ge 0}} \frac{2^{-(n+1)} \pi^{\frac{1}{2}} (2\pi t_2 - 8 \log 2)^{-\frac{n+1}{2}}}{\Gamma(\frac{n}{2} + 1)} \frac{\mathrm{d}^n}{\mathrm{d}P^n} \bigg|_{P=0} \rho_0^{(b)}(P) C(P_1, P_2, P) C(P_3, P_4, P),
$$

$$(7.28a)$$

$$
\int_{\mathcal{C}} \mathrm{d}\widehat{P} \, \frac{(i\widehat{P})^2}{2\rho_0^{(b)}(i\widehat{P})} \widehat{C}_b(iP_1, iP_2, \widehat{P}) \widehat{C}_b(iP_3, iP_4, \widehat{P}) \, \mathrm{e}^{-(2\pi t_2 - 8 \log 2)\widehat{P}^2}
$$

$$
\sim \sum_{m \in 2\mathbb{Z}_{\ge 0}} \frac{2^{-n} \pi^{\frac{1}{2}} (2\pi t_2 - 8 \log 2)^{-\frac{n+1}{2}}}{\Gamma(\frac{n}{2} + 1)} \frac{\mathrm{d}^m}{\mathrm{d}\widehat{P}^m} \bigg|_{\widehat{P}=0} \frac{(i\widehat{P})^2 \widehat{C}_b(iP_1, iP_2, \widehat{P}) \widehat{C}_b(iP_3, iP_4, \widehat{P})}{2\rho_0^{(b)}(i\widehat{P})},
$$

$$(7.28b)$$

and similarly for the other five terms in (7.24). For example, taking the first nonzero terms in the asymptotic expansions (7.28) we obtain that the full moduli integral in the sphere four-point string diagram (7.24) behaves as

$$
6 \times \frac{1}{32\pi} \int^\infty \mathrm{d}t_2 \, t_2^{-2},
$$

$$(7.29)$$

and is therefore convergent, as discussed in section 3.1.

With the four-point sphere diagram written in the form (7.24), we can then follow precisely the same strategy of numerical integration that we employed in the computation of the torus one-point string diagram described in the previous section.[19] We split the fundamental domain $F_0$ in the complex $t$-plane into two regions: (I) $t \in F_0$ with $t_2 \le t_2^{\max}$, where we first perform the integrals over the intermediate Liouville momenta $P$ and $\widehat{P}$, and then over the modulus $t$ numerically, and (II) $t \in F_0$ with $t_2 \ge t_2^{\max}$, where we use the asymptotic expansions of the form (7.28) and perform the moduli integral over $t$ analytically, including a sufficient number of terms in the asymptotic expansions such that the resulting integral is accurate to order $(t_2^{\max})^{-4}$. In the direct numerical evaluation of (7.24), we compute the elliptic conformal blocks $\mathcal{H}_{0,4}^{(b)}(P_i; P|q)$ via the recursion relation (C.10) with a central charge corresponding to $b = (\frac{m}{n})^{\frac{1}{2}} + \delta$ and $\hat{b} = (\frac{m}{n})^{\frac{1}{2}} + \delta$ with $m, n \in \mathbb{Z}_{\ge 1}$ and the choice of small $\delta = 10^{-6}$ for the $c \ge 25$ and $\hat{c} \le 1$ Liouville CFT sectors, respectively, both truncated to order $q^8$; parametrize the contour of integration $\mathcal{C}$ over the intermediate $\hat{c} \le 1$ Liouville momentum by $\widehat{P} = p + i\varepsilon$ with $p \in \mathbb{R}$ and $\varepsilon = 10^{-1}$; and set $t_2^{\max} = 15$.

---

[19]In [40], the moduli integral of the sphere four-point diagram was numerically computed directly in the cross-ratio variable $z \in$ region I, which led to less precise results compared to the computations performed in this paper. More importantly, [40] followed a different strategy in which the order of integrations is switched – first integrate over the cross-ratio $z$ and then over the intermediate Liouville momenta $P$ and $\widehat{P}$; this order proved to be more convenient in the numerical evaluation of string scattering amplitudes in two-dimensional string theory of [127, 129, 130]. With that order of integrations, it was necessary to introduce regulator counterterms to the moduli integral (7.21), which appears to have led to a systematic error in the numerical results for the sphere four-point diagram $V_{0,4}^{(b)}$. In the notation of equation (3.11) of [40], the results of the present paper are $\alpha = 8$ and $\beta = 16$.

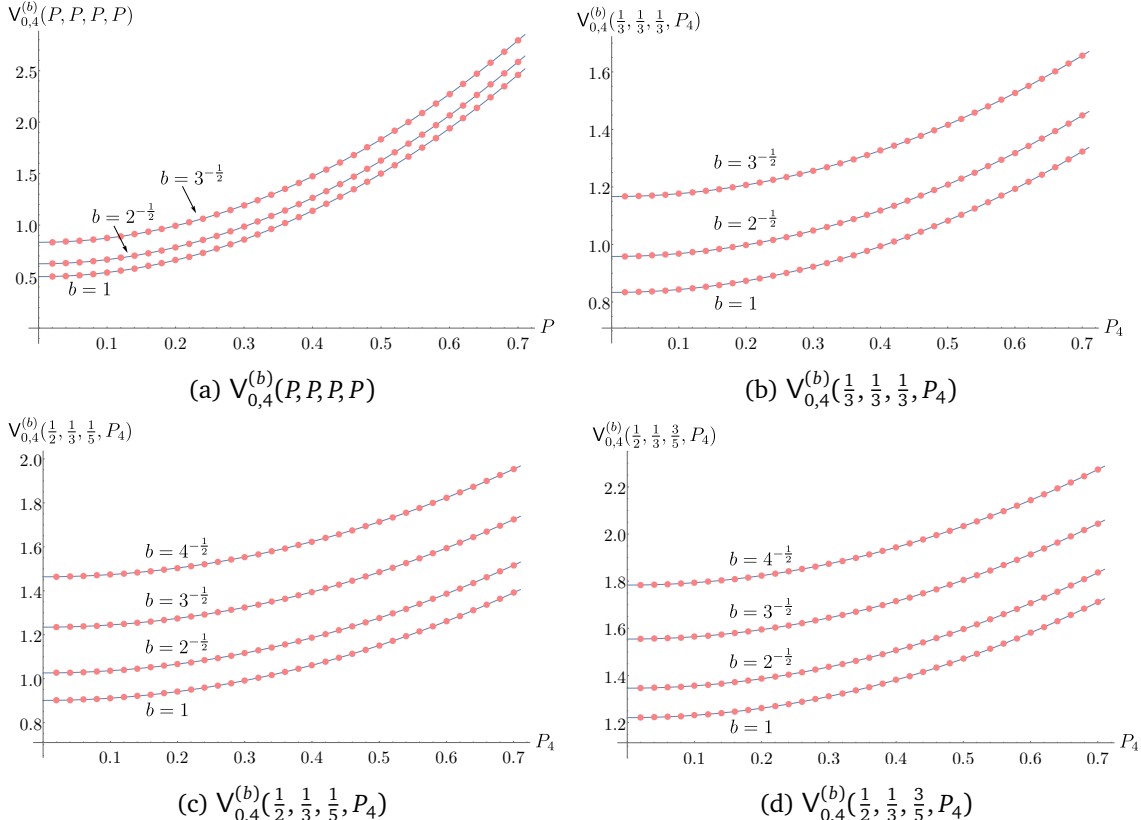

Figure 15: Shown in dots are the numerical results for the four-point string diagram (7.24) in Virasoro minimal string theory with the choices (7.30) for the external momenta of the asymptotic closed string states. The exact result (7.31) is shown in the solid curve.

For the direct numerical evaluation of the four-point string diagram (7.24) we will make the following choices for the external momenta of the asymptotic closed string states and for the Liouville parameter $b$ of the $c \geq 25$ Liouville CFT sector of the Virasoro minimal string background (1.1):

$$(i) \qquad P_1 = P_2 = P_3 = P_4 \equiv P, \qquad P \in [0, 0.7], \quad \text{for } b = 1, 2^{-\frac{1}{2}}, 3^{-\frac{1}{2}}, \tag{7.30a}$$

$$(ii) \qquad P_1 = P_2 = P_3 = \frac{1}{3}, \qquad P_4 \in [0, 0.7], \quad \text{for } b = 1, 2^{-\frac{1}{2}}, 3^{-\frac{1}{2}}, \tag{7.30b}$$

$$(iii) \qquad P_1 = \frac{1}{3}, P_2 = \frac{1}{2}, P_3 = \frac{1}{5}, \qquad P_4 \in [0, 0.7], \quad \text{for } b = 1, 2^{-\frac{1}{2}}, 3^{-\frac{1}{2}}, 4^{-\frac{1}{2}}, \tag{7.30c}$$

$$(iv) \qquad P_1 = \frac{1}{3}, P_2 = \frac{1}{2}, P_3 = \frac{3}{5}, \qquad P_4 \in [0, 0.7], \quad \text{for } b = 1, 2^{-\frac{1}{2}}, 3^{-\frac{1}{2}}, 4^{-\frac{1}{2}}. \tag{7.30d}$$

The numerical results for the four-point sphere string diagram (7.24) for the choice of external closed string momenta (7.30), computed with the strategy outlined above, are shown in figure 15. We again find that the numerical results demonstrate a remarkable agreement with the exact form for the string four-point diagram (2.19a),

$$V_{0,4}^{(b)}(P_1, P_2, P_3, P_4) = \frac{c - 13}{24} + P_1^2 + P_2^2 + P_3^2 + P_4^2. \tag{7.31}$$

This agreement provides again highly nontrivial evidence for our proposed duality. For the results presented in figure 15, the largest discrepancy between the numerical results in the

data sets (7.30) and the exact result (7.31) is of order $10^{-4}\,\%$ for $b = 1, 2^{-\frac{1}{2}}$ and $10^{-3}\,\%$ for $b = 3^{-\frac{1}{2}}, 4^{-\frac{1}{2}}$.

## 7.3 Sphere partition function and other exceptional cases

So far, we have discussed $\mathsf{V}_{g,n}^{(b)}$ for $2g - 2 + n \geq 0$, where the moduli space $\mathcal{M}_{g,n}$ in (2.7) is well-defined. However, one can also discuss the remaining exceptional cases, which we do now. Especially on the sphere, this is subtle, because the volume of the conformal Killing vector group is infinite for $n \leq 2$ and because of non-compactness of the worldsheet CFT the result is formally given by a ratio $\frac{\infty}{\infty}$. Our main tool is to assume that the dilaton (4.15a) and string equations (4.15b) continue to hold, which allows us to relate these lower-point functions to higher-point functions.

**Torus partition function.** Let us start with the torus partition function. The dilaton equation implies that the torus partition function diverges:

$$0 \cdot \mathsf{V}_{1,0}^{(b)} = \mathsf{V}_{1,1}^{(b)}\left(P = \tfrac{i\widehat{Q}}{2}\right) - \mathsf{V}_{1,1}^{(b)}\left(P = \tfrac{iQ}{2}\right) = \frac{1}{24} \neq 0 \,. \tag{7.32}$$

Since the right-hand-side is non-zero, this implies that the torus partition function is infinite. This can also be checked directly from the worldsheet and is a reflection of the fact that the torus partition function of Liouville theory diverges.

**Sphere two-point function.** The sphere two-point function needs to satisfy the dilaton equation, but this does not give any non-trivial information. Instead, we observe from the worldsheet definition (2.7) that the two-point functions on the worldsheet are only non-vanishing for $P_1 = P_2$ and thus we necessarily have[20]

$$\mathsf{V}_{0,2}^{(b)}(P_1, P_2) = F(P_1)\delta(P_1 - P_2) \,. \tag{7.33}$$

We can fix $F(P_1)$ by looking at the string equation (4.15b)

$$1 = \sum_{j=1}^{2} \int_0^{P_j} 2P_j \, \mathrm{d}P_j \, \mathsf{V}_{0,2}^{(b)}(P_1, P_2) \,, \tag{7.34}$$

which fixes

$$\mathsf{V}_{0,2}^{(b)}(P_1, P_2) = \frac{1}{2P_1}\,\delta(P_1 - P_2) = \delta(h_1 - h_2) \,, \tag{7.35}$$

where we expressed it in terms of the conformal weight in the last step. This could have been expected from the spacetime picture, since we can obtain a double-trumpet either by gluing two trumpets or by using eq. (2.18) for $g = 0$ and $n = 2$. Thus the two-point volume should be just a delta-function in the natural measure $2P\,\mathrm{d}P$. We also would have concluded this from the inverse Laplace transform of the resolvent $R_{0,2}^{(b)}$ (5.9).

---

[20] The worldsheet two-point function is actually proportional to $\delta(P_1 - P_2)^2$, since we get a delta-function from both spacelike and timelike Liouville theory. The square in the delta-function can then get cancelled by the infinite volume of the conformal Killing vector group [131].

**Sphere one-point function.** The one-point function on the sphere can be obtained directly from (7.35) via the dilaton equation (4.15a). We have

$$V_{0,1}^{(b)}(P) = V_{0,2}^{(b)}\left(P, \tfrac{iQ}{2}\right) - V_{0,2}^{(b)}\left(P, \tfrac{i\widehat{Q}}{2}\right) = \delta(h) - \delta(h-1). \tag{7.36}$$

This could again be expected from the disk partition function, since gluing a trumpet to this object according to (2.18) gives back the disk partition function (2.16a). In particular, for states in the spectrum for which $P > 0$, the one-point function on the sphere vanishes. Vanishing of the generic sphere one-point function was anticipated in [41] based on the well-behavedness of the string perturbation expansion.

**Sphere partition function.** Finally, the zero-point function on the sphere follows again from the dilaton equation:

$$V_{0,0}^{(b)} = \frac{1}{2}\left(V_{0,1}^{(b)}\left(\tfrac{iQ}{2}\right) - V_{0,1}^{(b)}\left(\tfrac{i\widehat{Q}}{2}\right)\right) = \frac{1}{2}\left(\delta(0) + \delta(0)\right) = \infty. \tag{7.37}$$

Like the torus partition function, also the sphere partition is divergent. This feature is also believed to be a property of JT gravity [132, 133].

# 8 Asymptotic boundaries and ZZ-instantons

In this section we elucidate the worldsheet boundary conditions needed to describe configurations with asymptotic boundaries in Virasoro minimal string theory. We will see that this involves pairing a non-standard basis of FZZT branes for spacelike Liouville CFT described in section 3.2 together with ZZ-like boundary conditions (a good choice turns out to be the "half-ZZ" branes introduced in section 3.2) for timelike Liouville CFT. Equipped with these boundary conditions, we will then derive the disk and trumpet partition functions (given in equations (2.16a) and (2.16b) respectively), as well as the double-trumpet partition function directly from the worldsheet BCFT. We then proceed to investigate non-perturbative effects mediated by ZZ-instantons on the worldsheet. In particular, we determine the normalization of the one-instanton contributions to the free energy, finding a perfect match with the matrix integral as computed in section 6.1. Finally, we compute the leading non-perturbative corrections to the quantum volumes as mediated by ZZ-instantons.

## 8.1 Asymptotic boundaries

We now discuss the incorporation of asymptotically Euclidean AdS boundaries to Virasoro minimal string theory through conformal boundary conditions for the worldsheet CFT. The quantum volumes $V_{g,n}^{(b)}(P_1, \ldots, P_n)$ computed by closed string perturbation theory as in (2.7) correspond to configurations with $n$ geodesic boundaries with lengths that are given in the JT limit ($b \to 0$) by [134]

$$\ell_i = 4\pi b P_i. \tag{8.1}$$

In order to introduce asymptotic boundaries, we glue "trumpets" — punctured disks with boundary conditions to be described shortly — onto the string diagrams with finite boundaries as described in section 2.5. The punctures are labelled by a Liouville momentum $P_i$ and create finite boundaries (which are to be glued onto those of the quantum volumes), with lengths that in the JT limit are given by (8.1). Then what we seek is a boundary condition for the worldsheet CFT corresponding to an asymptotic boundary with fixed (renormalized) length $\beta_i$.

As reviewed in section 3.2, Liouville CFT admits two main families of conformal boundary conditions. In order to develop some intuition for them and their interpretation in Virasoro

minimal string theory, recall that the Virasoro minimal string admits a reformulation in terms of two-dimensional dilaton gravity defined in (2.1), where the dilaton and Weyl factor of the target space metric can be recast in terms of the spacelike and timelike Liouville fields $\phi$ and $\chi$ as in (2.2). The one-parameter family of FZZT branes [84, 85] admit a semiclassical reformulation in terms of a modified Neumann boundary condition for the Liouville fields, and hence may heuristically be thought of as extended branes. In contrast, the ZZ conformal boundary conditions [83] may semiclassically be thought of as Dirichlet boundary conditions for the Liouville field and hence represent localized branes. Indeed, as reviewed in section 3.2, the open-string spectrum of the cylinder partition functions with FZZT boundary conditions is continuous, while it is discrete for the ZZ-type boundary conditions.

Thus in order to introduce asymptotic boundaries in Virasoro minimal string theory, we will need to equip the spacelike and timelike Liouville sectors of the worldsheet CFT with a suitable combination of FZZT and ZZ-type boundary conditions. In particular, we claim that an ansatz that correctly reproduces matrix integral results is to equip the spacelike Liouville theory with FZZT boundary conditions and the timelike Liouville theory with the "half-ZZ" boundary conditions introduced in section 3.2.

Let us first discuss the FZZT boundary conditions for spacelike Liouville theory. Recall that the FZZT branes are labeled by a continuous parameter $s$. We claim that fixing the renormalized length of the asymptotic boundary is achieved by working in a basis of FZZT boundary states that is Laplace-dual to the fixed-$s$ basis, as

$$
\int_0^\infty \mathrm{d}s\, e^{-\beta s^2} \, |\mathrm{FZZT}^{(b)}(s)\rangle \,. \tag{8.2}
$$

Heuristically, since $s$ labels the Liouville momentum of an open string stretched between FZZT and ZZ branes, we think of $s^2$ as an energy and the Laplace transform as implementing the change to an ensemble of fixed $\beta$.

Having fixed the renormalized boundary length with FZZT-like boundary conditions on the spacelike Liouville theory, fixing the asymptotic value of the dilaton as usual in dilaton gravity requires ZZ-like (Dirichlet) boundary conditions for the timelike Liouville theory. Indeed, any of the "half-ZZ" boundary conditions described in section 3.2 is sufficient, when paired with a suitable modification of the FZZT BCFT data for the spacelike Liouville CFT. Following previous literature on Liouville gravity (although our prescription varies significantly in the details, see e.g. [38]), we think of the resulting combined boundary condition as introducing a "marked" disk in Virasoro minimal string theory. The idea is that in string theory equipped with worldsheet boundaries one computes partition functions on *unmarked* disks, in the sense that translations along the boundary circle are gauged (there is no marked reference point). To undo the effect of the gauging, one should multiply by the volume of translations along the boundary. This is how we interpret the necessary modification of the FZZT boundary state to be described presently.

For example, suppose we equip the timelike Liouville theory with $(m, \pm)$ "half-ZZ" boundary conditions. Then we claim that the FZZT boundary conditions on the spacelike Liouville theory should be modified so that the disk one-point function is given by

$$
\Psi^{(b)}(s; P) \to \Psi^{(b)}_{(m,\pm)}(s; P) \equiv \frac{P \rho_0^{(b)}(P)}{\sqrt{2}\sinh(2\pi m b^{\pm 1} P)} \Psi^{(b)}(s; P), \tag{8.3}
$$

where the unmarked one-point function $\Psi^{(b)}$ is given in (3.32). This redefinition is independent of the FZZT brane parameter $s$ so the transformation to the fixed-length basis is unaffected.

To summarize, we claim that the worldsheet boundary conditions that introduce an asymptotic boundary of fixed renormalized length $\beta$ involve combining the Laplace transform of the

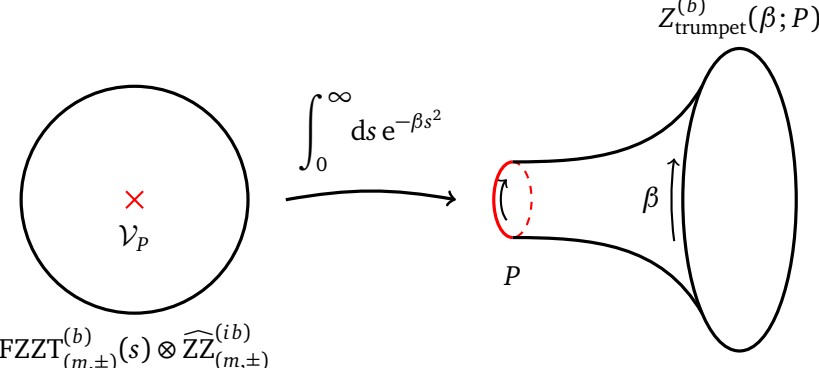

Figure 16: The Laplace transform of the (marked) disk one-point diagram of an on-shell vertex operator $\mathcal{V}_P$ subject to FZZT($s$) boundary conditions in the spacelike Liouville sector and half-ZZ boundary conditions in the timelike Liouville sector of the Virasoro minimal string theory computes the partition function of a "trumpet" with Liouville momentum $P$ and an asymptotic boundary of renormalized length $\beta$.

marked FZZT boundary conditions for spacelike Liouville CFT with the corresponding half-ZZ boundary conditions for timelike Liouville CFT:

$$\int_0^\infty ds\, e^{-\beta s^2} |\text{FZZT}^{(b)}_{(m,\pm)}(s)\rangle \otimes |\widehat{\widetilde{ZZ}}^{(ib)}_{(m,\pm)}\rangle\,, \tag{8.4}$$

where the subscript on the FZZT boundary state indicates the marking. In what follows we will see that all choices of $(m,\pm)$ are in a sense BRST-equivalent.

We note that both the transformation from the fixed-$s$ to the fixed-length basis (8.2) and the marking prescription (8.3) differ substantially from the conventions adopted in previous work on the minimal string. Nevertheless, we will see that the combined BCFTs define the correct conformal boundary conditions that match with the matrix integral.

In particular, the energy in the dual matrix model will be identified with $s^2$ instead of $\cosh(2\pi bs)$ as is the case e.g. in the minimal string. In those cases, this relation can be motivated from the path integral, but we do not have a sufficiently good understanding of the boundary conditions of timelike Liouville theory to perform such a derivation here. Instead, we remark that the identification of the energy with $s^2$ is uniquely fixed by requiring that the density of states computed from the disk partition function matches with the spectral curve given in eq. (5.15). We also remark that this identification is quite natural in this context given that from the definition of the FZZT parameter in (3.31) $s^2$ is the conformal weight in the open-string channel, which is the energy in the Virasoro algebra.

**Punctured disk diagram: The trumpet and the disk.** We start by computing the trumpet partition function in Virasoro minimal string theory directly from the worldsheet BCFT. The starting point for this computation is the punctured disk diagram, with FZZT boundary conditions on the spacelike Liouville sector and (say) $(m,\pm)$ half-ZZ boundary conditions on the timelike Liouville sector. Figure 16 summarizes the relationship between the punctured disk diagram and the trumpet partition function in Virasoro minimal string theory. Taking into account the prescription (8.3), the marked disk diagram is given by the following product of

disk one-point functions

$$
\begin{aligned}
Z_{\mathrm{disk}}^{(b)}(s;P) &= \widetilde{C}_{\mathrm{D}^2}\mathrm{N}\Psi_{(m,\pm)}^{(b)}(s;P)\widehat{\Psi}_{(m,\pm)}^{(ib)}(P)\\
&= \widetilde{C}_{\mathrm{D}^2}\mathrm{N}\,\frac{P\rho_0^{(b)}(P)}{\sqrt{2}\sinh(2\pi m b^{\pm 1}P)}\,\frac{2\sqrt{2}\cos(4\pi sP)}{\rho_0^{(b)}(P)}\,\frac{4\sinh(2\pi m b^{\pm 1}P)}{P}\\
&= 2\sqrt{2}\widetilde{C}_{\mathrm{D}^2}\mathrm{N}\times 2\sqrt{2}\cos(4\pi sP)\,,
\end{aligned}
\tag{8.5}
$$

where $\widehat{\Psi}_{(m,\pm)}^{(i\hat{b})}$ is given in (3.36) and we used (2.5) and (2.6). Here, $\widetilde{C}_{\mathrm{D}^2}$ is the normalization of the string theory path integral; the tilde indicates that it also includes the volume of the residual U(1) automorphism group of the punctured disk. Equation (8.5) is equivalent to the modular S matrix that decomposes a Virasoro character with Liouville momentum $s$ into a complete basis of characters in the dual channel with Liouville momenta $P$.

The trumpet partition function, with an asymptotic boundary of renormalized length $\beta$, is then given by the Laplace transform (8.2) of the marked disk one-point function (8.5):

$$
Z_{\mathrm{trumpet}}^{(b)}(\beta;P) = \int_0^\infty \mathrm{d}s\, e^{-\beta s^2} Z_{\mathrm{disk}}^{(b)}(s;P) = 2\sqrt{2}\widetilde{C}_{\mathrm{D}^2}\mathrm{N}\times\sqrt{\frac{2\pi}{\beta}}\,e^{-\frac{4\pi^2 P^2}{\beta}}\,.
\tag{8.6}
$$

As explained in sections 2.5 and 4.4, this should be nothing but the Virasoro character of a primary of conformal weight $h_P$ in the dual channel (with modulus $\tau = \frac{2\pi i}{\beta}$) with the contributions of the descendants stripped off. This fixes the normalization

$$
\widetilde{C}_{\mathrm{D}^2} = \frac{1}{2\sqrt{2}\mathrm{N}} = \frac{\pi^2}{8\sqrt{2}}\,,
\tag{8.7}
$$

where we used that N is given by (7.9). We can then recognize that

$$
Z_{\mathrm{trumpet}}^{(b)}(\beta;P) = \eta\left(\frac{i\beta}{2\pi}\right)\chi_P^{(b)}\left(\frac{2\pi i}{\beta}\right)\,.
\tag{8.8}
$$

This is because the partition function of the Virasoro minimal string on the disk is equivalent to that of (the chiral half of) 3d gravity on the solid cylinder, which computes the corresponding Virasoro character. We get the character in the dual channel because the length of the thermal circle in 3d gravity is related to the length of the boundary disk by a modular S transformation, see section 4.4. Up to an overall scale factor, this is actually equivalent to the trumpet partition function of JT gravity for all values of $b$ (where $P$ is related to the geodesic length as in (2.22) and the inverse temperature is rescaled as in (2.23)).

**The empty disk diagram.** To compute the empty disk diagram in Virasoro minimal string theory, and hence the disk partition function, we appeal to the dilaton equation (4.15a). The dilaton equation implies that the empty (marked) disk diagram is given by the following difference of punctured disk diagrams

$$
Z_{\mathrm{disk}}^{(b)}\left(s;P=\tfrac{iQ}{2}\right) - Z_{\mathrm{disk}}^{(b)}\left(s;P=\tfrac{i\widehat{Q}}{2}\right) = \rho_0^{(b)}(s)\,.
\tag{8.9}
$$

Thus the disk partition function in Virasoro minimal string theory is given by

$$
Z_{\mathrm{disk}}^{(b)}(\beta) = \int_0^\infty \mathrm{d}s\, e^{-\beta s^2}\rho_0^{(b)}(s) = \sqrt{\frac{2\pi}{\beta}}\left(e^{\frac{\pi^2 Q^2}{\beta}} - e^{\frac{\pi^2\widehat{Q}^2}{\beta}}\right) = \eta\left(\frac{i\beta}{2\pi}\right)\chi_{(1,1)}^{(b)}\left(\frac{2\pi i}{\beta}\right)\,.
\tag{8.10}
$$

As indicated in the last line, this is equivalent to the Virasoro vacuum character in the dual channel with the descendant-counting eta function stripped off.

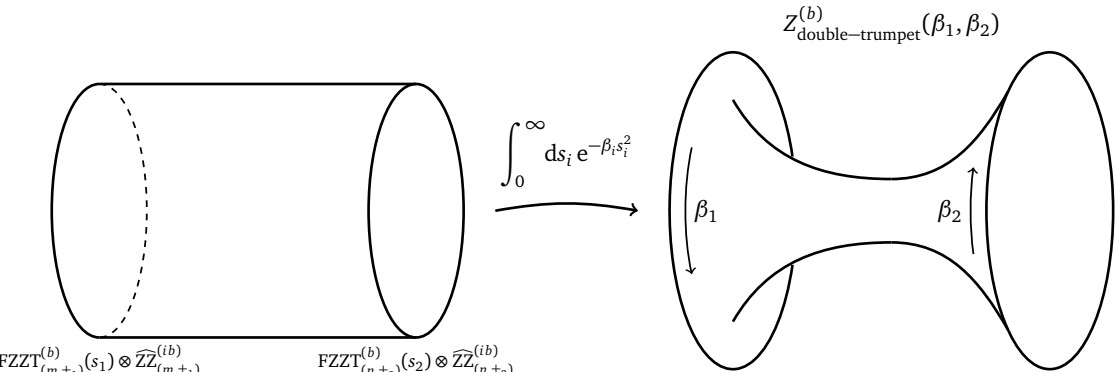

Figure 17: The Laplace transform of the cylinder diagram in Virasoro minimal string theory with FZZT($s_1$) and FZZT($s_2$) boundary conditions together with half-ZZ boundary conditions on the two ends computes the partition function on the double-trumpet, with asymptotic boundaries of renormalized lengths $\beta_1$ and $\beta_2$.

**Cylinder diagram: The double-trumpet.** We now discuss the computation of the double-trumpet partition function from the worldsheet in Virasoro minimal string theory. We start by considering the cylinder diagram with $(s_1, s_2)$ FZZT boundary conditions on the spacelike Liouville theory and any combination of half-ZZ boundary conditions on the timelike Liouville theory, subject to the marking prescription (8.3). For concreteness in what follows we will put $(m, +)$ and $(n, +)$ half-ZZ boundary conditions on the timelike Liouville CFT, but we emphasize that the analysis for any other combination proceeds similarly. The relationship between the cylinder diagram and the double-trumpet partition function is recapitulated in figure 17. The (marked) cylinder diagram is computed as the following integral of the cylinder partition functions of the ghost, spacelike Liouville and timelike Liouville CFTs over the modulus $t$ [21]

$$
\begin{aligned}
Z_{\text{cylinder}}^{(b)}(s_1, s_2) &= \int_0^\infty dt\, \eta(it)^2 \int_0^\infty dP\, \rho_0^{(b)}(P) \Psi_{(m,+)}^{(b)}(s_1; P) \Psi_{(n,+)}^{(b)}(s_2; P) \chi_P^{(b)}(it) Z_{(m,+;n,+)}^{(ib)}(t) \\
&= \sqrt{2} \int_0^\infty dt \int_0^\infty dP\, \eta(it)^2 \frac{\cos(4\pi s_1 P)\cos(4\pi s_2 P)}{\sinh(2\pi b P)\sinh(2\pi b^{-1} P)} \chi_P^{(b)}(it) \\
&\quad \times \sum_{r\overset{2}{=}|m-n|+1}^{m+n-1} \sum_{s\overset{2}{=}1}^{\infty} \chi_{(r,s)}^{(ib)}\left(\tfrac{i}{t}\right) \left(\frac{P\rho_0^{(b)}(P)}{\sqrt{2}\sinh(2\pi m b P)}\right) \left(\frac{P\rho_0^{(b)}(P)}{\sqrt{2}\sinh(2\pi n b P)}\right),
\end{aligned}
\tag{8.11}
$$

where $Z_{(m,+;n,+)}^{(ib)}$ is given in (3.37). We then exchange the integral over the cylinder modulus with that over the Liouville momentum $P$ and use the following identity [22]

$$
\sum_{r\overset{2}{=}|m-n|+1}^{m+n-1} \sum_{s\overset{2}{=}1}^{\infty} \int_0^\infty dt\, \eta(it)^2 \chi_P^{(b)}(it) \chi_{(r,s)}^{(ib)}\left(\tfrac{i}{t}\right) = \frac{\sinh(2\pi m b |P|)\sinh(2\pi n b |P|)}{\sqrt{2}|P|\sinh(2\pi b |P|)\sinh(2\pi b^{-1}|P|)},
\tag{8.12}
$$

where the characters are defined in (3.25) and (3.27) respectively. We then arrive at the following simple expression for the cylinder diagram

$$
Z_{\text{cylinder}}^{(b)}(s_1, s_2) = \int_0^\infty (2P\, dP)\left(2\sqrt{2}\cos(4\pi s_1 P)\right) \times \left(2\sqrt{2}\cos(4\pi s_2 P)\right).
\tag{8.13}
$$

---

[21] Here we consider a cylinder of length $\pi t$ and unit radius. We should also note that there is no counterterm on the annulus since it admits a flat metric. Thus there is no need to introduce a further arbitrary normalization $C_{A^2}$.

[22] In arriving at this identity we have implicitly assumed that $b^2 < \frac{1}{m-n+1}$. For $n = m$ this is always satisfied for the relevant values of the central charge.

Notice that this is entirely independent of $b$. This universality is expected given the duality with the double-scaled matrix integral. Indeed, although formally divergent as written, it is as expected simply the result of gluing two punctured disk diagrams (corresponding to trumpet partition functions in the fixed-length basis) together with the measure $2P\,\mathrm{d}P$. This also justifies our marking procedure given by (8.3). A similar calculation leads to the same result for the $(m,+;n,-)$ and $(m,-;n,-)$ assignment of half-ZZ boundary conditions for the timelike Liouville sector.

The double-trumpet partition function $Z_{0,2}^{(b)}$ in Virasoro minimal string theory is computed by transforming the marked cylinder diagram (8.11) to the fixed-length basis via the Laplace transform (8.2). We find the following universal result

$$
\begin{aligned}
Z_{0,2}^{(b)}(\beta_1,\beta_2) &= \int_0^\infty \mathrm{d}s_1 \int_0^\infty \mathrm{d}s_2\, e^{-\beta_1 s_1^2 - \beta_2 s_2^2} Z_{\text{cylinder}}^{(b)}(s_1,s_2) \\
&= \frac{2\pi}{\sqrt{\beta_1\beta_2}} \int_0^\infty (2P\,\mathrm{d}P)\, e^{-4\pi^2 P^2 \left(\frac{1}{\beta_1}+\frac{1}{\beta_2}\right)} = \frac{\sqrt{\beta_1\beta_2}}{2\pi(\beta_1+\beta_2)}.
\end{aligned}
\tag{8.14}
$$

This is of course equivalent to the result of gluing two trumpet partition functions according to (2.18).

Let us remark that the final results in this section are always independent in the end of the choice of $(m,\pm)$ for the half-ZZ boundary condition in the timelike Liouville sector. We take this to mean that these boundary conditions, while different in the worldsheet theory, are equivalent in the full string theory, i.e. after taking the BRST cohomology on the worldsheet. For the case of the minimal string, a similar phenomenon occurs [30].

## 8.2 ZZ-instantons on the worldsheet

We now turn our attention towards the computation of non-perturbative corrections to the partition function.[23] As anticipated in section 6.1 from the matrix integral, they are given by ZZ-instantons on the worldsheet. We shall discuss the case $b \neq 1$, since the case $b = 1$ has further zero-modes and is much more subtle.

We shall start by discussing the appropriate boundary conditions for such ZZ-instantons. The boundary condition should not involve any continuous parameters and thus the most general choice is to take the direct product of boundary states

$$
|\mathrm{ZZ}_{(m,n)}^{(b)}\rangle \otimes |\widehat{\mathrm{ZZ}}_{(k,\pm)}^{(ib)}\rangle,
\tag{8.15}
$$

which were introduced in section 3.2. We shall later restrict attention to a subset of these.

The quantum volume $V_{g,n}^{(b)}(P_1,\ldots,P_n)$ receives non-perturbative corrections of order $\exp(-e^{S_0})$ from each ZZ-instanton boundary condition, which themselves admit a perturbative expansion schematically of the form

$$
\exp\!\Big(\;\bigcirc\; +\; \ominus\; +\; \ominus\!\!\ominus\; +\; \bigcirc\!\!\bigcirc\; + \cdots \Big)
$$

$$
\times\left[\;\otimes\cdots\otimes\; +\; \otimes\!\otimes\cdot\otimes\cdots\otimes\; +\; \otimes\!\bigcirc\cdot\otimes\cdots\otimes\; + \cdots \right].
\tag{8.16}
$$

All boundaries of the diagram end on the same ZZ-instanton boundary conditions labelled by $(m,n)$ and $(k,\pm)$. We will focus our attention to the leading non-perturbative correction.

---

[23]This matching of the leading non-perturbative effects in the Virasoro matrix integral to those of half-ZZ instantons on the string worldsheet has been independently observed by [135].

Counting powers of the string coupling according to the Euler characteristic, only the disk and cylinder diagram contribute to this order in the exponential, while we also only keep the product of $n$ once-punctured disk diagrams. Thus to leading order, the non-perturbative correction reads

$$\exp\left(\bigcirc + \bigcirc\right)\otimes\cdots\otimes. \tag{8.17}$$

We will thus discuss the computation of the punctured disk diagram and the cylinder diagram in the following. The empty disk diagram can be obtained from the punctured disk diagram by resorting to the dilaton equation as in section 8.1.

**The punctured disk.** As in section 8.1, the punctured disk is given by the product of the wavefunctions,

$$\begin{aligned}
Z_{\text{disk}}^{(b)}(m,n,k,\pm;P) &= \frac{1}{2\sqrt{2}}\,\Psi_{(m,n)}^{(b)}(P)\,\widehat{\Psi}_{(k,\pm)}^{(ib)}(iP) \\
&= \frac{1}{2\sqrt{2}}\,\frac{4\sinh(2\pi mbP)\sinh(2\pi nb^{-1}P)\sinh(2\pi kb^{\pm1}P)}{\sinh(2\pi bP)\sinh(2\pi b^{-1}P)\,P}.
\end{aligned} \tag{8.18}$$

The factor of $\frac{1}{2\sqrt{2}}$ comes from the normalization of the disk partition function as in (8.5), which we determined in (8.7) to be $\widetilde{C}_{\text{D}^2}\text{N} = \frac{1}{2\sqrt{2}}$. Thus there is no parameter left in this subsection to adjust.

Notice that this is a redundant basis of boundary conditions. We have for example

$$Z_{\text{disk}}^{(b)}(m,1,k,+;P) = \sum_{r=|m-k|+1}^{m+k-1} Z_{\text{disk}}^{(b)}(1,1,r,+;P). \tag{8.19}$$

Similar to [30], we take this as an indication that in the full string theory, these boundary conditions are actually BRST equivalent to each other. In particular, this motivates us to restrict to the $(1,1)$ ZZ boundary condition in the spacelike Liouville theory.[24] For these, we get the simpler answer

$$Z_{\text{disk}}^{(b)}(k,\pm;P) \equiv Z_{\text{disk}}^{(b)}(1,1,k,\pm;P) = \frac{\sqrt{2}\sinh(2\pi kb^{\pm1}P)}{P}. \tag{8.20}$$

To obtain the empty disk diagram, we apply the dilaton equation as in (8.9) and obtain

$$\begin{aligned}
Z_{\text{disk}}^{(b)}(k,\pm) &= Z_{\text{disk}}^{(b)}\left(k,\pm;P=\tfrac{iQ}{2}\right) - Z_{\text{disk}}^{(b)}\left(k,\pm;P=\tfrac{i\widehat{Q}}{2}\right) \\
&= 2\sqrt{2}\left(\frac{\sin(\pi b^{\pm1}kQ)}{Q} - \frac{\sin(\pi b^{\pm1}k\widehat{Q})}{\widehat{Q}}\right) \\
&= \frac{4\sqrt{2}(-1)^k\,b^{\pm1}\sin(\pi kb^{\pm2})}{1-b^{\pm4}}.
\end{aligned} \tag{8.21}$$

---

[24]However it seems that not all boundary conditions parametrized by $m,n,k,\pm$ can be reduced to this case in a simple way, but only boundary conditions with $m=n=1$ seem to be present in the matrix integral, at least at the level of single-instanton calculus considered in this paper. A similar result was observed in the analysis of multi-instanton effects in $c=1$ string theory of [9]. There, only the class of ZZ-instantons of type $(m,1)$ gave a non-vanishing contribution to string S-matrix elements, as deduced by matching to the dual $c=1$ matrix quantum mechanics.

**The cylinder diagram.** We can similarly compute the string cylinder diagram associated to the $(k, \pm)$ ZZ-instanton. We already computed the cylinder partition function of timelike Liouville theory with two $(k, \pm)$ boundaries on both sides in (3.37). Let us focus on the '+'-case, for which we have

$$Z_{\text{cyl}}^{(b)}(k, +) = \int_0^\infty \frac{dt}{2} \, \eta(it)^2 \chi_{(1,1)}^{(b)}\left(\tfrac{i}{t}\right) \sum_{r \overset{2}{=} 1}^{2k-1} \sum_{s \overset{2}{=} 1}^\infty \chi_{(r,s)}^{(ib)}\left(\tfrac{i}{t}\right)$$

$$= \int_0^\infty \frac{dt}{2t} \, \eta(it)^2 \chi_{(1,1)}^{(b)}(it) \sum_{r \overset{2}{=} 1}^{2k-1} \sum_{s \overset{2}{=} 1}^\infty \chi_{(r,s)}^{(ib)}(it), \tag{8.22}$$

where we mapped $t \to \frac{1}{t}$ in the second line. The ingredients are similar to (8.11): The integral over $t$ integrates over the width of the cylinder, $\eta(it)^2$ is the ghost partition function and the factor $\frac{1}{2}$ originates from the $\mathbb{Z}_2$-symmetry that exchanges the two boundaries. The volume of the U(1) automorphism group of the cylinder is 1 in these conventions.

We will continue to work with the representation in the second line of (8.22). The integral is convergent in the region $t \to 0$, which becomes obvious when writing it as

$$Z_{\text{cyl}}^{(b)}(k, +) = \int_0^\infty \frac{dt}{2t} (1 - e^{-2\pi t}) \sum_{r \overset{2}{=} 1}^{2k-1} \sum_{s \overset{2}{=} 1}^\infty e^{-\frac{\pi t}{2}((r-1)b - (s+1)b^{-1})((r+1)b - (s-1)b^{-1})}(1 - e^{-2\pi rst}), \tag{8.23}$$

since the infinite sum over $s$ is absolutely convergent and the factor $(1 - e^{-2\pi t})$ vanishes for $t \to 0$. However, the integral is divergent in the region $t \to \infty$ and this divergence is somewhat subtle. One can make sense of this integral using string field theory, as was explained in [60] for the case of the ordinary minimal string. Let us review the argument. Consider first a single term

$$\int_0^\infty \frac{dt}{2t} e^{-\frac{\pi t}{2}((r-1)b - (s+1)b^{-1})((r+1)b - (s-1)b^{-1})}(1 - e^{-2\pi t})(1 - e^{-2\pi rst}). \tag{8.24}$$

Assuming that the integral is convergent, i.e.

$$((r-1)b - (s+1)b^{-1})((r+1)b - (s-1)b^{-1}) > 0, \tag{8.25}$$

the integral over $t$ converges and can be evaluated to

$$\int_0^\infty \frac{dt}{2t} e^{-\frac{\pi t}{2}((r-1)b - (s+1)b^{-1})((r+1)b - (s-1)b^{-1})}(1 - e^{-2\pi t})(1 - e^{-2\pi rst})$$

$$= \frac{1}{2} \log \left( \frac{((r-1)b \pm (s-1)b^{-1})((r+1)b \pm (s+1)b^{-1})}{((r-1)b \pm (s+1)b^{-1})((r+1)b \pm (s-1)b^{-1})} \right), \tag{8.26}$$

where we take the product over both choices of sign in the logarithm. Within string field theory, this formula is also taken to be valid when the argument of the exponential is positive. However, in that case the argument of the logarithm might be negative and hence the branch is ambiguous. Different branches correspond to different definitions of the integration contour in the string field space.

Assuming that $b^2 \notin \mathbb{Q}$, this deals with all cases $(r, s) \neq (1, 1)$, where the argument of the logarithm is non-singular. In the case $(r, s) = (1, 1)$, we should compute the integral

$$\int_0^\infty \frac{dt}{2t} \left( e^{2\pi t} - 2 + e^{-2\pi t} \right), \tag{8.27}$$

which of course diverges badly and cannot be rendered finite by contour deformation. The origin of this divergence is a breakdown of the Siegel gauge-fixing condition. One can instead fix the gauge in a different way as explained by Sen [11]. We will not repeat the full string field theory analysis here, which may be found in [60], but use the result that it leads to the interpretation

$$
\int_0^\infty \frac{dt}{2t}\left(e^{2\pi t} - 2 + e^{-2\pi t}\right) = -\frac{1}{2}\log\left(-2^5\pi^3 T_{k,+}^{(b)}\right).
\tag{8.28}
$$

Here, $(T_{k,+}^{(b)})^{-\frac{1}{2}}$ is the instanton action as computed by the empty disk diagram, $T_{k,+}^{(b)} = -Z_{\mathrm{disk}}^{(b)}(k,+)$. Again, the choice of branch cut in the logarithm is ambiguous.

Putting together the ingredients, we hence find

$$
\begin{aligned}
Z_{\mathrm{cyl}}(k,+) = &-\frac{1}{2}\log\left(-2^5\pi^3 T_{k,+}^{(b)}\right) \\
&+\frac{1}{2}\log\left(\prod_{\substack{r=1 \\ (r,s)\neq(1,1)}}^{2k-1}\prod_{s=1}^{\infty}\frac{((r-1)b \pm (s-1)b^{-1})((r+1)b \pm (s+1)b^{-1})}{((r-1)b \pm (s+1)b^{-1})((r+1)b \pm (s-1)b^{-1})}\right) \\
= &-\frac{1}{2}\log\left(-2^5\pi^3 T_{k,+}^{(b)}(1-b^4)k^2\right),
\end{aligned}
\tag{8.29}
$$

where we used that the infinite product telescopes.

**The leading ZZ-instanton correction to the quantum volumes.** It is now simple to compute the leading ZZ-instanton correction to the resummed quantum volumes (2.8). The leading ZZ-instanton correction takes the form

$$
\begin{aligned}
V_n^{(b)}(S_0; P_1,\ldots,P_n)_{k,\pm}^{[1]} &= \exp\left(e^{S_0}Z_{\mathrm{disk}}^{(b)}(k,\pm) + Z_{\mathrm{cyl}}^{(b)}(k,\pm)\right)\prod_{j=1}^{n}Z_{\mathrm{disk}}^{(b)}(k,\pm;P_j) \\
&= \frac{i\,e^{-T_{k,\pm}^{(b)}}}{2^{\frac{5}{2}}\pi^{\frac{3}{2}}(T_{k,\pm}^{(b)})^{\frac{1}{2}}(1-b^{\pm 4})^{\frac{1}{2}}k}\prod_{j=1}^{n}\frac{\sqrt{2}\sinh(2\pi k b^{\pm 1}P_j)}{P_j},
\end{aligned}
\tag{8.30}
$$

with

$$
T_{k,\pm}^{(b)} = \frac{4\sqrt{2}\,e^{S_0}b^{\pm 1}(-1)^{k+1}\sin(\pi k b^{\pm 2})}{1-b^{\pm 4}},
\tag{8.31}
$$

which matches with the value computed in the matrix model (6.10). In both formulas, the sign is ambiguous. Overall, we hence precisely reproduce (6.12), giving strong evidence for the proposal even at the non-perturbative level.

# Part IV

# Discussion

## 9  Loose ends

Let us mention some further applications of our duality and some loose ends.

**Positivity of the volumes.** For $b \in \mathbb{R}$, i.e. $c \geq 25$ and $P_j \in \mathbb{R}$, the quantum volumes are all positive, as is appropriate for "volumes". This is obvious from the worldsheet definition (2.7). Indeed, all the OPE data and conformal blocks are positive so that the integrand is positive. Hence also the volumes are positive.

In fact, something stronger is true. Writing the volumes as a Laurent polynomial in $b^2$ and a polynomial in the momenta $P_j$, all non-zero coefficients of the polynomial are positive. This follows directly recursively from the deformed Mirzakhani recursion (2.13). Indeed, all terms in the recursion come with a plus sign and all the coefficients in the basic integrals (5.37) are strictly positive. Together with the correctness of the statement for the initial conditions, the recursion proves this statement.

If we however leave the regime $c \geq 25$, then positivity of the volumes no longer holds. For large enough genus, the asymptotic formula (6.20) implies that the quantum volumes $\mathsf{V}_{g,0}^{(b)}$ have a zero near $c = 25$ and one can directly check that such a zero exists in explicit examples. For example, all the zeros of $\mathsf{V}_{12,0}^{(b)}$ lie in the interval $c \in [1, 25]$, the maximal of which is $c \approx 24.0046$.

**Dilaton equation of timelike Liouville theory.** The duality discussed in this paper has an interesting consequence purely within CFT. The path integral of timelike Liouville induced from the action (2.3b) suggests that the operator $e^{2b\chi}$ is an exactly marginal operator, just like in spacelike Liouville theory. It should merely change the value of the cosmological constant $\mu_{\text{tL}}$. From KPZ scaling [136], $\mu_{\text{tL}}$ appears in correlation functions of both types of Liouville theory as a universal prefactor raised to the Euler characteristic. The marginal operator becomes $\widehat{V}_{\hat{h}=1}$ in the quantum theory, where by a slight abuse of notation we label the operator by its conformal weight rather than its Liouville momentum. Hence the path integral formulation of the theory suggests that

$$\int d^2z \left\langle \widehat{V}_{\hat{h}=1}(z) \prod_{j=1}^{n} \widehat{V}_{\hat{P}_j}(z_j) \right\rangle_g \overset{?}{\propto} (2g-2+n) \left\langle \prod_{j=1}^{n} \widehat{V}_{\hat{P}_j}(z_j) \right\rangle_g. \tag{9.1}$$

However, this equation turns out to need refinement. The problem is that the field $\widehat{V}_{\hat{h}=1}(z)$ has singular correlation functions because the structure constant of timelike Liouville theory has a simple pole at $\hat{h} = 1$ (i.e. $\widehat{P} = \frac{1}{2}(b + b^{-1})$). We can define a residue field $\text{Res}_{\hat{h}=1} \widehat{V}_{\hat{h}}(z)$ whose correlation functions are given by the residue of the timelike Liouville correlation functions at $\hat{h} = 1$. However, the field $\text{Res}_{\hat{h}=1} \widehat{V}_{\hat{h}}(z)$ has special properties. It satisfies

$$\operatorname*{Res}_{\hat{h}=1} \widehat{V}_{\hat{h}}(z) = -\frac{1}{2} \partial \bar{\partial} \widehat{V}_{\hat{h}=0}(z). \tag{9.2}$$

Here, the field appearing on the right-hand-side is the unique primary field of conformal dimension 0 in the spectrum of timelike Liouville theory. As was discussed in the literature [56], and summarized in section 3.1, this field is however not the identity operator and in particular its derivative does not vanish. (9.2) is the analogue of the first higher equation of motion of spacelike Liouville theory [137]. It can easily be checked at the level of the three-point functions, which then ensures that (9.2) holds in any correlation function by conformal symmetry. In particular (9.2) implies that

$$\int d^2z \left\langle \operatorname*{Res}_{\hat{h}=1} \widehat{V}_{\hat{h}}(z) \prod_{j=1}^{n} \widehat{V}_{\hat{P}_j}(z_j) \right\rangle_g = 0, \tag{9.3}$$

instead of (9.1).

However, one can derive the correct version of the dilaton equation in timelike Liouville theory from the dilaton equation of the quantum volumes (4.15a). Since it holds for arbitrary

operator insertions on the worldsheet, we can remove most of the integrals on the worldsheet in (4.15a) and get an equation where we only integrate over the location of the $(n+1)$-st marked point on the LHS. We set for simplicity $n = 0$, since the other vertex operators are only spectators. We denote the partition functions by an empty correlation function $\langle 1 \rangle_g$ and $\langle \widehat{1} \rangle_g$, respectively. We get

$$
\begin{aligned}
(2g-2)\langle 1 \rangle_g \langle \widehat{1} \rangle_g &= \lim_{h \to 1} \int \mathrm{d}^2 z \, \langle V_h(z) \rangle_g \langle \widehat{V}_{1-h}(z) \rangle_g - \lim_{h \to 0} \int \mathrm{d}^2 z \, \langle V_h(z) \rangle_g \langle \widehat{V}_{1-h}(z) \rangle_g \\
&= \int \mathrm{d}^2 z \, \langle V_1(z) \rangle_g \langle \widehat{V}_0(z) \rangle_g - \lim_{h \to 0} \bigg[ -\frac{1}{h} \int \mathrm{d}^2 z \, \langle 1 \rangle_g \Big\langle \operatorname*{Res}_{\hat{h}=1} \widehat{V}_{\hat{h}}(z) \Big\rangle_g \\
&\quad - \int \mathrm{d}^2 z \, \langle V_0'(z) \rangle_g \Big\langle \operatorname*{Res}_{\hat{h}=1} \widehat{V}_{\hat{h}}(z) \Big\rangle + \int \mathrm{d}^2 z \, \langle 1 \rangle_g \Big\langle \widehat{V}_1^{\mathrm{ren}}(z) \Big\rangle_g \bigg] \\
&= \frac{1}{2} \int \mathrm{d}^2 z \, \Big( \langle \partial \bar{\partial} V_0'(z) - \tfrac{1}{4} \mathcal{R} \rangle_g \langle \widehat{V}_0(z) \rangle_g - \langle V_0'(z) \rangle_g \langle \partial \bar{\partial} \widehat{V}_0(z) \rangle_g \Big) \\
&\quad - \langle 1 \rangle_g \int \mathrm{d}^2 z \, \Big\langle \widehat{V}_1^{\mathrm{ren}}(z) \Big\rangle_g \\
&= -\langle 1 \rangle_g \int \mathrm{d}^2 z \, \Big\langle \widehat{V}_1^{\mathrm{ren}}(z) + \tfrac{1}{8} \mathcal{R} \widehat{V}_0(z) \Big\rangle_g .
\end{aligned}
\tag{9.4}
$$

In going from the first to the second line, we Laurent expanded the second term. Here we used the notation

$$
\widehat{V}_1^{\mathrm{ren}}(z) \equiv \lim_{\hat{h} \to 1} \Big[ \widehat{V}_{\hat{h}}(z) - \frac{1}{\hat{h}-1} \operatorname*{Res}_{\hat{h}=1} \widehat{V}_{\hat{h}}(z) \Big].
\tag{9.5}
$$

We also used the first higher equation of motion of ordinary Liouville theory,

$$
V_1(z) = \frac{1}{2} \partial \bar{\partial} V_0' - \tfrac{1}{8} \mathcal{R},
\tag{9.6}
$$

where $'$ denotes a derivative in the conformal weight and $\mathcal{R}$ is the Ricci curvature. The combination

$$
\Phi(z) = -\widehat{V}_1^{\mathrm{ren}}(z) - \tfrac{1}{8} \mathcal{R} \widehat{V}_0(z),
\tag{9.7}
$$

does indeed transform like a primary field of conformal weight 1, up to an inhomogeneous term that is a total derivative. We used integration by parts to cancel the two terms in the fourth line of (9.4). We thus learn that

$$
\int \mathrm{d}^2 z \, \Big\langle \Phi(z) \prod_{j=1}^n \widehat{V}_{\widehat{P}_j}(z_j) \Big\rangle_g = (2g-2+n) \Big\langle \prod_{j=1}^n \widehat{V}_{\widehat{P}_j}(z_j) \Big\rangle_g,
\tag{9.8}
$$

which is the correct version of (9.1). We did not manage to prove this equation directly in conformal field theory, but it is an interesting prediction of the present duality.

**Defect regime.** In the worldsheet description of the Virasoro minimal string (section 7), we took physical vertex operators to have $P_j \in \mathbb{R}$, i.e. the external spacelike Liouville momentum was real and the timelike Liouville momentum was imaginary. This can of course be relaxed and we can consider general complex momenta $P_j$, which should still give rise to the quantum volumes $\mathsf{V}_{g,n}^{(b)}(P_1, \ldots, P_n)$, but with complex values of the Liouville momenta. Let us reiterate however that the worldsheet moduli integrand can change non-smoothly when the external momenta $P_j$ are complexified. In particular, whenever there is a pair of momenta $P_j, P_k$ such

that $|\text{Im}(P_j \pm P_k)| > \frac{Q}{2}$, the spacelike Liouville CFT correlator may pick up additional contributions from sub-threshold states in particular OPE channels. These contributions can affect the convergence of the moduli integral and may require regularization. In these situations the string diagrams are presumably not simply the analytic continuation of the corresponding quantum volumes.

Given the relation of the Virasoro minimal string and JT gravity, one may expect that taking $P_j$ imaginary is related to the Weil-Petersson volumes of surfaces with conical defects as studied in [39, 50, 52, 54, 138]. Indeed, at least for sufficiently "sharp" defects, the corresponding Weil-Petersson volumes are simply obtained from the ordinary Weil-Petersson volumes by inserting purely imaginary values of the geodesic lengths. However, there is a subtlety. This prescription is only correct when the defects are sufficiently sharp; for blunter defects the Weil-Petersson volume changes in a non-analytic way. This mirrors the situation on the worldsheet described in the previous paragraph.

**Witten's deformations of JT gravity.** Witten proposed a duality between a large class of dilaton gravities and Hermitian matrix models [50]. The dilaton potentials in that duality are of the form

$$W(\Phi) = 2\Phi + 2 \sum_{i=1}^{r} \varepsilon_i e^{-\alpha_i \Phi}, \tag{9.9}$$

with $\pi < \alpha_i < 2\pi$. For $\sum_i \varepsilon_i = 0$, this class of dilaton gravities is described by a dual matrix model with leading density of states

$$\varrho_0(E) = \frac{e^{2\pi\sqrt{E}} W(\sqrt{E}) + e^{-2\pi\sqrt{E}} W(-\sqrt{E})}{8\pi\sqrt{E}}. \tag{9.10}$$

This formula is derived by deforming JT gravity by a gas of defects. Let us emphasize that the potential of the Virasoro minimal string is *not* of this form. Nevertheless, when plugging the sinh-dilaton potential given in eq. (2.1) into (9.9), one recovers the correct density of states of the Virasoro matrix integral (up to a rescaling of the energy). This gives some evidence that the equations of [50] hold beyond the assumptions stated above.

**Tau-scaling limit and cancellations in the quantum volumes.** Some interesting recent works [139–141] have investigated the perturbative sum over higher-genus contributions to the spectral form factor

$$\text{SFF}(T) = \sum_{g=0}^{\infty} e^{-2gS_0} \text{SFF}_g(T) = \sum_{g=0}^{\infty} e^{-2gS_0} Z_{g,n=2}(\beta + iT, \beta - iT), \tag{9.11}$$

of double-scaled matrix models and dilaton gravity models in the so-called "tau-scaling" limit, which is a late-time $T \to \infty$ limit with $T e^{-S_0}$ fixed. The linear growth of $\text{SFF}_{g=0}(T)$ at late times (the "ramp") is a universal feature of double-scaled matrix integrals, but these works argued that in the tau-scaling limit the full sum over genera in fact has a finite radius of convergence, providing perturbative access to the late time plateau of the spectral form factor. A key to this convergence is the fact that the genus-$g$ contribution to the spectral form factor only grows as $\sim T^{2g+1}$ at late times, rather than the expected $T^{3g-1}$. This slower growth is facilitated by novel cancellations due to the underlying integrable structure of the theory; in JT gravity these correspond to cancellations in the series expansion of the Weil-Petersson volumes in terms of the two geodesic lengths. In Virasoro minimal string theory, the quantum

volumes $V_{g,2}^{(b)}$ exhibit the exact same cancellations. Adapting the notation of [139] for the Weil-Petersson volumes, if one expands the quantum volumes as

$$V_{g,2}^{(b)}(P_1, P_2) = \sum_{d_1,d_2=0}^{d_1+d_2=3g-1} \frac{(4\pi^2)^{d_1}(4\pi^2)^{d_2}}{d_1! d_2!} v_{g,d_1,d_2}^{(b)} P_1^{2d_1} P_2^{2d_2}, \tag{9.12}$$

with some coefficients $v_{g,d_1,d_2}^{(b)}$, then the genus-$g$ contribution to the spectral form factor is given by gluing trumpets as in (2.18)

$$Z_{g,n=2}(\beta_1, \beta_2) = \sum_{d_1,d_2=0}^{d_1+d_2=3g-1} \frac{v_{g,d_1,d_2}^{(b)}}{8\pi^3} \beta_1^{d_1+\frac{1}{2}} \beta_2^{d_2+\frac{1}{2}}, \tag{9.13}$$

upon analytic continuation to $\beta_1 = \beta + iT$, $\beta_2 = \beta - iT$. One can indeed verify that[25]

$$\sum_{d=0}^{2q} (-1)^d v_{g,d,2q-d}^{(b)} = 0, \quad q > g, \tag{9.14}$$

leading to the expected slower late-time growth of the genus-$g$ contribution to the spectral form factor, $\mathrm{SFF}_g(T) \sim T^{2g+1}$.

**Near-extremal black holes.** Dilaton gravity is often introduced as a universal 2d theory of gravity that describes the physics of near-extremal black holes in higher dimensions. In fact this approach was used recently to successfully compute supersymmetric indices from the gravitational path integral [142–147]. In particular one can engineer also sinh-dilaton gravity from near-extremal limits of higher dimensional black holes.

From the definition, one setup is particularly straightforward. Consider an $\mathrm{AdS}_3/\mathrm{CFT}_2$ correspondence whose dual CFT is assumed to be irrational and with only Virasoro symmetry (as well as a discrete spectrum).[26] Then its torus partition function can be written as

$$Z_{\mathrm{CFT}}(\tau, \bar\tau) = \chi_{\mathrm{vac}}(\tau)\chi_{\mathrm{vac}}(-\bar\tau) + \sum_{h,\bar h>0} a_{h,\bar h} \chi_h(\tau) \chi_{\bar h}(-\bar\tau), \tag{9.15}$$

where $a_{h,\bar h}$ are positive integer degeneracies. One can take the CFT to be Lorentzian which amounts to making $\tau$ and $\bar\tau$ purely imaginary and independent, i.e. $\tau = i\beta$ and $-\bar\tau = i\bar\beta$. One can thus consider the limit $\bar\beta \to \infty$ with $\beta$ held fixed. This reduces the CFT partition function to the vacuum character, which is the disk partition function of the Virasoro minimal string. In the bulk, such a limit corresponds to a near-extremal limit of the BTZ black hole.[27] In particular, we learn that the Virasoro minimal string sits inside any irrational $\mathrm{AdS}_3/\mathrm{CFT}_2$ correspondence as a universal subsector.

**Relation to ensemble duality of 3d gravity.** The previous paragraph has in particular very concrete ramifications for the holographic dual of pure 3d gravity. It has been conjectured that 3d quantum gravity admits a holographic description in terms of an appropriate notion of an "ensemble of 2d CFTs" or "random 2d CFT" [151], and indeed many aspects of 3d gravity,

---

[25]We have checked this explicitly up to $g = 10$.

[26]Below we actually make the slightly stronger assumption that there is a nonzero gap in the spectrum of twists of non-vacuum Virasoro primaries.

[27]Usually, one considers a combined semiclassical and near-extremal limit in which $\bar\beta \sim c \to \infty$ combined with the further limit $\beta \lesssim c^{-1}$, where the model reduces to the Schwarzian or JT gravity in the bulk [51, 148]. At large $c$, the validity of this approximation requires a further sparseness assumption on the spectrum of the theory [149, 150].

particularly Euclidean wormhole partition functions, are nontrivially reproduced by statistical averages over 2d CFT data [152,153]. The precise nature of such an ensemble description remains elusive (but see [154] for recent progress), and many Euclidean wormhole partition functions may instead be interpreted in terms of coarse-graining microscopic data of individual CFTs [155–157]. The Virasoro minimal string now leads to the concrete prediction that the near-extremal limit as defined in the previous paragraph of the random ensemble of 2d CFTs is governed by the Virasoro matrix integral. This in particular lends further credence to the idea that 3d gravity is described holographically via a suitable ensemble of 2d CFTs.

## 10 Future directions

**Supersymmetric Virasoro minimal string.** A natural extension of the Virasoro minimal string would be to incorporate worldsheet supersymmetry. For $\mathcal{N} = 1$ supersymmetry, space-like Liouville theory is a unitary superconformal field theory with central charge $c \geq \frac{27}{2}$. Whereas the structure constants of $\mathcal{N} = 1$ spacelike Liouville theory have been bootstrapped (see e.g. [158–160]), the $\mathcal{N} = 1$ timelike counterpart with $\hat{c} \leq \frac{3}{2}$ has not been discussed much in the literature (see however [49,161] for a discussion of supersymmetric timelike Liouville theory from a path integral perspective). The spectrum and structure constants of supersymmetric timelike Liouville theory have not been explored. It would be interesting to understand whether a relation similar to (3.6) exists also in the supersymmetric case.

We expect that the $\mathcal{N} = 1$ supersymmetric Virasoro minimal string, defined as the worldsheet superconformal field theory

$$\begin{array}{c} c \geq \frac{27}{2} \, \mathcal{N} = 1 \\ \text{Liouville CFT} \end{array} \oplus \begin{array}{c} \hat{c} \leq \frac{3}{2} \, \mathcal{N} = 1 \\ \text{Liouville CFT} \end{array} \oplus \mathfrak{bc}\text{-ghosts} \oplus \beta\gamma\text{-ghosts}, \qquad (10.1)$$

also admits a dual matrix model description. As explained in [109] for the case of super JT gravity without time reversal symmetry, there are two such theories. On the bulk side, they differ whether we weigh odd spin structures with an opposite sign with respect to even spin structures or not, corresponding to type 0A and 0B GSO projections of (10.1). The former corresponds to a matrix model with odd $N$ and the latter to a matrix model with even $N$. Both cases can be reduced to a GUE ensemble for the supercharge, see [109, eqs. (2.19) and (2.20)]. For the super Virasoro minimal string, it is natural to conjecture that the leading density of states of the dual matrix integral is given by the following universal density of states in $\mathcal{N} = 1$ SCFT:[28]

$$\rho_0^{(b)}(P) = 2\sqrt{2}\cosh(\pi b P)\cosh(\pi b^{-1} P), \qquad (10.2)$$

with the parametrization

$$c = \frac{3}{2} + 3Q^2, \quad Q = b + b^{-1}, \quad h_P = \frac{c - \frac{3}{2}}{24} + \frac{P^2}{2} + \frac{\delta}{16}, \qquad (10.3)$$

where $\delta = 0$ in the NS-sector and $\delta = 1$ in the R-sector, and $P^2$ is again identified with the energy of eigenvalues in the matrix integral. In the limit $b \to 0$, this reduces to the density of states of super JT gravity found by Stanford and Witten [109].

One can also consider $\mathcal{N} = 2$ supersymmetry. $\mathcal{N} = 2$ JT gravity was recently analyzed [162] and one can imagine coupling $\mathcal{N} = 2$ spacelike and timelike Liouville together which define a critical $\mathcal{N} = 2$ superstring. $\mathcal{N} = 2$ supersymmetric Liouville theory stands on less

---

[28]SC is grateful to Henry Maxfield for discussions explaining this formula.

firm footing. For $c > 3$ spacelike Liouville is a unitary superconformal field theory, with its timelike counterpart restricted to the regime $c < 3$. The spectrum and structure constants for neither theory have been established. However, at least the spacelike structure constants are conjecturally known via the duality to the supersymmetric $SL(2,\mathbb{R})/U(1)$ Kazama-Suzuki supercoset model [163].

**Different matrix model statistics.** There are three classes of bosonic matrix models, the GUE, GOE or GSE type. In this paper, we discussed Hermitian matrix integrals, which correspond to GUE. In the bulk, this corresponds to only summing over orientable surfaces. It is also possible to consider the other two matrix model statistics, which also involve summing over non-orientable surfaces in the bulk, possibly with a phase $(-1)^{\chi(\Sigma)}$, where $\chi(\Sigma)$ is the Euler characteristic of the surface. This was explored for JT gravity in [109]. Similarly, one can consider the different Altland-Zirnbauer classes of supersymmetric matrix models [164] which are expected to be dual to the different varieties of the supersymmetric Virasoro minimal string.

**Two spacelike Liouville theories.** In the Virasoro minimal string we combine spacelike Liouville with central charge $c \geq 25$ and timelike Liouville theory with central charge $26 - c$. Another natural 'minimal string' worldsheet is two coupled *spacelike* Liouville theories with central charges $c_+$ and $c_-$ such that $c_+ + c_- = 26$. In particular one can consider any complex central charge $c_\pm \in \mathbb{C} \backslash (-\infty, 1] \cup [25, \infty)$. This model seems to be more complicated than the Virasoro minimal string because for example the product of two DOZZ structure constants does not cancel out. Thus already the three-point function is non-trivial. The product of two DOZZ structure constants has in fact an elliptic structure with modular parameter $\tau = b^2 \in \mathbb{H}$ [43].

In the special case $c_\pm \in 13 \pm i\mathbb{R}$, one may suspect a relation to $\mathrm{dS}_3$ quantum gravity, which is described by purely imaginary central charge (up to order $\mathcal{O}(1)$ corrections) and thus this worldsheet theory seems to be more suitable to describe two-dimensional quantum gravity with a positive cosmological constant.

**Non-analytic Virasoro minimal string.** There is another variant of the Virasoro minimal string that we might call the non-analytic Virasoro minimal string. To define it, we have to specialize to the rational case $b^2 = \frac{q}{p} \in \mathbb{Q}$. Then there exists a distinct theory from timelike Liouville theory that we can consider as a matter theory. Its structure constants for real external $\widehat{P}_j$ are given by [78, 165, 166]

$$\widehat{C}_{\hat{b}}^{\text{non-ana}}(\widehat{P}_1, \widehat{P}_2, \widehat{P}_3) = \widehat{C}_{\hat{b}}(\widehat{P}_1, \widehat{P}_2, \widehat{P}_3)\sigma(\widehat{P}_1, \widehat{P}_2, \widehat{P}_3), \tag{10.4}$$

where

$$\sigma(\widehat{P}_1, \widehat{P}_2, \widehat{P}_3) = \begin{cases} 1, & \prod_{\pm,\pm} \sin \pi\left(\frac{1}{2}(p-q) + \sqrt{pq}(\widehat{P}_1 \pm \widehat{P}_2 \pm \widehat{P}_3)\right) < 0, \\ 0, & \text{else}, \end{cases} \tag{10.5}$$

and $\widehat{C}_{\hat{b}}$ are the timelike Liouville structure constants discussed in section 3.1. This matter theory is called non-analytic Liouville theory, while for the special case $\hat{c} = 1$, it is known as the Runkel-Watts theory. The non-analytic quantum volumes defined by this matter theory are presumably closely related to the quantum volumes $V_{g,n}^{(b)}(P_1, \ldots, P_n)$. However, since it is not obvious how to extend the structure constants (10.4) to complex values of $\widehat{P}_j$, the definition is at least naively restricted to the defect regime with $P_j \in i\mathbb{R}$.

**Multi-instanton effects and $g_s$-sub-leading contributions.** Another interesting direction for future research is to study non-perturbative multi-instanton effects [9, 167, 168]. A general worldsheet instanton configuration with a number of instantons of type $(k_i, \pm_i)$ in the timelike Liouville sector is expected to correspond to the non-perturbative contribution to the Virasoro matrix integral. They stem from a configuration with multiple eigenvalues integrated along the steepest descent contour of the extrema at $E^*_{k_i, \pm_i}$. This was recently considered for the minimal string [168]. Furthermore, it would be interesting to study sub-leading corrections in $g_s$ at a given instanton configuration coming from worldsheet diagrams at higher open string loop level, as depicted in (8.16), which would require a more systematic string field theory analysis [14, 20, 169].

**Off-shell topologies in 3d gravity.** The Virasoro minimal string is presumably also useful to compute certain off-shell partition functions of 3d quantum gravity. While on-shell partition functions are by now fully understood [88], it has been argued that especially Seifert manifolds play an important role in the holography of 3d gravity. In particular, it was argued in [51] that they give off-shell contributions to the 3d gravity partition function that save the negativities in the Maloney-Witten partition function [93, 170, 171] by summing up to a non-perturbative shift of the extremality bound for BTZ black holes. The negativities precisely appear in the near-extremal limit described above and thus the tool to argue for their resolution involved the reduction to JT gravity. The 3d gravity partition function on Seifert manifolds was argued to be related to the JT gravity partition function on a Riemann surface with additional insertions of conical defects at the singular points of the Seifert fibration. The Virasoro minimal string should lead to a precise refinement of this argument and thus it would be interesting to reconsider it in this new light.

**Direct derivation of the deformed Mirzakhani recursion.** We derived the deformation of Mirzakhani's recursion given by eq. (2.13) in a rather convoluted way by first finding the dual matrix model and then translating its loop equations to the deformed Mirzakhani recursion in the bulk. It would be more satisfying to give a direct derivation of the recursion relation from the worldsheet, much like Mirzakhani managed to use a generalization of McShane's identity [172] to give a direct derivation of the recursion relation [61]. For the minimal string, such a derivation is in principle available, thanks to the existence of higher equations of motions in Liouville theory [137, 173], even though it was so far only applied to low $g$ and $n$ [53, 173–175]. Higher equations of motion do not seem to help for the Virasoro minimal string since the relevant vertex operators are not degenerate. However, it is possible that techniques known from topological string theory can lead to such a direct derivation [176].

**Cohomological interpretation of the minimal string.** We found a very satisfying realization of the Virasoro minimal string in terms of intersection theory on $\overline{\mathcal{M}}_{g,n}$, see eq. (4.12). Such a clear interpretation is to our knowledge not available for the usual minimal string and it would be interesting to find one, thus potentially leading to a more direct understanding of the duality in that case.

# Acknowledgments

We would like to thank Dionysios Anninos, Aleksandr Artemev, Teresa Bautista, Raghu Mahajan, Juan Maldacena, Alex Maloney, Sridip Pal, Sylvain Ribault, Nati Seiberg, Yiannis Tsiares, Joaquin Turiaci, Herman Verlinde and Edward Witten for discussions.

**Funding information** SC was supported by the Sam B. Treiman fellowship at the Princeton Centre for Theoretical Science. LE is supported by the grant DE-SC0009988 from the U.S. Department of Energy. LE thanks the University of Amsterdam for hospitality where part of this work was carried out. BM is supported in part by the Simons Foundation Grant No. 385602 and the Natural Sciences and Engineering Research Council of Canada (NSERC), funding reference number SAPIN/00047-2020. BM also gratefully acknowledges hospitality at the Institute for Advanced Study where part of the research for this paper was performed. VAR is supported in part by the Simons Foundation Grant No. 488653 and by the Future Faculty in the Physical Sciences Fellowship at Princeton University.

# Part V

# Appendices

## A  $\psi$- and $\kappa$-classes

In this appendix, we briefly review the definition of the cohomology $\psi_i$- and $\kappa_m$-classes that enter the intersection number formula for the volumes (4.14). We refer e.g. to [177] for more details.

We always consider the cohomology with complex coefficients and will not indicate this always explicitly. One can construct $n$ line bundles $\mathbb{L}_1, \ldots, \mathbb{L}_n$ over $\overline{\mathcal{M}}_{g,n}$ whose fiber at $\Sigma_{g,n}$ is the cotangent space at the $i$-th marked point on the surface.[29] One can then take the first Chern class of these bundles and obtain the $\psi$-classes

$$\psi_i = c_1(\mathbb{L}_i). \tag{A.1}$$

Topological gravity computes the intersection number of $\psi$-classes [63]:

$$\int_{\overline{\mathcal{M}}_{g,n}} \psi_1^{d_1} \cdots \psi_n^{d_n}, \quad d_1 + \cdots + d_n = 3g - 3 + n. \tag{A.2}$$

For our purposes we also need the so-called $\kappa$-classes. Let $\pi : \overline{\mathcal{M}}_{g,n+1} \longrightarrow \overline{\mathcal{M}}_{g,n}$ be the forgetful map that forgets the location of the last marked point. The fiber of this map describes the location of the $(n+1)$-st marked point and is hence isomorphic to the Riemann surface itself. One can then take a cohomology class in $\overline{\mathcal{M}}_{g,n+1}$ and consider the pushforward to $\overline{\mathcal{M}}_{g,n}$, which means that we integrate it over the fiber of the map. For $\alpha$ a $k$-form we have

$$\pi_* \alpha = \int_{\Sigma_{g,n}} \alpha \in \mathrm{H}^{k-2}(\overline{\mathcal{M}}_{g,n}). \tag{A.3}$$

We can then define the Mumford-Morita-Miller classes $\kappa_m$ as follows:

$$\kappa_m = \pi_*(\psi_{n+1}^{m+1}). \tag{A.4}$$

Notice that $\kappa_m$ is a class in $\mathrm{H}^{2m}(\overline{\mathcal{M}}_{g,n})$. In fact, all cohomology classes we consider are even cohomology classes and thus commute.

In particular, $\kappa_1$ plays a very important role. It is a class in $\mathrm{H}^2(\overline{\mathcal{M}}_{g,n})$ and is known to represent the cohomology class of the Weil-Petersson form on a surface with cusps [178,179]:

$$\kappa_1 = \frac{1}{2\pi^2} [\omega_{\mathrm{WP}}(0, \ldots, 0)]. \tag{A.5}$$

---

[29]The definition of the line bundle on the boundary of moduli space is a bit subtle and we again refer e.g. to [177] for details.

Here, it is important that we consider the Weil-Petersson form on a surface where all the punctures are represented by cusps in the hyperbolic language. If we have a surface with geodesic boundaries, the class of the Weil-Petersson form is instead modified to [96]

$$[\omega_{\text{WP}}(\ell_1,\dots,\ell_n)] = 2\pi^2\kappa_1 + \frac{1}{2}\sum_i \ell_i^2\psi_i. \tag{A.6}$$

# B List of quantum volumes

Let us present a list of the quantum volumes $V_{g,n}^{(b)}$ as computed by the topological recursion. We borrow the following notation from [180]

$$m_{(\ell_1,\dots,\ell_k)} = P_1^{2\ell_1}P_2^{2\ell_2}\cdots P_k^{2\ell_k} + \text{ permutations}, \tag{B.1}$$

where we sum over all distinct permutations of $(\ell_1,\ell_2,\dots,\ell_k,0,\dots,0)$ (with $n-k$ additional zeros). For example,

$$m_{(1)} = \sum_{j=1}^n P_j^2, \tag{B.2a}$$

$$m_{(1,1)} = \sum_{1\le j<k\le n} P_j^2 P_k^2, \tag{B.2b}$$

$$m_{(2,1)} = \sum_{j\ne k}^n P_j^4 P_k^2. \tag{B.2c}$$

We then have

$$V_{0,4}^{(b)} = \frac{c-13}{24} + m_{(1)}, \tag{B.3a}$$

$$V_{1,1}^{(b)} = \frac{c-13}{576} + \frac{m_{(1)}}{24}, \tag{B.3b}$$

$$V_{0,5}^{(b)} = \frac{5c^2-130c+797}{1152} + \frac{c-13}{8}m_{(1)} + \frac{m_{(2)}}{2} + 2m_{(1,1)}, \tag{B.3c}$$

$$V_{1,2}^{(b)} = \frac{(c-17)(c-9)}{9216} + \frac{c-13}{288}m_{(1)} + \frac{m_{(2)}}{48} + \frac{m_{(1,1)}}{24}, \tag{B.3d}$$

$$V_{0,6}^{(b)} = \frac{(c-13)(61c^2-1586c+9013)}{82944} + \frac{13c^2-338c+2101}{576}m_{(1)} + \frac{c-13}{8}m_{(2)}$$
$$+ \frac{c-13}{2}m_{(1,1)} + \frac{m_{(3)}}{6} + \frac{3}{2}m_{(2,1)} + 6m_{(1,1,1)}, \tag{B.3e}$$

$$V_{1,3}^{(b)} = \frac{(c-13)(7c^2-182c+967)}{497664} + \frac{13c^2-338c+2053}{27648}m_{(1)} + \frac{c-13}{288}m_{(2)} + \frac{c-13}{96}m_{(1,1)}$$
$$+ \frac{m_{(3)}}{144} + \frac{m_{(2,1)}}{24} + \frac{m_{(1,1,1)}}{12}, \tag{B.3f}$$

$$V_{2,0}^{(b)} = \frac{(c-13)(43c^2-1118c+5539)}{238878720}, \tag{B.3g}$$

$$V_{0,7}^{(b)} = \frac{6895c^4-358540c^3+6759690c^2-54565420c+158417599}{39813120}$$
$$+ \frac{5(c-13)(91c^2-2366c+13795)}{82944}m_{(1)} + \frac{5(c^2-26c+163)}{144}m_{(2)}$$
$$+ \frac{5(c^2-26c+163)}{36}m_{(1,1)} + \frac{5(c-13)}{72}m_{(3)} + \frac{5(c-13)}{8}m_{(2,1)} + \frac{5(c-13)}{2}m_{(1,1,1)}$$

$$+ \frac{m_{(4)}}{24} + \frac{2m_{(3,1)}}{3} + \frac{3m_{(2,2)}}{2} + 6m_{(2,1,1)} + 24m_{(1,1,1,1)}, \tag{B.3h}$$

$$\begin{aligned} V_{1,4}^{(b)} = {}& \frac{2645c^4 - 137540c^3 + 2562510c^2 - 20136740c + 55808069}{955514880} \\ & + \frac{(c-13)(187c^2 - 4862c + 27139)}{1990656} m_{(1)} + \frac{41c^2 - 1066c + 6593}{55296} m_{(2)} \\ & + \frac{17c^2 - 442c + 2729}{6912} m_{(1,1)} + \frac{7(c-13)}{3456} m_{(3)} + \frac{c-13}{72} m_{(2,1)} + \frac{c-13}{24} m_{(1,1,1)} \\ & + \frac{m_{(4)}}{576} + \frac{m_{(3,1)}}{48} + \frac{m_{(2,2)}}{24} + \frac{m_{(2,1,1)}}{8} + \frac{m_{(1,1,1,1)}}{4}, \end{aligned} \tag{B.3i}$$

$$\begin{aligned} V_{2,1}^{(b)} = {}& \frac{145c^4 - 7540c^3 + 138742c^2 - 1058772c + 2782913}{5096079360} \\ & + \frac{(c-13)(169c^2 - 4394c + 23713)}{159252480} m_{(1)} + \frac{139c^2 - 3614c + 22099}{13271040} m_{(2)} \\ & + \frac{29(c-13)}{829440} m_{(3)} + \frac{m_{(4)}}{27648}. \end{aligned} \tag{B.3j}$$

# C  Liouville CFT compendium

In this appendix we specify the conventions we follow for the three-point coefficients in $c \le 1$ and $c \ge 25$ Liouville CFT and list some of their properties, as well as present a brief review of the recursion relations that we employ to compute the sphere four-point and torus one-point Virasoro conformal blocks numerically.

## C.1  Liouville CFT structure constants

In our convention the structure constant for spacelike Liouville theory is given by (3.1)

$$\langle V_{P_1}(0) V_{P_2}(1) V_{P_3}(\infty) \rangle = C_b(P_1, P_2, P_3) \equiv \frac{\Gamma_b(2Q)\Gamma_b(\frac{Q}{2} \pm iP_1 \pm iP_2 \pm iP_3)}{\sqrt{2}\Gamma_b(Q)^3 \prod_{k=1}^3 \Gamma_b(Q \pm 2iP_k)}, \tag{C.1}$$

while the timelike structure constant (3.6) is given by

$$\langle \widehat{V}_{\widehat{P}_1}(0) \widehat{V}_{\widehat{P}_2}(1) \widehat{V}_{\widehat{P}_3}(\infty) \rangle = \widehat{C}_{\hat{b}}(\widehat{P}_1, \widehat{P}_2, \widehat{P}_3) = \frac{\sqrt{2}\Gamma_{\hat{b}}(\hat{b} + \hat{b}^{-1})^3 \prod_{k=1}^3 \Gamma_{\hat{b}}(\hat{b} + \hat{b}^{-1} \pm 2\widehat{P}_k)}{\Gamma_{\hat{b}}(2\hat{b} + 2\hat{b}^{-1}) \Gamma_{\hat{b}}(\frac{\hat{b} + \hat{b}^{-1}}{2} \pm \widehat{P}_1 \pm \widehat{P}_2 \pm \widehat{P}_3)}. \tag{C.2}$$

$C_b(P_1, P_2, P_3)$ is invariant under reflections $P_i \to -P_i$ and under permutations of $P_1, P_2, P_3$. The same with hatted variables holds true for $\widehat{C}_{\hat{b}}(\widehat{P}_1, \widehat{P}_2, \widehat{P}_3)$.

The double Gamma function is a meromorphic function that can be defined as the unique function satisfying the functional equations

$$\Gamma_b(z + b) = \frac{\sqrt{2\pi} \, b^{bz - \frac{1}{2}}}{\Gamma(bz)} \Gamma_b(z), \qquad \Gamma_b(z + b^{-1}) = \frac{\sqrt{2\pi} \, b^{-b^{-1}z + \frac{1}{2}}}{\Gamma(b^{-1}z)} \Gamma_b(z), \tag{C.3}$$

together with the normalization $\Gamma_b(\frac{Q}{2}) = 1$. It admits an explicit integral representation in the half-plane $\text{Re}(z) > 0$.

$$\log \Gamma_b(z) = \int_0^\infty \frac{dt}{t} \left( \frac{e^{\frac{t}{2}(Q-2z)} - 1}{4 \sinh(\frac{bt}{2}) \sinh(\frac{t}{2b})} - \frac{1}{8}(Q-2z)^2 e^{-t} - \frac{Q-2z}{2t} \right). \tag{C.4}$$

$\Gamma_b(z)$ has simple poles for

$$z = -(r-1)b - (s-1)b^{-1}, \quad r,s \in \mathbb{Z}_{\geq 1}, \tag{C.5}$$

and consequently $C_b(P_1, P_2, P_3)$ has

- **zeros** when $P_k = \pm \frac{i}{2}(rb + sb^{-1})$, $r,s \in \mathbb{Z}_{\geq 1}$, $k \in \{1,2,3\}$,

- **poles** when $\pm P_1 \pm P_2 \pm P_3 = i(r - \frac{1}{2})b + i(s - \frac{1}{2})b^{-1}$, $r,s \in \mathbb{Z}_{\geq 1}$.

The zeros are associated to the case where one of the external operators corresponds to a degenerate representation of the Virasoro algebra. On the other hand, the poles are associated with multi-twist operators in non-rational two-dimensional conformal field theory [181,182]. These poles may cross the contour of integration in the OPE of the spacelike Liouville correlator (3.5) when there exists a pair of external operators with $|\text{Im}(P_i \pm P_j)| > \frac{Q}{2}$, leading to additional discrete contributions to the conformal block decomposition. Similarly we find that the timelike structure constant $\widehat{C}_{\hat{b}}(\widehat{P}_1, \widehat{P}_2, \widehat{P}_3)$ has

- **zeros** when $\pm \widehat{P}_1 \pm \widehat{P}_2 \pm \widehat{P}_3 = (r - \frac{1}{2})\hat{b} + i(s - \frac{1}{2})\hat{b}^{-1}$, $r,s \in \mathbb{Z}_{\geq 1}$,

- **poles** when $\widehat{P}_k = \pm \frac{1}{2}(r\hat{b} + s\hat{b}^{-1})$, $r,s \in \mathbb{Z}_{\geq 1}$, $k \in \{1,2,3\}$.

Let us note the identity (see [183] for the case $m = 2$, $n = 1$)

$$\Gamma_b(z) = \lambda_{m,n,b} (mn)^{\frac{1}{4}z(Q-z)} \prod_{k=0}^{m-1} \prod_{l=0}^{n-1} \Gamma_{\frac{b\sqrt{m}}{\sqrt{n}}} \left( \frac{z + kb + lb^{-1}}{\sqrt{mn}} \right), \tag{C.6}$$

for $m, n \in \mathbb{Z}_{\geq 1}$. Here, $\lambda_{m,n,b}$ is some irrelevant constant that will cancel out of every formula we ever need, since we always have equally many $\Gamma_b$'s in the numerator and denominator.

To prove this identity, one merely need to check that the LHS satisfies the expected functional equation (C.3). Most factors on the RHS telescope and the remaining factors combine into a single Gamma-function with the help of the multiplication formula of the Gamma function, which gives the expected result. Given that

$$\Gamma_1(z) = \frac{(2\pi)^{\frac{z}{2}}}{G(z)}, \tag{C.7}$$

we hence have the following formula for $\Gamma_{\sqrt{\frac{m}{n}}}(z)$ in terms of $G(z)$:

$$\Gamma_{\sqrt{\frac{m}{n}}}(z) = \lambda_{m,n}(mn)^{\frac{1}{4}z(\frac{m+n}{\sqrt{mn}} - z)} \prod_{k=0}^{m-1} \prod_{l=0}^{n-1} \Gamma_1 \left( \frac{z}{\sqrt{mn}} + \frac{k}{m} + \frac{l}{n} \right)$$

$$= \lambda_{m,n}(2\pi)^{\frac{1}{2}\sqrt{mn}z}(mn)^{\frac{1}{4}z(\frac{m+n}{\sqrt{mn}} - z)} \prod_{k=0}^{m-1} \prod_{l=0}^{n-1} G \left( \frac{z}{\sqrt{mn}} + \frac{k}{m} + \frac{l}{n} \right)^{-1}. \tag{C.8}$$

This formula is numerically useful when computing the double Gamma function on rational values of $b^2$, since the Barnes $G$ function has efficient numerical implementations.

## C.2  Zamolodchikov recursion for conformal blocks

Let us now review the explicit recursion relations that we use to efficiently compute the sphere four-point and the torus one-point Virasoro conformal blocks, originally derived in [124] and in [121, 122], respectively.

We parametrize the central charge of the Virasoro algebra as $c = 1 + 6Q^2$ with $Q = b + b^{-1}$, and the holomorphic Virasoro weights of external primaries as $h_{P_i} = \frac{Q^2}{4} + P_i^2$. We also define

$$P_{r,s} = i\,\frac{rb + sb^{-1}}{2}, \qquad A_{r,s} = \frac{1}{2}\prod_{\substack{p=1-r \\ (p,q)\neq(0,0),(r,s)}}^{r}\prod_{q=1-s}^{s}\frac{1}{pb + qb^{-1}}\,. \tag{C.9}$$

The sphere four-point elliptic conformal block $\mathcal{H}_{0,4}^{(b)}(P_4, P_3, P_2, P_1; P|q)$ introduced in (7.19) admits a power series expansion in the elliptic nome $q(z)$, defined in (7.20), and satisfies the following recursion relation,

$$\mathcal{H}_{0,4}^{(b)}(P_i; P|q) = 1 + \sum_{r,s\geq 1}(16q)^{rs}\frac{A_{r,s}B_{r,s}(P_1, P_2)B_{r,s}(P_4, P_3)}{P^2 - P_{r,s}^2}\mathcal{H}_{0,4}^{(b)}\left(P_i; P\to(P_{r,s}^2 + rs)^{\frac{1}{2}}|q\right), \tag{C.10}$$

where the "fusion polynomials" $B_{r,s}$ are given by

$$B_{r,s}(P_1, P_2) = \prod_{p\overset{2}{=}1-r}^{r-1}\prod_{q\overset{2}{=}1-s}^{s-1}\frac{2iP_1 \pm 2iP_2 + pb + qb^{-1}}{2}\,, \tag{C.11}$$

and we take the product over both sign choices.

Similarly, the torus one-point elliptic conformal block $\mathcal{H}_{1,1}^{(b)}(P_1; P|q)$ introduced in (7.3) admits a power series expansion in $q = e^{2\pi i\tau}$ and obeys the recursion relation,

$$\mathcal{H}_{1,1}^{(b)}(P_1; P|q) = 1 + \sum_{r,s\geq 1}q^{rs}\frac{A_{r,s}B_{r,s}(P_1,(P_{r,s}^2 + rs)^{\frac{1}{2}})B_{r,s}(P_1, P_{r,s})}{P^2 - P_{r,s}^2}\mathcal{H}_{1,1}^{(b)}\left(P_1; P\to(P_{r,s}^2 + rs)^{\frac{1}{2}}|q\right)\,. \tag{C.12}$$

In this case, the product of the fusion polynomials may be written as

$$B_{r,s}(P_1,(P_{r,s}^2 + rs)^{\frac{1}{2}})B_{r,s}(P_1, P_{r,s}) = \prod_{p\overset{2}{=}1}^{2r-1}\prod_{q\overset{2}{=}1}^{2s-1}\frac{2iP_1 \pm pb \pm qb^{-1}}{2}\,, \tag{C.13}$$

where we take the product over all four sign choices.

The Liouville CFT sphere four-point functions decomposed into conformal blocks are

$$G(1234|z) \equiv \left\langle V_{P_1}(0)V_{P_2}(z,\bar{z})V_{P_3}(1)V_{P_4}(\infty)\right\rangle_{g=0}$$
$$= \int_0^\infty dP\,\rho_0^{(b)}(P)C_b(P_1, P_2, P)C_b(P_3, P_4, P)$$
$$\times \mathcal{F}_{0,4}^{(b)}(P_1, P_2, P_3, P_4; P|z)\mathcal{F}_{0,4}^{(b)}(P_1, P_2, P_3, P_4; P|\bar{z})\,, \tag{C.14a}$$

$$\widehat{G}(1234|z) \equiv \left\langle \widehat{V}_{\widehat{P}_1}(0)\widehat{V}_{\widehat{P}_2}(z,\bar{z})\widehat{V}_{\widehat{P}_3}(1)\widehat{V}_{\widehat{P}_4}(\infty)\right\rangle_{g=0}$$
$$= \int_{\mathcal{C}} d\widehat{P}\,\frac{(i\widehat{P})^2}{2\rho_0^{(\hat{b})}(i\widehat{P})}\,\widehat{C}_{\hat{b}}(\widehat{P}_1, \widehat{P}_2, \widehat{P})\widehat{C}_{\hat{b}}(\widehat{P}_3, \widehat{P}_4, \widehat{P})$$
$$\times \mathcal{F}_{0,4}^{(i\hat{b})}(\widehat{P}_1, \widehat{P}_2, \widehat{P}_3, \widehat{P}_4; \widehat{P}|z)\mathcal{F}_{0,4}^{(i\hat{b})}(\widehat{P}_1, \widehat{P}_2, \widehat{P}_3, \widehat{P}_4; \widehat{P}|\bar{z})\,. \tag{C.14b}$$

The four-point crossing symmetry relations take the form,

$$G(1234|z) = G(3214|1-z), \tag{C.15a}$$

$$\widehat{G}(1234|z) = \widehat{G}(3214|1-z), \tag{C.15b}$$

and

$$G(1234|z) = |z|^{2(h_4-h_3-h_2-h_1)}G(1324|z^{-1}), \tag{C.16a}$$

$$\widehat{G}(1234|z) = |z|^{2(\hat{h}_4-\hat{h}_3-\hat{h}_2-\hat{h}_1)}\widehat{G}(1324|z^{-1}), \tag{C.16b}$$

where $h_i = \frac{Q^2}{2} + P_i^2$ and $\hat{h}_i = -\frac{\widehat{Q}^2}{2} + \widehat{P}_i^2$. Similarly, the modular covariance of the torus one-point functions (7.2b) read,

$$\left\langle V_{P_1}(0)\right\rangle_{g=1}^{(-\frac{1}{\tau})} = |\tau|^{2h_1}\left\langle V_{P_1}(0)\right\rangle_{g=1}^{(\tau)}, \tag{C.17a}$$

$$\left\langle \widehat{V}_{\widehat{P}_1}(0)\right\rangle_{g=1}^{(-\frac{1}{\tau})} = |\tau|^{2\hat{h}_1}\left\langle \widehat{V}_{\widehat{P}_1}(0)\right\rangle_{g=1}^{(\tau)}, \tag{C.17b}$$

where $h_1 = \frac{Q^2}{2} + P_1^2$ and $\hat{h}_1 = -\frac{\widehat{Q}^2}{2} + \widehat{P}_1^2$. (C.15), (C.16) and (C.17) may be directly verified numerically using the recursion relations described in this appendix.

## D  Derivation of dilaton and string equations

In this appendix, we derive the dilaton and string equation (4.15a) and (4.15b) from the definition of the quantum volumes in terms of intersection numbers (4.14). This requires some algebraic geometry on $\overline{\mathcal{M}}_{g,n}$ which we will explain in the derivation.

### D.1  Dilaton equation

We first derive the dilaton equation (4.15a). By definition, the left-hand-side equals

$$\begin{aligned}
\text{LHS} &= \int_{\overline{\mathcal{M}}_{g,n+1}} e^{\frac{c-13}{24}\kappa_1 + \sum_i P_i^2\psi_i - \frac{c-1}{24}\psi_{n+1} - \sum_m \frac{B_{2m}}{(2m)(2m)!}\kappa_{2m}}\left(e^{\psi_{n+1}} - 1\right) \\
&= \int_{\overline{\mathcal{M}}_{g,n+1}} \psi_{n+1} \, e^{\frac{c-13}{24}(\kappa_1 - \psi_{n+1}) + \sum_i P_i^2\psi_i - \sum_m \frac{B_{2m}}{(2m)(2m)!}(\kappa_{2m} - \psi_{n+1}^{2m})}.
\end{aligned} \tag{D.1}$$

We used that we have by definition of the Bernoulli numbers

$$e^x - 1 = x \, e^{\frac{x}{2} + \sum_{m\geq 1}\frac{B_{2m}}{(2m)(2m)!}x^{2m}}, \tag{D.2}$$

as a formal power series. The strategy is now to reduce the integral over $\overline{\mathcal{M}}_{g,n+1}$ to an integral over $\overline{\mathcal{M}}_{g,n}$, which means that we want to integrate out the fiber. This is precisely the definition of the pushforward $\pi_*$ by the forgetful map $\pi : \overline{\mathcal{M}}_{g,n+1} \longrightarrow \overline{\mathcal{M}}_{g,n}$ in cohomology. Thus we need to compute the pushforward of the integrand. The pushforward interacts with the pullback via the projection formula,

$$\pi_*(\alpha \, \pi^*\beta) = (\pi_*\alpha)\beta. \tag{D.3}$$

We can use this for our integrand with $\alpha = \psi_{n+1}$. For $\beta$, we have to find a class which pulls back to the exponential. To do so, we first have to understand the behaviours of $\psi_i$ and $\kappa_m$

under pullback, which we explain here for completeness. See e.g. [177] for a more complete explanation.

We have

$$\pi^*(\psi_i) = \psi_i - \delta_{\{i,n+1\}}. \tag{D.4}$$

Here $\delta_{\{i,n+1\}}$ denotes the class in $H^2(\overline{\mathcal{M}}_{g,n+1})$ that is Poincaré dual to the boundary divisor where the $i$-th and the $(n+1)$-st point approach,

$$\delta_{\{i,n+1\}} \longleftrightarrow \qquad . \tag{D.5}$$

(D.4) follows from the fact that the line bundle $\mathbb{L}_i$ on $\mathcal{M}_{g,n}$ defining the $\psi$-classes (A.1) pulls back naturally to the corresponding line bundle on $\mathcal{M}_{g,n+1}$. However, once we pass to the compactification, we have to be careful. sections of the line bundle $\mathbb{L}_i$ are allowed to have simple poles at the boundary divisors. Since the pullback $\pi^*(\mathbb{L}_i)$ does not see the $(n+1)$-st marked point, we have to correct the formula by $\delta_{\{i,n+1\}}$ to take this into account.

One can derive the pullback of $\kappa_m$ from (D.4) as follows. Consider the maps

$$\begin{array}{ccc} & \overline{\mathcal{M}}_{g,n+2} & \\ {}^{\pi_1}\swarrow & & \searrow^{\pi_2} \\ \overline{\mathcal{M}}_{g,n+1} & & \overline{\mathcal{M}}_{g,n+1}, \end{array} \tag{D.6}$$

where $\pi_1$ forgets the $(n+1)$-st marked point and $\pi_2$ forgets the $(n+2)$-st marked point. We then have

$$\pi_1^*(\psi_{n+2}^{m+1}) = (\psi_{n+2} - \delta_{\{n+1,n+2\}})^{m+1} = \psi_{n+2}^{m+1} - (-1)^m \delta_{\{n+1,n+2\}}^{m+1}. \tag{D.7}$$

In the last step we used that the line bundle $\mathbb{L}_{n+2}$ is trivial once we restrict it to the boundary divisor defined by $\delta_{\{n+1,n+2\}}$ which implies that their product vanishes and thus there are no cross terms. We now pushforward this equation by the map $\pi_2$. For this we first have to compute

$$\begin{aligned} (\pi_2)_*(\delta_{\{n+1,n+2\}}^{m+1}) &= (\pi_2)_*\left(\delta_{\{n+1,n+2\}}^m (\psi_{n+1} - \pi_2^*(\psi_{n+1}))\right) \\ &= -(\pi_2)_*\left(\delta_{\{n+1,n+2\}}^m \pi_2^*(\psi_{n+1})\right) \\ &= -\psi_{n+1}(\pi_2)_*(\delta_{\{n+1,n+2\}}^m) \\ &= (-1)^m \psi_{n+1}^m (\pi_2)_*(\delta_{\{n+1,n+2\}}) \\ &= (-1)^m \psi_{n+1}^m. \end{aligned} \tag{D.8}$$

Here we used again the pullback (D.4) in the first line and the fact that $\psi_{n+1}\delta_{\{n+1,n+2\}} = 0$ in the second line. We then used the projection formula (D.3) and induction to reduce to the case $m = 0$. We then have $(\pi_2)_*(\delta_{\{n+1,n+2\}}) = 1$ since the corresponding divisor intersects the fiber precisely once. Combining (D.7) and (D.8) gives

$$\pi_1^* \kappa_m = \pi_1^*(\pi_2)_*(\psi_{n+2}^{m+1}) = (\pi_2)_* \pi_1^*(\psi_{n+2}^{m+1}) = \kappa_m - \psi_{n+1}^m. \tag{D.9}$$

Here we used the definition of $\kappa_m$, as well as the fact that we can commute the pullbacks and pushforwards of $\pi_1$ and $\pi_2$ since those fibers are independent. This is the desired pullback of $\kappa_m$.

Coming back to our original integrand (D.1), we realize that

$$\psi_{n+1}\,\pi^*\mathrm{e}^{\frac{c-13}{24}\kappa_1+\sum_i P_i^2\psi_i-\sum_m \frac{B_{2m}}{(2m)(2m)!}\kappa_{2m}}$$

$$=\psi_{n+1}\,\mathrm{e}^{\frac{c-13}{24}(\kappa_1-\psi_{n+1})+\sum_i P_i^2(\psi_i-\delta_{\{i,n+1\}})-\sum_m \frac{B_{2m}}{(2m)(2m)!}(\kappa_{2m}-\psi_{n+1}^{2m})}$$

$$=\psi_{n+1}\,\mathrm{e}^{\frac{c-13}{24}(\kappa_1-\psi_{n+1})+\sum_i P_i^2\psi_i-\sum_m \frac{B_{2m}}{(2m)(2m)!}(\kappa_{2m}-\psi_{n+1}^{2m})}\,,\tag{D.10}$$

where we used again that $\psi_{n+1}\,\delta_{\{i,n+1\}}=0$ and thus we can omit the boundary classes in the exponent. Hence the integrand is of the form of the projection formula (D.3). We thus have

$$\mathrm{LHS}=\int_{\overline{\mathcal{M}}_{g,n}}\pi_*\Big(\psi_{n+1}\,\pi^*\mathrm{e}^{\frac{c-13}{24}\kappa_1+\sum_i P_i^2\psi_i-\sum_m \frac{B_{2m}}{(2m)(2m)!}\kappa_{2m}}\Big)$$

$$=\pi_*(\psi_{n+1})\int_{\overline{\mathcal{M}}_{g,n}}\mathrm{e}^{\frac{c-13}{24}\kappa_1+\sum_i P_i^2\psi_i-\sum_m \frac{B_{2m}}{(2m)(2m)!}\kappa_{2m}}$$

$$=\pi_*(\psi_{n+1})\,\mathsf{V}_{g,n}^{(b)}(\mathbf{P})\,.\tag{D.11}$$

Here we used that $\pi_*(\psi_{n+1})$ has degree zero and can thus be identified with a number and taken out of the integral. The remaining integral is precisely again the definition of the quantum volume (4.14). It thus remains to compute $\pi_*(\psi_{n+1})$. By definition $\psi_{n+1}$ is the first Chern class of the line bundle $\mathbb{L}_{n+1}$. A section of $\mathbb{L}_{n+1}$ on the fiber is a holomorphic differential on the surface that is allowed to have poles at the marked points. The pushforward is then simply computing the degree of this line bundle. The degree of the canonical line bundle (the line bundle of holomorphic differentials) is known to be $2g-2$ and every marked point adds one to this. Thus

$$\pi_*(\psi_{n+1})=2g-2+n\,,\tag{D.12}$$

which finishes the proof of the dilaton equation (4.15a).

## D.2 String equation

The derivation of the string equation (4.15b) is now very similar. The left hand side is equal to

$$\mathrm{LHS}=\int_{\overline{\mathcal{M}}_{g,n+1}}\frac{\mathrm{e}^{\psi_{n+1}}-1}{\psi_{n+1}}\,\mathrm{e}^{\frac{c-13}{24}\kappa_1+\sum_i P_i^2\psi_i-\frac{c-1}{24}\psi_{n+1}-\sum_m \frac{B_{2m}}{(2m)(2m)!}\kappa_{2m}}$$

$$=\int_{\overline{\mathcal{M}}_{g,n+1}}\mathrm{e}^{\frac{c-13}{24}(\kappa_1-\psi_{n+1})+\sum_i P_i^2\psi_i-\sum_m \frac{B_{2m}}{(2m)(2m)!}(\kappa_{2m}-\psi_{n+1}^{2m})}$$

$$=\int_{\overline{\mathcal{M}}_{g,n+1}}\mathrm{e}^{\sum_i P_i^2\delta_{\{i,n+1\}}}\pi^*\mathrm{e}^{\frac{c-13}{24}\kappa_1+\sum_i P_i^2\psi_i-\sum_m \frac{B_{2m}}{(2m)(2m)!}\kappa_{2m}}\,.\tag{D.13}$$

We inserted again the definition of the Bernoulli numbers (D.2) and then used the same pull-back as above. Contrary to before, we can however not omit the boundary classes since no $\psi_{n+1}$ prefactor is present. We thus compensated for them by including them in the prefactor.

We can now pushforward to $\overline{\mathcal{M}}_{g,n}$ and use the projection formula (D.3). This gives

$$
\begin{aligned}
\text{LHS} &= \int_{\overline{\mathcal{M}}_{g,n}} \pi_* \left( e^{\sum_j P_j^2 \delta_{\{j,n+1\}}} \right) e^{\frac{c-13}{24}\kappa_1 + \sum_i P_i^2 \psi_i - \sum_m \frac{B_{2m}}{(2m)(2m)!}\kappa_{2m}} \\
&= \sum_{j=1}^n \sum_{k\geq 1} \frac{P_j^{2k}}{k!} \int_{\overline{\mathcal{M}}_{g,n}} \pi_* \left( \delta_{\{j,n+1\}}^k \right) e^{\frac{c-13}{24}\kappa_1 + \sum_i P_i^2 \psi_i - \sum_m \frac{B_{2m}}{(2m)(2m)!}\kappa_{2m}} \\
&= \sum_{j=1}^n \sum_{k\geq 1} \frac{P_j^{2k}}{k!} \int_{\overline{\mathcal{M}}_{g,n}} (-\psi_j)^{k-1} e^{\frac{c-13}{24}\kappa_1 + \sum_i P_i^2 \psi_i - \sum_m \frac{B_{2m}}{(2m)(2m)!}\kappa_{2m}} \\
&= \sum_{j=1}^n \int_{\overline{\mathcal{M}}_{g,n}} \frac{e^{P_j^2 \psi_j} - 1}{\psi_j} e^{\frac{c-13}{24}\kappa_1 + \sum_{i\neq j} P_i^2 \psi_i - \sum_m \frac{B_{2m}}{(2m)(2m)!}\kappa_{2m}} \\
&= \sum_{j=1}^n \int_0^{P_j} (2P_j \, dP_j) \int_{\overline{\mathcal{M}}_{g,n}} e^{\frac{c-13}{24}\kappa_1 + \sum_i P_i^2 \psi_i - \sum_m \frac{B_{2m}}{(2m)(2m)!}\kappa_{2m}} \\
&= \sum_{j=1}^n \int_0^{P_j} (2P_j \, dP_j) \, V_{g,n}^{(b)}(\mathbf{P}) .
\end{aligned}
\tag{D.14}
$$

Going from the first line to the second line in (D.14) we used that the divisors corresponding to $\delta_{\{i,n+1\}}$ and $\delta_{\{j,n+1\}}$ do not intersect for $i \neq j$ and thus $\delta_{\{i,n+1\}}\delta_{\{j,n+1\}} = 0$ for $i \neq j$. We can also omit the constant term in the power series expansion since $\pi_*(1) = 0$ for dimensional reasons. We then used the pushforward of the boundary classes derived in eq. (D.8). The rest is simple algebra and recognizing the definition of the quantum volume (4.14).

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
