# Peer review of "The Virasoro Minimal String"

_SciPost Physics, doi:SciPost Phys. 16, 057 (2024)_

## Round 2 · Referee Report · Sylvain Ribault (Referee 1) · 2023-10-30

Report
This article defines the Virasoro minimal string and derives many nontrivial properties using a variety of independent methods, which all agree with one another. This is very appealing in particular because the quantum volumes $V_{g,n}^{(b)}$, which are the main observables, are relatively simple polynomials that obey nice recursion relations, and can therefore be computed up to relatively high values of $g,n$, see Appendix B. This contrasts with Minimal Liouville gravity, whose matter factor is a minimal model instead of $cleq 1$ Liouville theory, leading to rather opaque formulas involving absolute values, see for instance ref. [172].
The most direct construction of the Virasoro minimal string uses worldsheet conformal field theory. However, this is not very efficient computationally, because the worldsheet correlation functions depend non-trivially on worlsheet moduli, which then have to be integrated in order to get the observables of the Virasoro minimal string. The authors only manage to do this numerically and in simple cases. Fortunately, they also find simpler descriptions in terms of a matrix integrals and intersection theory on moduli spaces. In these descriptions, the quantum volumes are manifestly polynomial in momenta, and recursion relations are derived relatively easily.
The motivations and evidence for the main results in sections 3, 4, 5 and 7 are very solid and convincing. This emboldens the authors to study non-perturbative effects in sections 6 an 8. While still impressive, the results are somewhat less transparent conceptually, and the evidence somewhat less compelling, especially in section 8. Maybe some clarifications can improve that.
When it comes to form, I would first commend the authors for presenting so much interesting material in one single coherent and well-organized text, rather than splitting it into several articles. It remains challenging to read the article in its entirety due to the variety of subjects and techniques, but the level of detail is generally appropriate. I will suggest a number of perturbative improvements.
Comments about substance.
-
The idea of coupling two spacelike Liouville theories comes a bit late on page 89, when it might be relevant to the speculation about analytic continuation in $c$ in Section 4.5, and to leaving the $c\geq 25$ regime as discussed page 82. It would be nice to tie these points together.
-
The sense in which the two- and three-point structure constants are universal, as discussed on page 19, could be made more precise. From the point of view of the conformal bootstrap, these quantities suffer from the ambiguity of renormalizing the primary fields $V_P \to \lambda(P)V_P$ in each Liouville factor. (Not to be confused with the normalization in (2.6).) Does the quantity $\rho_0^{(b)}$ have an intrinsic, normalization-independent definition in CFT? Or does the relation with the matrix integral somehow distinguish a particular normalization in Liouville theory?
-
As stated on page 25, it is certainly interesting to compare (3.16) with the leading behaviour of $\omega_{WP}$. Having done the comparison, what do we learn? There is probably the technical statement that the limits $z\to 0$ and $b\to 0$ do not commute, is there something more conceptual to say?
-
The interpretation and notation for the half-ZZ branes are not too clear. According to the wavefunctions, these look like $(m,0)$ or $(0,n)$ ZZ branes, or alternatively like derivatives of an FZZT brane at certain values of their parameter. Quite confusingly they are called $(m,\pm)$, and even compared to $(m,\infty)$ and $(\infty, n)$ ZZ branes. It might be useful to study these branes in spacelike Liouville theory. (The divergence observed in footnote 14 might make them more FZZT-like than ZZ-like?)
-
In Section 7, I am a bit skeptical that the advantage of having Barnes $G$-functions justifies the pain of having $b^2$ almost rational in computing blocks, and the ensuing loss of generality in the choice of the central charge. As is clear in (C.8), we are replacing one $\Gamma_b$ with a number of $G$s that depends on $b^2$. But $\Gamma_b$ is not that hard to compute numerically: either by the integral formula (in which case it is advantageous to compute a product of several $\Gamma_b$s as one integral) or by the product formula of Alexanian and Kuznetsov (https://arxiv.org/abs/2208.13876). Moreover, I am surprised by the discussion (footnote 19) about which integral to perform first in the triple integral: aren’t multiple integrals faster to compute together rather than sequentially?
-
Is the code for Section 7 publicly available? If not the reader would have to take the authors’ word for that section’s results. (The displayed curves would be very easy to generate without doing the hard work.) I do not consider this a major issue since there is already ample evidence for the main claims in the rest of the article.
-
The prescription (8.3) looks ad hoc. Basically, we divide by a factor such that eventually in (8.5) and (8.13) nothing depends on the boundary state in the timelike Liouville theory. Is this cancellation specific to half-ZZ branes? Or could we have chosen any boundary state in timelike Liouville theory? This phenomenon is ascribed to BRST equivalence, but it is strange that BRST equivalence would apply the timelike Liouville factor alone. This is not what the equivalence is supposed to do in the case of ZZ-instantons (8.19). These two behaviours ascribed to BRST equivalence do not seem consistent with one another, given that in ordinary spacelike Liouville theory, a ZZ-brane is a linear combination of two FZZT-branes.
-
The relation between the Virasoro minimal string (VMS) and Minimal Liouville gravity (MLG) could be exploited. Since $cleq 1$ Liouville theory correlators are limits of minimal model correlators, it should be possible to deduce the VMS from MLG. The present article does not need this for its main claims. But when it comes to boundaries and instantons, things are on less solid ground: the half-ZZ branes are basically guesses, and invoking BRST equivalence without working it out is speculative. These issues are under better control in MLG. In particular, BRST cohomology has been studied in [30]. Is it possible to take limits of MLG in order to recover boundaries and instantons in the VMS? Presumably, one could recover either the asymptotic boundaries or the instantons, depending on how one takes the limit for boundary conditions.
-
The quantum volumes $V_{g,n}^{(b)}$ are reminiscent of the numbers of lattice points in moduli spaces of curves, as studied by Do and Norbury (https://arxiv.org/abs/1012.5923). Numbers of lattice points are functions of discrete parameters instead of $P_i$. Still, their expressions (in Appendix A of that work) are quite similar to those of $V_{g,n}^{(b)}$. This might be related to the comment on Weil-Petersson volume on page 84.
Comments about form.
-
“Solubility”.
-
Figure 1: it is nice to have a road map, but it could be clearer and simpler. The relation between the worldsheet and the matrix integral is the article’s main result. According to the beginning of Section 2.6, intersection theory is the bridge between the worldsheet and the matrix integral. But this is not clearly apparent in the figure, and the meaning of the arrows is not always clear. Moreover, boxes are sometimes redundant, with no hierarchy between important topics and minor points. The boxes about boundary conditions, large $g$ asymptotics and properties of $V_{g,n}^{(b)}$ might be superfluous.
-
The material in Section 3.2 could be moved to Section 8, since it is apparently not needed in Sections 4-7.
-
The claim in Section 2.5 that the matrix integral only captures primaries, and the stripping of the $eta$ function, deserve more explanations. The intuitive explanation is OK, but it is not clear how to make it technically more precise.
-
$\Psi_j$ is missing in the first line of (2.20).
-
The claim that (2.21) determines the quantum volumes at $g=0,1$ deserves more explanations.
-
(3.1) is a product of factors, not of terms.
-
Page 24, the expression “gaps in the spectra” sounds imprecise.
-
Page 28, the statement that conformal boundaries are “relatively weakly constrained” is vague. More precise statements would be welcome, such as which consistency constraints the half-ZZ branes obey and how this compares with the FZZT and ZZ branes.
-
Page 33, the calculation (4.7) is a crucial step in the proof of the article’s main result, and could be explained more carefully, by splitting (4.7) into several formulas interspersed with text. Moreover, $\dim\mathcal{H}_{g,n}$ is infinite, and it comes back in (4.10). Both times it is stated that one could easily regularize this infinity. Would it be so tedious to actually do it?
-
(4.12) and (4.14) reproduce (2.20) (minus the typo): this should be stated explicitly.
-
Footnote 16 is probably superflous for any reader who survives that far.
-
In Section 4.5, the argument about degrees of forms that underlies the fact that the degree is $3g-3+n$ as well as equation (4.19) could be made more explicitly.
-
The end of Section 5.2 is not that clear. I am not sure about the status of equation (5.14): is it announcing a result that will later be proved? Defining a new object that will turn out to coincide with the previously known object of the same name? Moreover, it is not clear how we deduce (5.13) from (5.12).
-
The triangle diagram Figure 6 is very helpful, but it could be made a bit clearer. To begin with it would be good to choose one notation $R_{g,n}^{(b)}$ or $\omega_{g,n}^{(b)}$. Then it is a bit confusing that two inverse Laplace transforms of the resolvent yield different results. It could be helpful to write in the diagram the relevant integral transforms in some concise notation (say for only one of the variables). Then the fact that the diagram commutes does not even seem to be stated although it is not trivial. As far as I can tell it boils down to an integral identity of the type (6.5) = (6.6), which could be stated and proved explicitly, as there could be subtleties with integration contours and absolute values.
-
The definition and role of $S_0$ could be clearer. Is (5.10) its definition? It would be nice to recall what is $S_0$ at the beginning of Section 6.
-
On page 50 it is announced that the ambiguities are the same as in Section 8.2. But it is not shown in detail why the (vaguely stated) ambiguities in deforming the integration contour correspond to the (rather precise) ambiguities of sign and branch cuts in 8.2.
-
In the first sentence of 6.2, is it a typo to refer to the correction “to” $V_n^{(b)}(\dots)^{[1]}$?
-
In Figure 10, it could be made clearer that these are functions of $b$.
-
The definition of asymptotic boundaries is not clear to me. An asymptotic boundary is characterized by a renormalized length, which is computed as (8.4) for a particular type of boundary states, although it is not clear what it would be for a more generic boundary state. There is a statement that we need FZZT times ZZ-type (page 72), whose derivation (page 71) is hard to follow because it does not proceed from a precise definition of an asymptotic boundary. There is some tension between this statement, and the later summoning of BRST equivalence to say that things do not depend that much on the boundary state. Maybe FZZT times ZZ-type should be presented as an ansatz? It is called a “consistent prescription”, and later “sensible conformal boundary conditions”, but what do consistent and sensible mean in this context?
Author: Lorenz Eberhardt on 2024-01-22 [id 4274]
(in reply to Report 1 by Sylvain Ribault on 2023-10-30)See the pdf for comments and changes.
Author: Lorenz Eberhardt on 2024-01-22 [id 4273]
(in reply to Report 2 on 2023-11-24)Please see the pdf for our comments and changes.
Attachment:
Report2.pdf

---

## Round 2 · Referee Report · Anonymous (Referee 2) · 2023-11-24

Report
Dear Editor,
In this work the authors analyse and solve the string theoretic model of spacelike and timelike Liouville theory plus bc ghosts. This is an irrational variant of the minimal string studied extensively in earlier literature. They develop techniques in parallel to those studied in JT gravity to fully analyse multiboundary and higher topological contributions to calculations, including some non-perturbative effects. The paper is long and technical, combining techniques from random matrix theory and topological recursion, intersection theory, and a worldsheet analysis. The paper is impressive and detailed, and of current interest to the community.
I have several comments and questions on the work, some of which I believe are quite important:
-
I think calling this model the "Virasoro minimal string" is unnecessarily misleading, since it has nothing to do with the minimal string. The authors mention this point in footnote 5, but I still think, given the names of earlier models, this is an unfortunate and confusing name.
-
The earlier work 1704.07410 is also relevant for the translation of the sinh dilaton model to the Liouville description.
-
The authors mention in footnote 3 that earlier works focus solely on the minimal string. This is not correct. Some references (e.g. ref 38) work with general Liouville gravity, where any matter CFT is coupled to the usual spacelike Liouville and ghost CFTs. For the higher genus multiboundary analysis, the minimal string was explicitly considered in those works in order to have concrete matrix model descriptions, but most of the other results deal with generic Liouville gravity models. E.g. the correlators with boundary tachyon vertex operators were argued to hold for any choice of matter sector, where in fact the minimal string boundary correlators were discussed to have a special ´´degenerate'' structure (as discussed separately in 2007.00998 by one of those authors). The claims there would be that these are also applicable to the current situation of timelike Liouville CFT as the matter sector.
-
The main question and concern I have is about the precise density of states of the model in eq. (1.3). The authors do not explicitly discuss this, but with their choice of asymptotic boundary condition in eq (8.2), the disc partition function is essentially the Virasoro vacuum character (without the secondaries). This implies a Cardy scaling and in particular, a JT-like density of states in the thermodynamic limit: $T \sim \sqrt{E}$. However, the classical black hole solution in this dilaton gravity model has a thermodynamic relation of the form $T \sim \sqrt{E^2-1}$ (in suitable units) as discussed in ref. 38 using general results on dilaton gravity models. The classical black hole solution for the hyperbolic sine dilaton potential has the property that far away it deviates from AdS whereas deep in the interior it is AdS (and hence JT-like). This suggests that a ´´conformal'' $T(E)$ relation (as in JT and as in the vacuum Virasoro character), is not a priori expected. Perhaps one can get away with this by e.g. changing the time coordinate w.r.t. which one defines thermodynamics and/or choosing different boundary terms/conditions, but this is not a priori clear. It would be useful if the authors can clarify this point.
Relatedly, the transformation (8.2) the authors use deviates from that used in the past by ZZ and used by all papers (I am aware of) that build on this. The difference is essentially the energy variable being $s^2$ instead of $\mu_B \sim \cosh 2\pi b s$, the boundary cosmological constant of the FZZT brane. This is precisely the reason for their density of states (1.3) taking the form it does, and it not being the same as that found for generic Liouville gravity models in the literature. I would ask the authors to elaborate on where equation (8.2) comes from precisely, purely from the worldsheet or dilaton gravity perspective.
In this light, I also think section 8.1 is quite important but its placement in the paper does not suggest this. Perhaps it could be useful to have this discussion earlier on.
-
In the discussion section 9, the authors discuss the potential application to near-extremal black holes. Given the previous comments, this again is confusing to me. The sinh dilaton gravity black hole solution does not look at all like the BTZ black hole, except in the JT limit. The Ricci scalar of the sinh dilaton black hole rises without bound as one goes further away from the horizon. This is a property not found for the BTZ black hole (which has constant value of the Ricci scalar). Focussing instead then on the near-horizon region we get back to just JT gravity. Do the authors have any additional insight in this proposed regime of the BTZ black hole?
-
Whereas the authors give ample references in their work, sometimes those are not necessarily placed everywhere where they are important. E.g. equation (8.1) is well-known from mainly Teschner's work on quantum Teichm\"uller space and Liouville CFT, so a reference to these earlier works at this point can be useful.
In all, I believe the authors perform highly non-trivial calculations and obtain new and interesting results. In my opinion, the paper would benefit from a more detailed comparison with some of the previous results on Liouville gravity and sinh dilaton gravity, as highlighted by the above set of questions.

---

## Round 3 · Referee Report · Sylvain Ribault (Referee 1) · 2024-1-25

Report

The new version features a number of small improvements, mainly clarifications about subtle points, and corrections of typographical errors. The article will always remain challenging to read in detail, because of its thematic and technical breadth. However, I am satisfied that the main ideas and results are presented in a clear an convincing way --- this did not change from the previous version. I have no further modifications to suggest.

---

## Round 3 · Referee Report · Anonymous (Referee 2) · 2024-2-7

Report

The authors made adequate changes to the paper to address some comments and concerns, and I recommend publication.

---

## Editorial Decision

published